# DIVER-1 : Deep Integration of Vast Electrophysiological Recordings at Scale

## Abstract

Electrophysiology signals such as EEG and iEEG are central to neuroscience, brain–computer interfaces, and clinical applications, yet existing foundation models remain limited in scale despite clear evidence that scaling improves performance. We introduce DIVER-1, a family of EEG and iEEG foundation models trained on the largest and most diverse corpus to date—5.3k hours of iEEG and 54k hours of EEG (1.6M channel-hours from over 17.7k subjects)—and scaled up to 1.82B parameters. We present the first systematic scaling law analysis for this domain, showing that they follow data-constrained scaling laws: for a given amount of data and compute, smaller models trained for extended epochs consistently outperform larger models trained briefly. This behavior contrasts with prior electrophysiology foundation models that emphasized model size over training duration. To achieve strong performance, we also design architectural innovations including any-variate attention, sliding temporal conditional positional encoding, and multi-domain reconstruction. DIVER-1 iEEG and EEG models each achieve state-of-the-art performance on their respective benchmarks, establishing a concrete guidelines for efficient scaling and resource allocation in electrophysiology foundation model development.[1]

## 1 Introduction

Scaling has been a fundamental driver of progress in artificial intelligence, from early perceptrons to modern large language models. Systematic increases in data, compute, and model size have yielded reliable performance gains across language and vision, motivating principled investigations of neural scaling laws (Kaplan et al., 2020; Zhai et al., 2022).

Electrophysiology (Ephys)–including intracranial (iEEG) and scalp EEG–presents a distinct opportunity for foundation modeling in neuroscience, BCI, and clinical applications. The domain is marked by heterogeneity across subjects, recording sessions, montages, and neural states (Ebadi et al., 2025). Recent EEG foundation models (EFMs)[2] leverage self-supervised pretraining over large unlabeled corpora and have reported consistent improvements on downstream decoding tasks (Jiang et al., 2024; Zhang et al., 2023; Wang et al., 2024c).

Despite rapid progress, two gaps remain. First, no research has **scaled EFMs beyond the levels** necessary to fully exploit this potential—previous efforts have been limited by computational resources and dataset availability. Second, **no systematic, *quantitative* analysis of EFM scaling behavior has been conducted**, leaving fundamental questions unanswered about optimal resource allocation and scaling strategies for neural data—in particular, how to allocate a fixed compute budget across model size and number of training epochs.

We address these gaps of limited scale exploration and the lack of quantitative scaling analysis by conducting **the first systematic investigation of scaling laws in EFMs** while **pushing the boundaries of scale** across all dimensions. Specifically, we expand across: (1) **Data**—assembling pretraining corpora with more than 77× more than the previous iEEG state-of-the-art model Chau et al. (2025) and 1.2× the channel-hours of iEEG data compared to Zhang et al. (2023), plus about 10×

---

[1]Code available at: https://anonymous.4open.science/r/DIVER-1

[2]Although other terms such as large brain models (LBM) exist, we use EFM as it explicitly ties our setting to EEG (both iEEG and scalp EEG) and emphasizes that it is a foundation, not necessarily a large, model.

more EEG data than existing state-of-the-art (Wang et al., 2024c); (2) **Compute**—utilizing substantially more computational resources than previous EFM studies; and (3) **Model size**—systematically evaluating architectures ranging from 13M to 1.82B parameters, 3× the size of the largest open-source iEEG foundation model (Zhang et al., 2023)

Through this comprehensive scaling effort, we make a key discovery: **EFMs precisely follow the data-constrained scaling laws of Muennighoff et al. (2023).** Unlike classical Kaplan-style scaling (Kaplan et al., 2020; Hoffmann et al., 2022)—which assumes unlimited data and thus optimizes for one-epoch training—Ephys is inherently data-limited and requires multi-epoch training. The data-constrained framework of Muennighoff et al. (2023) generalizes Kaplan-style scaling to this setting, and our empirical scaling curves quantitatively align with its predicted isoFLOPs trade-offs and exponents. This constraint fundamentally alters the optimal scaling strategy—we find that for a fixed data and compute budget, large models (≥1B parameters) trained for only a few epochs underperform smaller models trained for extended epochs, as additional training passes enable more effective utilization of the limited data.

Building on these insights, our experiments yield **DIVER-1**, a family of EEG and iEEG EFMs that achieves state-of-the-art performance across diverse neural decoding tasks. To maximize performance, the models incorporate architectural adaptations tailored for Ephys signals. The model employs sliding temporal conditional positional encoding for context-aware positioning while preserving channel-permutation equivariance, and any-variate attention mechanisms to handle variable electrode configurations with full spatio-temporal awareness. Additionally, spatiotemporal register tokens provide dedicated computational space without interfering with signal representations, while multi-domain reconstruction heads enable robust learning across temporal and spectral views without altering the encoder backbone.

The contributions of this work are fourfold:

- **DIVER-1 model family**: We introduce a family of Ephys foundation models (EFMs) for both iEEG and EEG that achieve state-of-the-art performance in their respective modalities by leveraging our scaling insights and architectural innovations.
- **First systematic scaling law analysis for EFMs**: We provide the first quantitative characterization of how EFMs scale with data, compute, and model size, revealing that EFMs follow data-constrained scaling laws (Muennighoff et al., 2023) due to the inherent scarcity of Ephys data. Building on this analysis, we show that under a fixed data and compute budget, smaller models trained for more epochs consistently outperform larger models trained briefly—offering clear guidance on how to allocate compute between model size and training epochs for efficient EFM development.
- **Unprecedented scale demonstration**: We scale EFM pretraining to previously unattained levels across data volume, model size, and compute.
- **Novel architectural innovations for Ephys**: We develop specialized components including any-variate attention mechanisms, sliding temporal conditional positional encoding (STCPE), register tokens, and multihead prediction architectures—enabling effective scaling analysis and performance gains.

Importantly, our findings show that prior approaches emphasizing model size as the primary axis of scaling are not well aligned with the realities of Ephys data (Appendix G). A more effective path under limited compute budgets is to prioritize training duration and subject diversity, a perspective that reframes how future EFMs should be developed.

## 2 ARCHITECTURE

DIVER-1 uses an architecture custom-designed for multimodal EEG data that enables effective self-supervised pretraining through masked patch reconstruction, as described in Figure 1. This architecture consists of four main components: (1) *patch encoding*, (2) *embedding enhancement*, (3) a stack of *MOIRAI blocks* (Woo et al., 2024), and (4) *multi-output projection*. During pretraining, we randomly mask 50% of the input patches. The model then reconstructs missing information across multiple signal domains through the projection layer. The details of pretraining and architectural hyperparameters can be found in Appendix B.5.

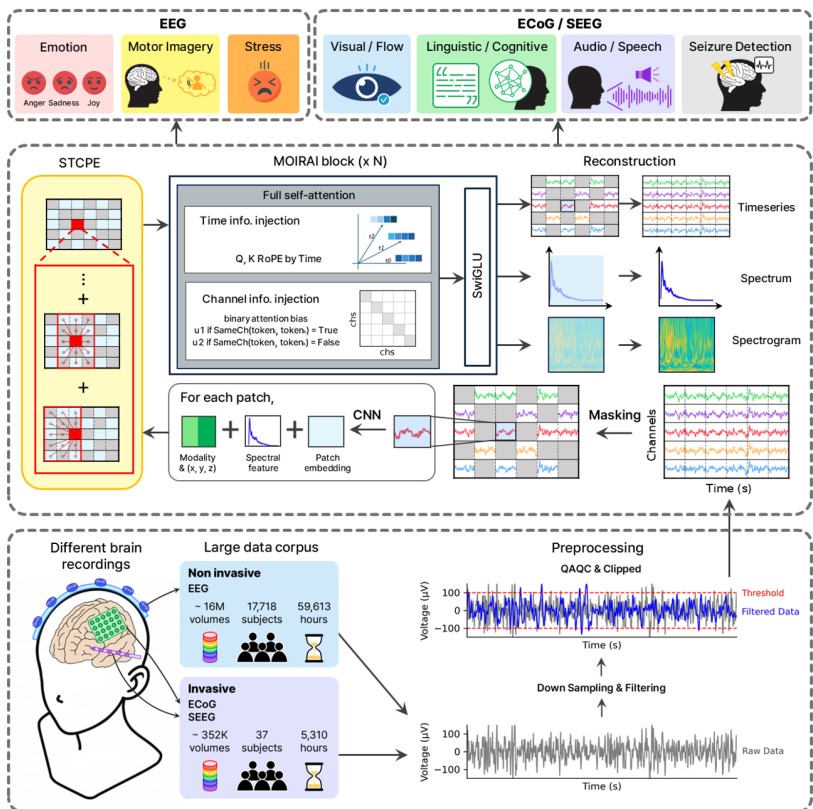

Figure 1: **Overview of DIVER-1 architecture and pretraining.** DIVER-1 is pretrained on a large EEG and iEEG data corpus. After preprocessing, input patches are randomly masked and enhanced by adding modality, spectral, and CNN-based patch embeddings, along with STCPE. The enhanced patches are processed through MOIRAI blocks and trained to reconstruct missing patches across multiple signal domains (time series, spectrum, spectrogram). The pretrained model is then applied to diverse downstream tasks.

## 2.1 STRUCTURE OVERVIEW

**Patch encoding** converts the raw signal into patch representations and extracts temporal features. Given an input time series $\mathbf{Y} \in \mathbb{R}^{C \times T}$, where $C$ is the number of channels (variables) and $T$ is the sequence length, we patchify $\mathbf{Y}$ into 1 or 0.1 second patches, each containing $P \in \{500, 50\}$ timepoints (at 500Hz sampling rate). During pretraining, $50\%$ of the patches are randomly masked with zero, and the remaining patches are passed through a three layer patch-wise convolutional neural network (CNN) to extract temporal features. This results in a basic representation $\mathbf{Y}_{\text{CNN}} \in \mathbb{R}^{C \times N \times d_{\text{model}}}$, where $N = T/P$ is the number of patches and $d_{\text{model}}$ is the embedding dimension.

**Embedding enhancement** augments the patch representations with signal and channel information through two mechanisms. First, *spectral embeddings* inject patch-wise frequency domain information, and *channel position and modality embeddings* jointly encode spatial electrode locations and electrode characteristics. Second, *spatio-temporal conditional positional embedding (STCPE)* is applied via a sliding window mechanism to capture both spatial channel relationships and local temporal information across the electrode array. The final transformer input is computed as:

$$\mathbf{X} = \mathbf{Y}_{\text{CNN}} + \mathbf{E}_{\text{spectral}} + [\mathbf{E}_{\text{position}}, \mathbf{E}_{\text{modality}}] + \mathbf{E}_{\text{STCPE}} \tag{1}$$

where $[\cdot, \cdot]$ denotes concatenation, and each embedding component $\mathbf{E}. \in \mathbb{R}^{C \times N \times d_{\text{model}}}$ maintains the same dimensional structure to ensure consistent element-wise addition.

**MOIRAI blocks** form the computational backbone of DIVER-1, adapted from MOIRAI (Woo et al., 2024) to model spatio-temporal dependencies in Ephys data. Our implementation incorpo-

rates three components: (1) *Any-Variate Attention* for improved multivariate attention, (2) *Gated Linear Units (GLU)* (Shazeer, 2020) for improved expressivity through gating mechanisms, and (3) *Spatio-Temporal Register Tokens* (our novel addition) for dedicated computational space without interfering with EEG signal representations, inspired by Darcet et al. (2023). The enhanced embedding $\mathbf{X} \in \mathbb{R}^{C \times N \times d_{\text{model}}}$ is processed through stacked MOIRAI blocks that preserve input dimensions while progressively refining spatio-temporal representations.

**Multi-output projection** enables simultaneous reconstruction across multiple signal domains during pretraining. Refined spatio-temporal representations are linearly projected to reconstruct patches in three complementary domains: raw time series, FFT, and STFT. This multi-domain reconstruction objective encourages comprehensive learning of temporal and spectral properties.

## 2.2 KEY COMPONENT DETAILS

**Embedding enhancement components**

*Spectral embeddings* $\mathbf{E}_{\text{spectral}}$ inject patch-wise frequency information by applying FFT to each patch and projecting the features to $d_{\text{model}}$ dimensions via a learnable linear layer.

*Channel position and modality embeddings* jointly encode spatial electrode locations and electrode modality to handle heterogeneous recording setups. While EEG follows standardized montages, iEEG channels are implanted in specific brain regions based on individual clinical needs. To address this heterogeneity, we encode both channel position embeddings $\mathbf{E}_{\text{position}}$ using the 3D spatial coordinates (registered to MNI space) and modality embeddings $\mathbf{E}_{\text{modality}}$ to distinguish between different electrode types. We then define the combined position–modality embedding term as:

$$[\mathbf{E}_{\text{position}}, \mathbf{E}_{\text{modality}}] = \left[ PE^{(x)}(x),\ PE^{(y)}(y),\ PE^{(z)}(z),\ \mathbf{E}_{\text{modality}}^{\text{type}} + \mathbf{E}_{\text{modality}}^{\text{subtype}} \right] \tag{2}$$

where electrode coordinates $(x, y, z)$ are in centimeters and:

$$PE_{2i}^{(j)} = \sin\left(\frac{j/256}{2000^{2i/d_j}}\right), \quad PE_{2i+1}^{(j)} = \cos\left(\frac{j/256}{2000^{2i/d_j}}\right) \tag{3}$$

for $j \in \{x, y, z\}$ and $d_x = d_y = d_z = d_{\text{model}}/4$, following PopT (Chau et al., 2025). If the coordinates x,y,z for a given channel are not available, the positional embedding for that channel is set to zero. The modality embedding $\mathbf{E}_{\text{modality}}$ is computed as $\mathbf{E}_{\text{modality}}^{\text{type}} + \mathbf{E}_{\text{modality}}^{\text{subtype}}$, where $\mathbf{E}_{\text{modality}}^{\text{type}}$ distinguishes between EEG and iEEG channels, and $\mathbf{E}_{\text{modality}}^{\text{subtype}}$ encodes electrode subtypes (grid, strip, or depth for iEEG electrodes) through learnable embedding vectors shared across all patches within the same category.

*Spatio-Temporal Conditional Positional Embedding (STCPE)* addresses the need for dynamic positional encodings that can adapt to heterogeneous electrode configurations. Transformers require positional information because self-attention is inherently permutation-invariant. While vision transformers typically rely on fixed positional encodings, *Conditional Positional Encoding* (CPE) (Chu et al., 2021) applied lightweight convolutions over local neighborhoods, allowing positional information to adapt to the structure present in each input.

Building on this encoding scheme, *Asymmetric Conditional Positional Encoding* (ACPE) (Wang et al., 2024c) applied asymmetric convolutions across channels and time for EEG, but these convolutions operate along fixed channel axes and hence does not maintain channel permutation equivariance—the learned spatial relationships depend on the training-time channel order. This is a critical limitation for Ephys, where models must be invariant to arbitrary electrode reorderings to generalize across heterogeneous montages, recording systems, and channel counts.

To address this, *STCPE* replaces convolution with a sliding-window MOIRAI transformer block that computes channel-permutation-equivariant positional encodings. After patch encoding, embeddings are first projected to a reduced dimension using $P_\downarrow : \mathbb{R}^{d_{\text{model}}} \rightarrow \mathbb{R}^{d_{\text{model}}/8}$ for computational efficiency. A temporal window of width $w$ (stride 1) is then applied to the projected sequence. Let $m = (w-1)/2$. For each window centered at index $t'$, MOIRAI receives the spatiotemporal slice $\mathbf{X}_{[:,\,t'-m:t'+m,:]} \in \mathbb{R}^{C \times w \times (d_{\text{model}}/8)}$ and produces $w$ outputs—one per relative temporal offset:

$$\mathbf{H}_{t'} = \text{MOIRAI}\left(\mathbf{X}_{[:,\,t'-m:t'+m,:]}\right) \in \mathbb{R}^{C \times w \times (d_{\text{model}}/8)}.$$

The *STCPE* embedding at absolute time $t$ aggregates contributions from all overlapping windows, and the original dimension is restored with $P_\uparrow$, yielding the final *STCPE* embedding:

$$\mathbf{E}_{\text{STCPE}}[:,t,:] = P_\uparrow\left(\sum_{k=-m}^{m} \mathbf{H}_{t+k}[:,\, k+m,:]\right) \in \mathbb{R}^{C\times w\times(d_{\text{model}})}, \qquad t = 1,\ldots,N. \quad (4)$$

*STCPE* thus provides input-dependent positional information while maintaining both **temporal translation equivariance** (via temporal sliding windows) and **channel permutation equivariance** (via MOIRAI blocks)—properties that ACPE lacks.

**MOIRAI block components**

*Any-variate attention* (Woo et al., 2024) enables adaptive spatial modeling across heterogeneous electrode configurations while maintaining full spatio-temporal attention capabilities. Unlike vanilla attention mechanisms that rely solely on input embeddings to differentiate between tokens and model their relationships, any-variate attention directly embeds spatio-temporal information into the attention computation itself through two main components: Rotary Position Embedding (RoPE) (Su et al., 2024) and binary attention bias.

Given input $\mathbf{X} \in \mathbb{R}^{C\times N\times d_{\text{model}}}$, the attention score between the $(i,m)$-th query (where $i$ denotes the patch index and $m$ denotes the channel index) and the $(j,n)$-th key is computed as $A_{ij,mn} = \frac{\exp\{E_{ij,mn}\}}{\sum_{k,o}\exp\{E_{ik,mo}\}}$, where $E_{ik,mo}$ is computed as (we omit layer and attention head indices as well as scaling factors for clarity):

$$E_{ij,mn} = (\mathbf{W}^Q \boldsymbol{x}_{i,m})^T \mathbf{R}_{i-j}(\mathbf{W}^K \boldsymbol{x}_{j,n}) + u^1 \cdot \mathbb{1}_{\{m=n\}} + u^2 \cdot \mathbb{1}_{\{m\neq n\}} \quad (5)$$

where $\mathbf{W}^Q \boldsymbol{x}_{i,m}, \mathbf{W}^K \boldsymbol{x}_{j,n} \in \mathbb{R}^{d_h}$ are the query and key vectors, $\mathbf{R}_{i-j} \in \mathbb{R}^{d_h\times d_h}$ is the rotary projection matrix encoding temporal relationships, $u^1, u^2 \in \mathbb{R}$ are learnable scalars that can differ across attention heads, and $\mathbb{1}_{\{cond\}} = 1$ if the condition is true and $0$ otherwise.

*Spatio-temporal register tokens* are inspired by register tokens in vision transformers Darcet et al. (2023). They consist of three types of learnable tokens each for channel, temporal, and combined spatio-temporal information, transforming input shape from $C \times N \times d_{\text{model}}$ to $(C+1) \times (N+1) \times d_{\text{model}}$. This transformation provides dedicated computational space to perform auxiliary computations without corrupting the primary Ephys signal representations.

## 3 TRAINING AND EXPERIMENTAL SETUP

### 3.1 PRETRAINING

DIVER-1 employs self-supervised pretraining based on masked patch reconstruction to learn robust representations of Ephys signals.

**Multi-domain reconstruction objective.** Rather than reconstructing only raw time series, DIVER-1 utilizes a multi-output projection architecture that maps each learned patch representation $\mathbf{h}_{c,n} \in \mathbb{R}^{d_{\text{model}}}$ to three complementary signal domains through parallel linear transformations. For each patch $(c,n)$, the representation is projected to: (1) raw time series $\hat{\mathbf{y}}_{c,n}^{\text{raw}} \in \mathbb{R}^P$ to reconstruct temporal dynamics, (2) FFT coefficients $\hat{\mathbf{y}}_{c,n}^{\text{FFT}} \in \mathbb{R}^{P/2+1}$ to capture frequency domain characteristics, and (3) STFT spectrogram $\hat{\mathbf{y}}_{c,n}^{\text{STFT}} \in \mathbb{R}^{F\times T_s}$ to model time-frequency relationships, with specific FFT and STFT parameters detailed in the implementation section in Appendix B.5.

The total pretraining loss aggregates reconstruction errors across all masked patches and all signal domains:

$$\mathcal{L}_{\text{total}} = \sum_{(c,n)\in\mathcal{M}} \left[\lambda_1 \mathcal{L}_{\text{MSE}}(\mathbf{y}_{c,n}^{\text{raw}}, \hat{\mathbf{y}}_{c,n}^{\text{raw}}) + \lambda_2 \mathcal{L}_{\text{MSE}}(\mathbf{y}_{c,n}^{\text{FFT}}, \hat{\mathbf{y}}_{c,n}^{\text{FFT}}) + \lambda_3 \mathcal{L}_{\text{MSE}}(\mathbf{y}_{c,n}^{\text{STFT}}, \hat{\mathbf{y}}_{c,n}^{\text{STFT}})\right] \quad (6)$$

where $\mathcal{M}$ denotes the set of masked patch indices. We used $(\lambda_{1,2,3}) = (1, 0.1, 1)$ for $P = 500$ (1 s patches) and $(1, 1, 0)$ for $P = 50$ (0.1 s patches). These coefficients were chosen so that the different

reconstruction losses operated on comparable numerical scales; for $P = 50$, the window is too short for a meaningful STFT, so only the FFT term was used. This multi-domain approach encourages the model to learn comprehensive representations that capture both temporal dynamics and spectral features essential for robust Ephys understanding.

**Input resampling.** To train the model to handle heterogeneous channel layouts and variable sequence lengths, we feed only a randomly resampled subset of each $30\,\text{s}$ training segment. A $30\,\text{s}$ window yields $N{=}30$ patches for $1\,\text{s}$ granularity and $N{=}300$ for $0.1\,\text{s}$. We then sample $C' \leq \min(C, 32)$ channels and $N' \leq 30$ temporal patches, with both $C'$ and $N'$ drawn from a scaled $\text{Beta}(3, 1)$ distribution that favors larger subsets. Capping $N'$ at 30 prevents the $0.1\,\text{s}$ model from processing the full 300-patch sequence, keeping its effective context comparable to the $1\,\text{s}$ model and avoiding excessive compute. This stochastic subsampling exposes the model to diverse channel sets and window lengths across epochs.

**Pretraining dataset.** DIVER was pretrained on, to our knowledge, the largest and most diverse collection of Ephys datasets compared to previous EFMs. Our pretraining data encompasses diverse recording conditions including task-based experiments, resting-state recordings, and sleep studies, ensuring robust representation learning across different brain states. The pretraining datasets are summarized in Table 2 with further details in Appendix F.1.

All data underwent QAQC, minimal preprocessing, and resampling with the goal of preserving as much of the original signal as possible. QAQC applied conservative amplitude clipping, removing electrodes only when $> 3.33\%$ of samples exceeded the clipping threshold and discarding whole segments only when $> 50\%$ of channels were affected, ensuring minimal data loss while preventing extreme values from destabilizing training. Preprocessing then normalized EEG and iEEG amplitudes (100 µV and 200 µV scales, respectively), applied minimal filtering (0.3–0.5 Hz high-pass, 60 Hz notch, no low-pass), and resampled all data to 500 Hz before segmenting into 30-second windows. Please refer to Appendix F.3 for more details.

**Scaling.** As DIVER-1 extends the scale of EFMs, we require methods to transfer optimal hyperparameters across different model sizes. In standard parameterizations, optimal hyperparameters are highly dependent on model width. Maximal Update Parametrization ($\mu P$) (Yang et al., 2022), by carefully scaling initializations and learning rates, ensures consistent weight update magnitudes as model width increases. This enables $\mu$Transfer, where optimal hyperparameters from small models can be directly transferred to large models, precluding expensive hyperparameter tuning.

DIVER-1 was pretrained on either 128, 48, 32 NVIDIA A100 GPUs or 32, 24, 16 H200 GPUs, depending on the experimental configuration. We tested models varying in number of parameters, from 12.72M to 1.83B as in Table 1. Additional training details are provided in Appendix section B.4. We conducted comprehensive scaling law experiments across four dimensions: compute (training epochs 1-64), model size (varying width, depth, and patch size), dataset size (1-100% of available data), and subject diversity (2-16 subjects with fixed dataset size). Detailed experimental configurations and scaling behaviors for each dimension are provided in the Appendix Table 1.

### 3.2 FINETUNING

**Benchmark datasets.** For iEEG downstream evaluation, we use Neuroprobe (Zahorodnii et al., 2025), which provides 15 auditory, visual, and language decoding tasks from naturalistic movie-watching iEEG, and MAYO dataset (Bbrinkm & Cukierski, 2014)(seizure detection). For EEG evaluation, we use: FACED (Chen et al., 2023) (emotion decoding), PhysioNet-MI (Goldberger et al., 2000) (motor-imagery classification), and MentalArithmetic (Zyma et al., 2019) (cognitive-workload decoding). Refer to Appendix F.2 for more detail.

**Baseline models.** We evaluate DIVER-1 against state-of-the-art foundation models across both modalities: LaBraM (Jiang et al., 2024) and CBraMod (Wang et al., 2024c) for EEG, and Brain-BERT (Wang et al., 2023), Population Transformer (PopT) (Chau et al., 2025) and Brant (Zhang et al., 2023) for iEEG. These transformer-based models employ self-supervised pretraining strategies; LaBraM, CBraMod, BrainBERT and Brant use masked patch reconstruction objectives while PopT applies discriminative self-supervised learning on BrainBERT embeddings.

**Finetuning method.** For iEEG downstream tasks, we evaluate both linear probing and full finetuning using a linear classifier head on the flattened token representations. Linear probing freezes the

encoder and trains only the classifier, providing a clean measure of representation quality. Full fine-tuning updates all encoder parameters together with the linear head. Details are in Appendix B.8. For EEG downstream tasks, we follow CBraMod's finetuning protocol (Wang et al., 2024c), jointly training the encoder and a three-layer MLP classifier. This MLP generally outperforms a linear head on EEG benchmarks, and depth-dependent results are provided in Appendix Table 22.

# 4 RESULTS AND DISCUSSION

## 4.1 SCALING LAW

We systematically investigate scaling laws across multiple dimensions to understand how EFMs scale with computational resources and data. Our analysis evaluates performance on (1) **pretext task loss** (reconstruction loss during self-supervised pretraining) and (2) **downstream task performance** across diverse neural decoding benchmarks. We vary traditional scaling axes (compute budget, dataset size, model size) and EFM-specific factors (training epochs, subject diversity).

**Overall scaling law validation.** Both iEEG and EEG EFMs exhibit precise neural scaling behavior consistent with established Kaplan scaling laws (Kaplan et al., 2020), as shown in Figure 2 (a-c, e-g). We fit our results to the data-constrained scaling law framework (Muennighoff et al., 2023) (see Appendix Section A.2 for detailed scaling law background), providing the first quantitative evidence that scaling EFMs across compute, dataset size, model parameters, and training epochs – a dimension largely unexplored in EFM scaling – predictably and logarithmically improves performance. The scaling relationships exhibit strong log-log fits in iEEG EFMs, with R² values of 0.8152 for patch size 1 second models and 0.7718 for patch size 0.1 second models (Appendix Table C.3), confirming the validity of the power-law scaling framework. The additional scaling result on EEG EFMs can be found at Appendix C.5.

Our empirical results follow the data-constrained scaling law (Muennighoff et al., 2023):

$$L(N, D) = \frac{A}{(N')^{\alpha}} + \frac{B}{(D')^{\beta}} + E \tag{7}$$

where $N'$ and $D'$ account for diminishing returns with more epochs:

$$D' = U_D + U_D R_D^* \Big(1 - e^{-\frac{R_D}{R_D^*}}\Big), \quad N' = U_N + U_N R_N^* \Big(1 - e^{-\frac{R_N}{R_N^*}}\Big) \tag{8}$$

where $U_D$, $R_D$, and $R_D^*$ each corresponds to unique data tokens, repetitions (epochs$-1$), and the "half-life" of repeated data. The fitted parameters are detailed in Appendix C.3.

**Standard scaling axes (model, data, compute).** We observe in Figure 2 (a-c, e-g) expected power-law relationships across standard dimensions. Model size, dataset size, and compute budget scaling logarithmically reduces loss, following Kaplan et al. (2020), though Ephys data limitations necessitate data-constrained formulation.

**EFM-specific scaling axes (epoch, subject number).** Beyond the traditional dimensions of model size, dataset size, and compute budget, we identify two novel scaling dimensions relevant to Ephys.

*Epoch scaling* in Figure 2 (d,h) shows increasing epochs improves performance across all model sizes, and larger models achieve lower loss when given sufficient repetitions. However, overly large models (e.g., XXL in Figure 2 (d) and Large in 2 (h)) perform worse at very small epoch counts, only surpassing smaller models after many more epochs—consistent with the data-constrained scaling law, where repeated passes increase the effective dataset size $D'$ required for large-capacity models.

*Subject diversity scaling* in Figure 2 (i) shows that under constant total training data volume, there exists an optimal balance between the number of subjects and data per subject.

**IsoLoss Analysis.** While increases in model size and training epochs both improve performance, they also raise compute cost, making it necessary to balance the two. To study this trade-off under fixed compute, we use the empirical IsoLoss landscape with IsoCompute curves (Figure 2(p), left), which map parameter–epoch pairs to equal FLOPs. The contours reveal a clear trend: at any fixed compute level, smaller and mid-sized models achieve lower loss than larger models. Thus, within

realistic compute budgets, allocating resources toward training smaller models longer is more effective than briefly training very large models. We revisit this in the Practical Implications section using predicted IsoLoss contours (Figure 2(p), right).

**Data Constrained Scaling Law Fitting.** Our fitted data-constrained scaling law parameters reveal important domain-specific characteristics (Appendix Table C.3). Most notably, EFMs exhibit smaller $R_N^*$ values (3.39 and 0.72 for 1s and 0.1s patch sizes, respectively) compared to language models (5.30) (Muennighoff et al., 2023). This indicates that increasing model parameters yields diminishing returns more rapidly in Ephys modeling than in language modeling, consistent with our finding that smaller models often suffice for EFM tasks.

The $R_D^*$ values exhibit interesting patch-size dependence: 8.91 for 1s patches versus 20.09 for 0.1s patches, compared to 15.38 for language models. This suggests that while 1s models experience faster diminishing returns from repeated epochs than language models, 0.1s models retain efficacy from additional epochs for longer durations. This behavior aligns with our data sampling strategy—0.1s models sample up to 3s context windows from original 30s segments, creating genuine "multiple views" of the underlying data compared to 1s models.

**Downstream performance scaling.** Following pretraining loss scaling, downstream performance (Figure 2 (i-o)) also scales across most dimensions. However, model size scaling exhibits different behavior across modalities: on iEEG tasks, larger models achieve similar performance to smaller models, while on EEG tasks, performance improves up to 813M parameters. The limited effect of model size on iEEG performance may reflect the simplicity of the binary classification tasks used in this benchmark, though comprehensive analysis are needed to fully characterize this behavior.

**Practical implications.** These findings have direct consequences for resource allocation in EFM. Because the domain is data-constrained, the optimal compute strategy differs from that of language or vision: *smaller models trained for more epochs outperform larger models trained briefly under fixed compute budgets*. The IsoLoss and IsoCompute structure in Figure 2(p) (left) indicates that large models are compute-inefficient at realistic epoch counts. Building on this observation, Figure 2(p) (right) defines a clear "compute-optimal frontier" that specifies the most efficient combinations of model size and training duration. Prior EFMs tend to fall outside this frontier, often emphasizing increased model size over additional training repetitions (estimation details in Appendix G). Our scaling framework therefore provides a principled tool for selecting compute-optimal configurations before launching expensive pretraining runs.

It should be noted, however, that when the goal is to achieve the highest possible performance for a given model size, training beyond the compute-optimal frontier is still beneficial. This was the case for our Small iEEG model, and also appears to have been the case for CBraMod (Wang et al., 2024c) and BIOT (Yang et al., 2023), which also trained substantially past their compute-optimal points.

## 4.2 DECODING PERFORMANCE

As shown in Figure 2(q,r,s,t), DIVER-1 iEEG and EEG each **achieve state-of-the-art decoding performance across iEEG and EEG benchmarks**. On the iEEG downstream dataset, our 13M-parameter model surpasses nearly all prior approaches—including BrainBERT (43M) (Wang et al., 2023), PopT (63M)(Chau et al., 2025), and Brant (506M) (Zhang et al., 2023)—across nearly all 15 tasks in Neuroprobe binaray-label and by a large margin in MAYO. For Neuroprobe Multi-label, our model still surpasses the linear STFT laplacian, which surpasses other models except ours in binary-label. On EEG downstream tasks, DIVER-1 also establishes new SOTA results, outperforming CBraMod (Wang et al., 2024c) and LaBraM (Jiang et al., 2024) on FACED emotion recognition (Chen et al., 2023), PhysioNet-MI motor imagery (Goldberger et al., 2000), and MentalArithmetic workload decoding (Zyma et al., 2019). Full task-level values for all benchmark results are provided in Appendix Section D.1. We also evaluated DIVER-1 on additional EEG downstream tasks, where we identified a methodological (reproduction) problem. To ensure fair comparison and facilitate reproduction, we conducted a controlled investigation by standardizing the finetuning procedure across models and compring performance over seven different finetuning methods. We find that under controlled finetuning settings, DIVER-1 also achieves comparable or superior performance to the other baseline models (Appendix Table 24). Detailed experimental procedures and results are provided in Appendix Section E

The strong performance of NeuroProbe results is particularly notable given the pretraining–finetuning distribution shift under which DIVER-1 was trained. Unlike PopT or BrainBERT, DIVER-1 was pretrained on different datasets than those used for finetuning, on adult data rather than pediatrics, and using both ECoG and SEEG modalities rather than exclusively SEEG data. Moreover, DIVER-1 is much smaller (13M) than the other models (BrainBERT : 43M, PopT : 63M). This highlights the robustness of our representations across dataset shifts and recording modalities.

Interestingly, linear probing outperforms full fine-tuning on Neuroprobe. The limited sample size (-1750 for each fold) may be causing overfitting during full fine-tuning. Another possibility is the relatively low difficulty of the task. However, given that linear probing also outperforms in the multi-label, more challenging targets may be needed (e.g., regression to GPT-derived embeddings).

### 4.3 ABLATION STUDIES

Because our performance gains reflect both architectural innovations and a much larger pretraining dataset, we disentangle these factors through architecture ablations and comparisons against baseline models trained on the same data.

Architecture ablations (Tables 17, 18) show that most components—any-variate attention, RoPE, STCPE, and the multi-domain reconstruction objective—generally improve performance across modalities. For iEEG specifically, several channel-wise embedding components (modality/subtype in some tasks and 3D position) produced slight improvements when removed, whereas in EEG these components typically *boosted* performance. This modality-dependent behavior suggests that future work may benefit from exploring architectures specialized for iEEG versus EEG.

To isolate architectural effects from data scale, we additionally compare models pretrained on the same dataset as existing baselines (Table 19). When pretrained on the BrainTreebank dataset used by BrainBERT and PopT, DIVER still achieves higher downstream performance, indicating that the architectural design itself yields stronger representations independent of data volume.

Further ablations and interpretability analysis are in Appendix D due to space constraints.

## 5 CONCLUSION

This work presents the first systematic investigation of scaling laws for Ephys foundation models (EFMs), introducing DIVER-1, a family of EEG/iEEG foundation models ranging from 13M to 1.82B parameters. Our analysis reveals that EFMs follow data-constrained scaling laws with critical domain-specific characteristics that fundamentally differ from language domains.

Through unprecedented scaling across data volume, model capacity, and computational resources, DIVER-1 achieves state-of-the-art performance on various iEEG and EEG tasks. Our novel architectural innovations including any-variate attention mechanisms, sliding temporal conditional positional encoding, and multi-domain reconstruction heads further enhance model performance.

DIVER also exhibits strong generality. It offers patch-size variants to accommodate different temporal scales across downstream tasks. Importantly, while it can exploit spatial location features, it is designed to function robustly even in the absence of some channel position. Unlike PoPT, BrainBERT, and Brant, which are trained exclusively on SEEG, DIVER is pretrained on both ECoG and SEEG while distinguish them by subtype embedding, and can therefore be applied to both. This versatility makes it a robust foundation model for a wide range of downstream applications.

Also, to better leverage the potential of large EFMs, future work should develop more sophisticated finetuning methodologies, including cross-subject learning frameworks that can handle heterogeneous electrode configurations across subjects, and data-efficient approaches such as LoRA that can effectively adapt large models without overfitting to limited subject-specific data.

Looking forward, these findings establish the foundation for a new generation of neuroscience AI applications by demonstrating that Ephys requires tailored scaling strategies rather than direct application of scaling laws from other domains. While the specific scaling law parameters and optimal ratios we derived are tied to our particular architectural choices and pretext task, the fundamental insight that data-constrained scaling necessitates prioritizing training duration over model size represents a broadly applicable principle for EFM development.

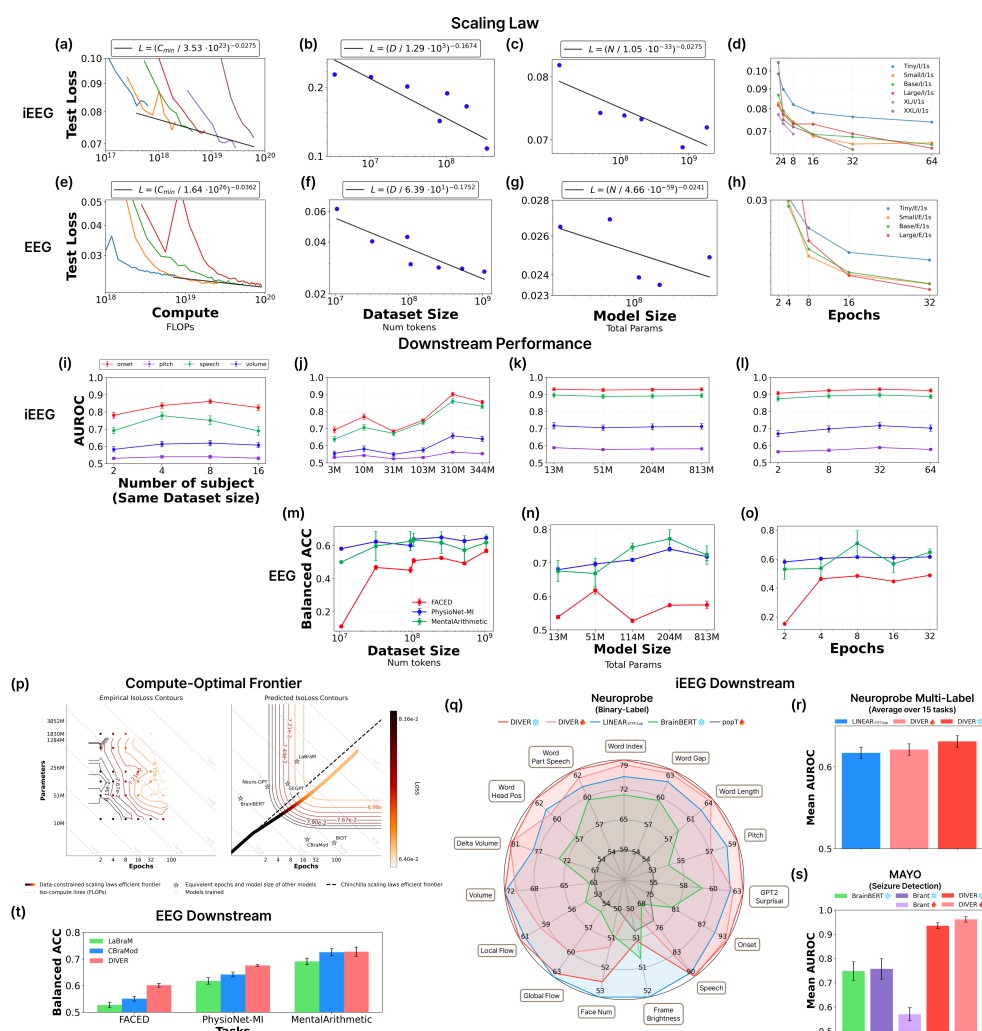

Figure 2: **Scaling laws and downstream performance of DIVER-1.** (a-h) **Scaling law valida-tion**: DIVER-1 follows data-constrained scaling laws across four dimensions for iEEG (a-d) and EEG (e-h) modalities. Loss decreases predictably with increased (a,e) compute (training FLOPs), (b,f) dataset size (number of tokens), (c,g) model size (parameters), and (d,h) training epochs, with strong log-log fits. iEEG experiments (a-d) used 100% of the dataset, while EEG experiments used 20% of the dataset for (e,h) and 100% for (f,g). (q-t) **Downstream performance**: Perfor-mance for iEEG (i-l) and EEG (m-o) across increasing (i) number of subjects while keeping dataset size identical, (j,m) dataset size (k,n) model sizes and (l,o) epochs. (p) **Compute-Optimal Fron-tier (IsoLoss analysis)**: Comparison between empirical isolation loss contours and predicted iso-lation loss contours, with model configurations plotted to show the relationship between training epochs and model parameters under fixed compute budgets. (q) **Neuroprobe benchmark results** Comprehensive performance (AUROC) comparison across multiple neural decoding tasks, with $\text{DIVER}_{\text{Tiny/I/0.1}s}$ achieving state-of-the-art or competitive results on most tasks. $\text{DIVER}_{\text{Tiny/I/0.1}s}$ with $d_{\text{model}} = 256$ and patch size 0.1s was pretrained on iEEG dataset for 32 epochs, past the com-pute optimal frontier for best performance. Performance with linear probing (red) and full finetuning (blue) are shown. (r, s) **iEEG downstream performance** (r) Neuroprobe multi-label classification results using $\text{DIVER}_{\text{Small/I/0.1}s}$. (s) MAYO(seizure detection task) results using $\text{DIVER}_{\text{Small/I/1}s}$ (t). **EEG downstream performance**: DIVER-1 showed competitive performance compared to other EEG foundation models (CBraMod and LaBraM-base) on the FACED, PhysioNet-MI, and Menta-lArithmetic datasets. Results shown are obtained using full finetuning. The DIVER model refers to $\text{DIVER}_{\text{Small/IE/1}s}$ with $d_{\text{model}} = 512$ and patch size 1s pretrained on iEEG and EEG datasets for 16 epochs. Other baseline results are replicated using their official code. Performance values for CBraMod and LaBraM are reported from their original publications.

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

## APPENDIX TABLE OF CONTENTS

## A    RELATED WORKS

### A.1    EPHYS DATA FOUNDATION MODELS

Ephys decoding has progressed from pipelines that coupled hand-crafted features with classical classifiers to end-to-end deep architectures that learn task-relevant representations directly from raw signals. Early EEG studies relied on feature engineering (e.g., band-power, CSP) paired with SVMs or LDA; subsequent work introduced convolutional backbones and sequence models that absorb spectral–temporal patterns with minimal preprocessing. Building on advances in self-supervised learning (SSL) and "foundation" paradigms, recent efforts pretrain large models on heterogeneous, weakly labeled or unlabeled EEG/iEEG corpora and adapt them to diverse downstream tasks (e.g., event detection, cognitive state decoding, BCI control). Representative lines include large-scale EEG transformers and montage-aware encoders (e.g., EEGFormer(Chen et al., 2024), NEURO-GPT(Cui et al., 2024), LaBraM(Jiang et al., 2024)), intracranial representation learners (e.g., BrainBERT(Wang et al., 2023), foundation models for intracranial neural signals), and broader neuro-sequence backbones emphasizing transfer and robustness (e.g., CBraMod(Wang et al., 2024c)).

### A.2    NEURAL SCALING LAW

**Kaplan (Kaplan et al., 2020) scaling law** provides a principled framework for predicting model performance as a function of model size and dataset size for large language models. They demonstrated that language model cross-entropy loss follows smooth power-law relationships with respect to model parameters ($N$) and training data ($D$). Concretely, they proposed a relation of the form

$$L(N, D) = \frac{A}{N^\alpha} + \frac{B}{D^\beta} + E \tag{9}$$

where $A, B, E$ are constants, and $\alpha, \beta > 0$ are scaling exponents. This formulation implies that increasing the number of parameters or training data yields a predictable reduction in loss, enabling systematic optimization of compute allocation across model size and training duration.

**Data-constrained scaling law.** In scientific domains, including Ephys, unique high-quality data are inherently limited. To address this, Muennighoff et al. (2023) extended the scaling framework to data-constrained regimes, where models must repeatedly train on the same corpus for multiple epochs. They proposed a modified law

$$L(N, D) = \frac{A}{(N')^\alpha} + \frac{B}{(D')^\beta} + E \tag{10}$$

where the effective number of parameters $N'$ and effective tokens $D'$ account for diminishing returns: repeated tokens are progressively less valuable, and excessively large models are less sample-efficient. They further define

$$D' = U_D + U_D R_D^* \left(1 - e^{-\frac{R_D}{R_D^*}}\right) \tag{11}$$

$$N' = U_N + U_N R_N^* \left(1 - e^{-\frac{R_N}{R_N^*}}\right) \tag{12}$$

where $U_D$ is the number of unique tokens, $R_D$ is the number of repetitions (epochs$-1$), and $R_D^*$ is a learned constant describing the "half-life" of repeated data. Analogously, $U_N$ is the compute-optimal parameter count for $U_D$, and $R_N^*$ governs diminishing returns beyond that point. Empirical results suggest that up to $\sim$4 epochs, repeated data is almost as useful as new data, but returns decrease sharply thereafter.

Compared to the Chinchilla law (Hoffmann et al., 2022), which assumes abundant data and one training epoch, this formulation makes epoch count itself a central axis of scaling. This perspective is crucial for domains constrained by limited samples, such as EEG and iEEG, and it guides how compute should be allocated between model size and additional training passes.

Finally, the data-constrained scaling law also connects scaling analysis to more fundamental theory. Recent work has suggested that the exponents governing power-law scaling, such as $\alpha$ and $\beta$, are intimately connected to the intrinsic dimension of the underlying data manifold (Sharma & Kaplan,

2022). In particular, it has been argued that the data exponent $\beta$ can be interpreted in terms of an effective dimension $d$ via an approximate inverse relation of the form:

$$\beta \approx \frac{2}{d}. \tag{13}$$

From this view, estimating $\beta$ provides not only a measure of how data volume translates into performance but also an insight into the intrinsic dimensionality of Ephys signals, where scarcity and structural complexity are defining features.

# B  EXPERIMENTAL SETUP DETAILS

## B.1  TESTED MODELS

We adopt a systematic naming convention for all DIVER model variants: $\text{DIVER}_{\text{Size}/\text{Modality}/\text{Granularity}}$. The *Size* component indicates model size (Tiny: 256, Small: 512, Base: 768, Large: 1024, XL: 2048, XXL: 3072 hidden dimensions). The *Modality* specifies input data types (I: iEEG-only, IE: iEEG+EEG). The *Granularity* indicates temporal resolution (0.1s or 1s window). For example, $\text{DIVER}_{\text{Base}/\text{I}/1s}$ represents a base-sized model trained on iEEG-only data with 1s temporal windows.

Table 1: **Model configurations with measured parameters and total FLOPs per epoch.**

| Models | # Parameters | Modality | Granularity | Hidden Dimension ($d_{\text{model}}$) | total FLOPs / epoch |
|---|---|---|---|---|---|
| $\text{DIVER}_{\text{Tiny}/\text{I}/0.1s}$ | 12.72M | iEEG | 0.1s | 256 | 76.34P |
| $\text{DIVER}_{\text{Small}/\text{I}/0.1s}$ | 50.75M | iEEG | 0.1s | 512 | 253.96P |
| $\text{DIVER}_{\text{Base}/\text{I}/0.1s}$ | 114.07M | iEEG | 0.1s | 768 | 532.77P |
| $\text{DIVER}_{\text{Large}/\text{I}/0.1s}$ | 202.70M | iEEG | 0.1s | 1024 | 912.83P |
| $\text{DIVER}_{\text{XL}/\text{I}/0.1s}$ | 810.19M | iEEG | 0.1s | 2048 | 3.40E |
| $\text{DIVER}_{\text{XXL}/\text{I}/0.1s}$ | 1.82B | iEEG | 0.1s | 3072 | 7.50E |
| $\text{DIVER}_{\text{Tiny}/\text{I}/1s}$ | 13.03M | iEEG | 1s | 256 | 77.52P |
| $\text{DIVER}_{\text{Small}/\text{I}/1s}$ | 51.36M | iEEG | 1s | 512 | 256.44P |
| $\text{DIVER}_{\text{Base}/\text{I}/1s}$ | 115.00M | iEEG | 1s | 768 | 536.82P |
| $\text{DIVER}_{\text{Large}/\text{I}/1s}$ | 203.95M | iEEG | 1s | 1024 | 918.56P |
| $\text{DIVER}_{\text{XL}/\text{I}/1s}$ | 812.85M | iEEG | 1s | 2048 | 3.46E |
| $\text{DIVER}_{\text{XXL}/\text{I}/1s}$ | 1.83B | iEEG | 1s | 3072 | 7.64E |
| $\text{DIVER}_{\text{Tiny}/\text{E}/1s}$ | 13.03M | EEG | 1s | 256 | 238.83P |
| $\text{DIVER}_{\text{Small}/\text{E}/1s}$ | 51.36M | EEG | 1s | 512 | 790.01P |
| $\text{DIVER}_{\text{Base}/\text{E}/1s}$ | 115.00M | EEG | 1s | 768 | 1.65E |
| $\text{DIVER}_{\text{Large}/\text{E}/1s}$ | 203.95M | EEG | 1s | 1024 | 2.83E |
| $\text{DIVER}_{\text{XL}/\text{E}/1s}$ | 812.85M | EEG | 1s | 2048 | 10.65E |
| $\text{DIVER}_{\text{Tiny}/\text{IE}/1s}$ | 13.03M | iEEG+EEG | 1s | 256 | 316.35P |
| $\text{DIVER}_{\text{Small}/\text{IE}/1s}$ | 51.36M | iEEG+EEG | 1s | 512 | 1.05E |
| $\text{DIVER}_{\text{Large}/\text{IE}/1s}$ | 203.95M | iEEG+EEG | 1s | 1024 | 2.19E |
| $\text{DIVER}_{\text{XL}/\text{IE}/1s}$ | 812.85M | iEEG+EEG | 1s | 2048 | 3.75E |

## B.2  PRETRAINING DATASET

DIVER-1 was pretrained on the largest and most diverse Ephys corpus to date, with $\text{DIVER}_{-/\text{I}/-}$ trained on 352k channel-hours from 37 subjects using iEEG data (ECoG/sEEG), $\text{DIVER}_{-/\text{IE}/-}$ trained on 1.66M channel-hours from 17,718 subjects combining both iEEG and EEG modalities across multiple datasets. Pretraining dataset description is given in Table 2.

Table 2: **Summary of DIVER-1 pretraining datasets.** The datasets are categorized by modality: iEEG (including ECoG and sEEG) and EEG. DIVER$_I$ was pretrained on iEEG, DIVER$_E$ on EEG, and DIVER$_{IE}$ utilized both. Note that for the self-collected iEEG dataset in DIVER$_{IE}$, we applied stricter QAQC criteria (3.33% threshold) compared to DIVER$_I$ (50% threshold) for consistency with EEG criteria.

| Datasets | Data Type | # Subj. | Volume (channel-hours) | Duration (hours) | Sampling Rate (Hz) |
|---|---|---|---|---|---|
| **iEEG (Used in DIVER$_I$ & DIVER$_{IE}$)** | | | | | |
| AJILE12 (Peterson et al., 2022) | ECoG | 12 | 124,423 | 1,282 | 1,000 |
| Self-collected iEEG (DIVER$_I$) | ECoG/sEEG | 25 | 227,612 | 4,028 | 2,000 |
| Self-collected iEEG (DIVER$_{IE}$) | ECoG/sEEG | 25 | 144,634 | 2,844 | 2,000 |
| **EEG (Used in DIVER$_E$ & DIVER$_{IE}$)** | | | | | |
| TUEG (Obeid & Picone, 2016) | EEG | 10,874 | 422,036 | 23,178 | 250–512 |
| HBN (Shirazi et al., 2024) | EEG | 2,782 | 61,703 | 572 | 500 |
| NCHSDB (Lee et al., 2022) | EEG | 3,673 | 163,146 | 26,055 | 256–512 |
| PEERS (Kahana et al., 2023) | EEG | 364 | 870,447 | 6,964 | 500 |
| **Total** (DIVER$_I$) | iEEG | 37 | 352,035 | 5,310 | — |
| **Total** (DIVER$_E$) | EEG | 17,693 | 1,517,332 | 56,769 | — |
| **Total** (DIVER$_{IE}$) | iEEG + EEG | 17,718 | 1,661,966 | 59,613 | — |

## B.3 Downstream Task and Dataset Overview

Table 3 provides a comprehensive overview of all downstream tasks and datasets used in our evaluation. Our evaluation spans two modalities (iEEG and EEG) and covers diverse neural decoding objectives across visual, auditory, and language domains.

**iEEG tasks.** We evaluated on 15 tasks from the Neuroprobe (LITE) benchmark (Zahorodnii et al., 2025), including visual perception (frame brightness, optical flow, face detection), auditory processing (volume, pitch, delta volume), and language processing (speech decoding, word prediction, onset detection, part-of-speech tagging). The Neuroprobe dataset contains depth electrode recordings from 6 subjects with 109-120 channels per subject, originally sampled at 2048Hz and was resampled to 500Hz to match our pretraining configuration. There are both binary-label and multi-label task options for Neuroprobe. [3] In the multi-label configuration, the speech, onset, and head word position tasks remain binary, the part-of-speech task uses 6 labels, and the remaining tasks use 3 labels. Throughout this paper, unless explicitly stated otherwise as multi-label, all reported Neuroprobe results correspond to the binary-label setting. Also, we evalauted on the seizure detection test on MAYO dataset (modified the dataset in kaggle challenge (Bbrinkm & Cukierski, 2014)). The original dataset consists of 1 s samples, but to match the minimum patch length of Brant, one of the baseline models, we concatenated samples in temporal order to create 6 s samples. We evaluated the model separately for each of the 8 participants. To address the issue of having more test data than training data, we swapped the train and test sets for each participant.

**EEG tasks.** We evaluated on three EEG benchmarks: FACED (emotion recognition from 32-channel EEG, 9-class classification, 123 subjects)(Chen et al., 2023), PhysioNet-MI (motor imagery with 64 channels, 4-class classification, 109 subjects)(Goldberger et al., 2000), and MentalArithmetic (mental stress detection with 20 channels reduced to 19, 2-class classification, 36 subjects)(Zyma et al., 2019). Sampling rates were standardized to 500Hz across all datasets to ensure consistency with our pretraining setup.

## B.4 Pretraining Setup and Model Scaling

Training experiments were conducted across two high-performance computing configurations. The primary server consisted of nodes each equipped with a single 2.8 GHz AMD EPYC Milan 7543P 32-core CPU and four NVIDIA A100 GPUs, which was more heavily utilized throughout the training process. For large model variants, we additionally employed a secondary server equipped with dual Intel Xeon Platinum 8480+ processors (112 cores total) and eight NVIDIA H200 GPUs with 144GB memory each. Training experiments were conducted using either 128, 32, 8 A100 GPUs or 32, 24, 16 H200 GPUs depending on the experimental configuration. We maintained a fixed global

---

[3]The multi-label setting is currently available only in the released code and has not yet been documented in the paper.

Table 3: **Overview of downstream tasks and datasets.** Sampling rates were adjusted to 500Hz across all datasets to match the pretraining configuration. Arrows ($\rightarrow$) indicate resampling or channel selection from the original dataset.

| Modality | Task Name | Datasets | Sampling Rate | # Ch. | # Subj. | Label |
|---|---|---|---|---|---|---|
| iEEG | frame_brightness (visual)
global_flow (visual)
local_flow (visual)
face_num (visual)
volume (auditory)
pitch (auditory)
delta_volume (auditory)
speech (language)
onset (language)
gpt2_surprisal (language)
word_length (language)
word_gap (language)
word_index (language)
word_head_pos (language)
word_part_speech (language) | Neuroprobe (LITE) | $2048 \rightarrow 500$Hz | Var. (109–120) | 6 | 2-class (2-6 in mulilabel) |
| | Seizure Detection | MAYO | $5000 \rightarrow 500$Hz | Var.(16–72) | 8 | 2-class |
| EEG | Emotion Recognition | FACED | $250 \rightarrow 500$Hz | 32 | 123 | 9-class |
| | Motor Imagery | PhysioNet-MI | $160 \rightarrow 500$Hz | 64 | 109 | 4-class |
| | Mental Stress Detection | MentalArithmetic | 500Hz | $20 \rightarrow 19$ | 36 | 2-class |

batch size of 192 across all training runs, with the per-GPU batch size adjusted dynamically based on the number of nodes employed.

We varied the model size by modifying the hidden dimension of the transformer, resulting in sizes of 13M, 51M, 115M, 203M, 813M, 1.83B parameters, while keeping the depth fixed at 12 layers. This capacity adjustment leverages the benefits of $\mu$ parameterization for stable training across different model sizes. DIVER-1 was implemented on the Python 3.12.3 and Pytorch 2.6.0 + cuda version 12.4. To enhance training efficiency, we employed DeepSpeed ZeRO Stage 2, BF16 precision. Optimization was performed using a custom implementation of the DeepSpeed's MuAdam optimizer (Yang et al., 2022) with utilizing DeepSpeed's FusedAdam backend (Rasley et al., 2020) for computational efficiency and learning rate calibration. A cosine annealing learning rate scheduler with warm-up restarts was applied, with cycle length matching the total training steps and minimum learning rate set to 0.01× the initial rate.

### B.5 DIVER Architecture and Pretext Task Hyperparamter Setting

**Architecture setting** Table 4 lists the detailed architectural hyperparameter settings used for DIVER-1 pretraining.

**FFT, STFT setting** For the FFT, we used a window size of 500 time points with a sampling frequency of 500 Hz. A cutoff frequency of 200 Hz was applied, and the FFT amplitudes were converted to absolute values, normalized, and then compressed using a log(1 + x) transform. For the STFT, we employed a multi-resolution approach with window sizes of 200 and 100 time points, respectively. Each window was shifted with 50% overlap and tapered with a Hann window function. Consistent with the FFT settings, a cutoff frequency of 200 Hz was applied, and the STFT amplitudes were converted to absolute values, normalized, and compressed using the log(1 + x) transform.

### B.6 Pretraining Hyperparameter Search using $\mu$-Parameterization ($\mu$P) and $\mu$Transfer

We employed a two-stage hyperparameter optimization approach to determine optimal learning rate (lr) and weight decay (wd) values; grid search followed by optuna optimization. The search was conducted using a 50M parameter model, a 12-layer architecture with 512-dimensional attention layers trained for different modalities: iEEG, EEG, and TUEG and iEEG; DIVER$_{Small/I/1s}$, DIVER$_{Small/E/1s}$, DIVER$_{Small/IE/1s}$. All hyperparameter searches were performed over 2 epochs to balance computational efficiency with reliable performance estimation.

Table 4: **Hyperparameters for DIVER-1 pretraining.** Two model variants were trained for inputs with 1s and 0.1s patch size respectively. The 1s and 0.1s models share all settings except for patch size, patchwise CNN embedding settings, and SSL weights. Some hyperparameters are defined as a function of $d_{\text{model}}$, which we vary across $\{256, 512, 768, 1024, 2048, 3072\}$.

|  | Hyperparameters | Settings |
|---|---|---|
| Input & Masking | Patch size | 500 (1s) |
|  |  | 50 (0.1s) |
|  | Mask ratio | 0.5 |
|  | Masking type | Patch random |
| Patch Encoder (CNN) | Intermediate channel ($C_{\text{inter}}$) | $d_{\text{model}}/8$ (1s) |
|  |  | $d_{\text{model}}/16$ (0.1s) |
|  | Input dimension | $\{1, C_{\text{inter}}, C_{\text{inter}}\}$ |
|  | Output dimension | $\{C_{\text{inter}}, C_{\text{inter}}, C_{\text{inter}}\}$ |
|  | Stride | $\{64, 3, 3\}$ (1s) |
|  |  | $\{4, 3, 3\}$ (0.1s) |
|  | Kernel size | $\{63, 3, 3\}$ |
|  | Padding | $\{31, 1, 1\}$ |
|  | Depth | 3 |
| Patch Encoder (Spectral) | Spectral FFT size | $d_{\text{model}}/2 + 1$ |
|  | Spectral dropout | 0.1 |
| STCPE | STCPE dimension | $d_{\text{model}}/8$ |
|  | STCPE layers | 1 |
|  | STCPE heads | $d_{\text{model}}/256$ |
|  | STCPE $d_{ff}$ | $d_{\text{model}}/2$ |
|  | Time window size | 7 |
| Positional Embedding | Channel type dimension | $d_{\text{model}}/4$ |
|  | Embedding style | CPE (Learnable) |
|  | Temperature | 2000 |
|  | Scale | 1/256 |
| Transformer | Model dimension | $d_{\text{model}}$ |
|  | Layers | 12 |
|  | Heads | $d_{\text{model}}/32$ |
|  | Feed-forward dimension | $4 * d_{\text{model}}$ |
|  | Activation | SiLU |
|  | Attention type | Flash attention |
|  | Dropout | 0.1 |
| SSL Head | Domain | Time, FFT, STFT |
|  | Loss weight ($\lambda_{\text{Time}}$) | 1.0 |
|  | Loss weight ($\lambda_{\text{FFT}}$) | 0.1 (1s) / 1.0 (0.1s) |
|  | Loss weight ($\lambda_{\text{STFT}}$) | 1.0 (1s) / 0.0 (0.1s) |
| Training | Parameterization | $\mu$P |

**Stage1: Grid search** We conducted an extensive grid search across learning rate and weight decay combinations, systematically exploring the hyperparameter space.

- a) Initial Learning Rate Exploration (wd=1e-2): Learning rates: 1e-5, 1e-4, 1e-3, 1e-2

- b) Weight Decay Exploration (lr=1e-3): Weight decay values: 1e-7, 1e-6, 1e-5, 1e-4, 1e-3, 1e-2, 1e-1

- c) Refined Learning Rate Search: Based on initial results, we refined the learning rate search between 1e-4 and 1e-2, testing: 2e-4, 3e-4, 5e-4, 8e-4, 2e-3, 3e-3, 5e-3, 6e-3, 8e-3

- e) Cross-combinations: Additional combinations around promising regions, including lr=6e-3 with various weight decay values: 1e-6, 1e-5, 1e-4, 1e-3, 1e-1

Table 5: **Stage 1: Grid search hyperparameter exploration.**

| Search Step | Hyperparameter Values |
|---|---|
| Initial Learning Rate Exploration | Learning rate: {1e-5, 1e-4, 1e-3, 1e-2} (wd=1e-2) |
| Weight Decay Exploration | Weight decay: {1e-7, 1e-6, 1e-5, 1e-4, 1e-3, 1e-2, 1e-1} (lr=1e-3) |
| Refined Learning Rate Search | Learning rate: {2e-4, 3e-4, 5e-4, 8e-4, 2e-3, 3e-3, 5e-3, 6e-3, 8e-3} |
| Cross-combinations | lr=6e-3 with wd: {1e-6, 1e-5, 1e-4, 1e-3, 1e-1} |

The grid search evaluated 30 distinct learning rate and weight decay combinations, revealing optimal configurations of lr=6.0e-03 with wd=1.0e-06 for $\text{DIVER}_{\text{Small}/\text{I}/1s}$ and lr=1.0e-03 with wd=2.0e-01 for $\text{DIVER}_{\text{Small}/\text{IE}/1s}$ when pretrained on the TUEG dataset (our largest EEG dataset).

For subsequent Optuna optimization of $\text{DIVER}_{\text{Small}/\text{IE}/1s}$, we used the geometric mean between the $\text{DIVER}_{\text{Small}/\text{I}/1s}$ Optuna results (lr=2.30e-03, wd=2.17e-07) and the TUEG-pretrained model's grid search results (lr=1.0e-03, wd=2.0e-01) as the starting point, yielding lr=1.51e-03 and wd=2.09e-04.

**Stage2: Optuna Optimization** We further refined the hyperparameters using Optuna(Akiba et al., 2019) or bayesian hyperparameter optimization. The search space was defined as ±1 order of magnitude around the best grid search configurations (range: ×0.1 to ×10), with 50 trials conducted to systematically explore this refined hyperparameter space. The optimal hyperparameter settings identified through Optuna optimization are presented in Table 6.

Table 6: **Optimal learning rate and weight decay by model configuration.**

| Models | Modality | Granularity | Learning Rate | Weight Decay |
|---|---|---|---|---|
| $\text{DIVER}_{\text{Small}/\text{I}/1s}$ | iEEG | 1s | 2.30e-03 | 2.17e-07 |
| $\text{DIVER}_{\text{Small}/\text{I}/0.1s}$ | iEEG | 0.1s | 4.91e-03 | 3.75e-06 |
| $\text{DIVER}_{\text{Small}/\text{E}/1s}$ | EEG | 1s | 7.70e-03 | 2.14e-07 |
| $\text{DIVER}_{\text{Small}/\text{IE}/1s}$ | iEEG+TUEG | 1s | 2.61e-03 | 1.36e-03 |

### B.7 $\mu$-PARAMETERIZATION ($\mu P$)

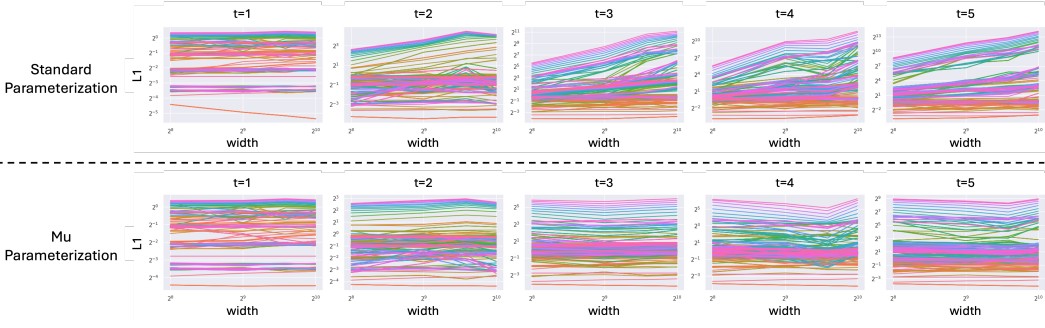

Figure 3: **Verification of the $\mu P$ implementation.** The L1 norm of activation vectors (y-axis) is plotted against model width (x-axis) for five training timesteps ($t$=1 to $t$=5) across four different widths ($256, 512, 768, 1024$). **(Top Row)** With standard parameterization, activation norms are unstable and diverge as model width increases. **(Bottom Row)** In contrast, our $\mu P$ implementation yields stable activation norms that are independent of model width. This confirms the model is correctly parameterized, a critical prerequisite for successful hyperparameter transfer via $\mu$Transfer.

The stable initialization shown in the **Figure 3** translates directly to stable training dynamics. We confirmed this by tracking the training loss while varying the model width across the same four configurations. With $\mu P$ enabled, the training loss remained low and stable for all model sizes. In stark contrast, training without $\mu P$ led to severe instability; the loss diverged rapidly as the model grew, with values exploding to over 100 for the largest width. This empirical result demonstrates that our use of $\mu P$ was essential for reliably training larger models in our scaling experiments.

## B.8 FINETUNING SETUP

For Neuroprobe (iEEG task), DIVER-1 was finetuned at lr = 2e-3 and wd = 1e-2, batch size 32, with AdamW for both frozen and full-finetuning. For the MAYO (iEEG) task, frozen models were fine-tuned using the same learning rate and weight decay as above, whereas full fine-tuning employed separate hyperparameters for each model size (lr = 1.19e-3, wd = 6.18e-1 for 256; lr = 4.19e-3, wd = 1.45e-2 for 512; lr = 7.14e-4, wd = 2.84e-1 for 768; lr = 1.40e-4, wd = 3.44e-2 for 1024; lr = 8.65e-4, wd = 9.90e-2 for 2048; selected via a learning-rate and weight-decay search based on the validation set of subject 1, fold 1).

Unlike Neuroprobe, for which PopT and BrainBERT baselines are provided as benchmarks (Zahorodnii et al., 2025), the MAYO seizure dataset is an extended dataset that we constructed to match Brant's minimum input length, and therefore required training on the baseline models.BrainBERT was set at lr = 1e-3 for the classifier with AdamW, as in the original paper (Wang et al., 2023), with batch size 32. The features in time [l-5:l+5] were concatenated along the channel dimension. Brant was set at lr = 1e-4 for the classifier and 1e-7 for encoder layers, with betas=(0.9, 0.999), eps=1e-8, batch size 4 and Adam, same as their publicly released code. We could not evaluate PopT on MAYO because LPI coordinates are not available. Brant was pretrained with a 6 s patch and a total of 15 patches (90 s), so we were therefore unable to evaluate it on the neuroprobe (1 s), and there is a mismatch with the original pretrained context in MAYO (6 s).

All models were trained for 40 epochs with a CosineAnnealing scheduler. The same validation splits were applied to the training set for each model, and we early-stopped if the validation AUROC did not increase for 10 epochs.

For all EEG downstream tasks, we use the same optimizer (AdamW) and learning rate scheduler (cosine annealing) as described in the iEEG finetuning configuration. The base learning rate is set to 2.00e-4, weight decay to 3.00e-1, and batch size to 64. We perform full finetuning without employing multi-lr strategies, applying the same learning rate to both the backbone and classifier. The classifier consists of a 3-layer MLP with ELU activation functions, where the first hidden layer has width $T \times 200$ ($T$ is the sample duration in seconds), the second hidden layer has width 200, and the output layer dimension matches the number of classes. Dropout rate is set to 0.1, label smoothing to 0.1, and gradient clipping value to 1.0. All models are trained for 50 epochs without early stopping. For model selection, we use the epoch that achieves the best validation performance (AUROC for binary classification, F1 score for multi-class classification), which is then evaluated on the test set. All results are reported as mean ± standard deviation across 5 random seeds (41, 42, 43, 44, 45). For the one-to-one comparison experiments with CBraMod (Table 24), we use task-specific hyperparameters to match CBraMod's training conditions: on Mumtaz2016(Mumtaz, 2016), linear probing uses learning rate 5.00e-6 and weight decay 6.25e-6, while multi-lr and linear classifier configurations use learning rate 6.25e-5 and weight decay 6.25e-6.

## C SCALING LAW

### C.1 SCALING LAW EXPERIMENT DETAILS

Pretraining loss curves for the trained models are presented in Figure 4 for the DIVER$_{-/I/1s}$ model family, Figure 5 for the DIVER$_{-/I/0.1s}$ model family and Figure 6 for the DIVER$_{-/E/1s}$ model family. For the iEEG models, we trained separate instances for each epoch to obtain the corresponding loss curves. In contrast, for EEG models, we trained all models with a fixed maximum of 32 epochs and extracted test loss values at the relevant epoch checkpoints for scaling analysis. This difference in training procedure was necessitated by computational constraints, as EEG model training requires substantially longer wall-clock time. Consequently, the iEEG plots display epoch-specific loss curves(4 and 5), while the EEG plot presents all scaling curves within a single figure(Figure 6). Importantly, this methodological difference affects the learning rate schedule: we employed cosine annealing with warmup, where the decay schedule depends on the total number of training epochs. Therefore, extracting the loss at epoch 2 from a model trained for 32 epochs differs from the loss at epoch 2 of a model trained for only 2 epochs, as the learning rate trajectories diverge under these configurations.

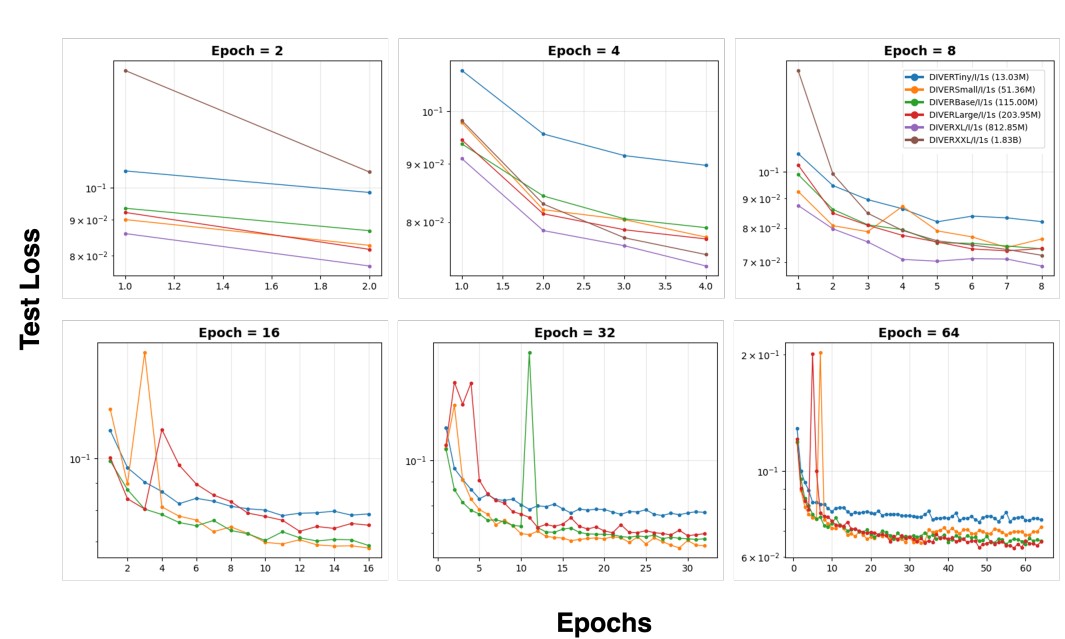

Figure 4: Loss curves of the DIVER$_{-/I/1s}$ model family. Test loss across epochs is shown.

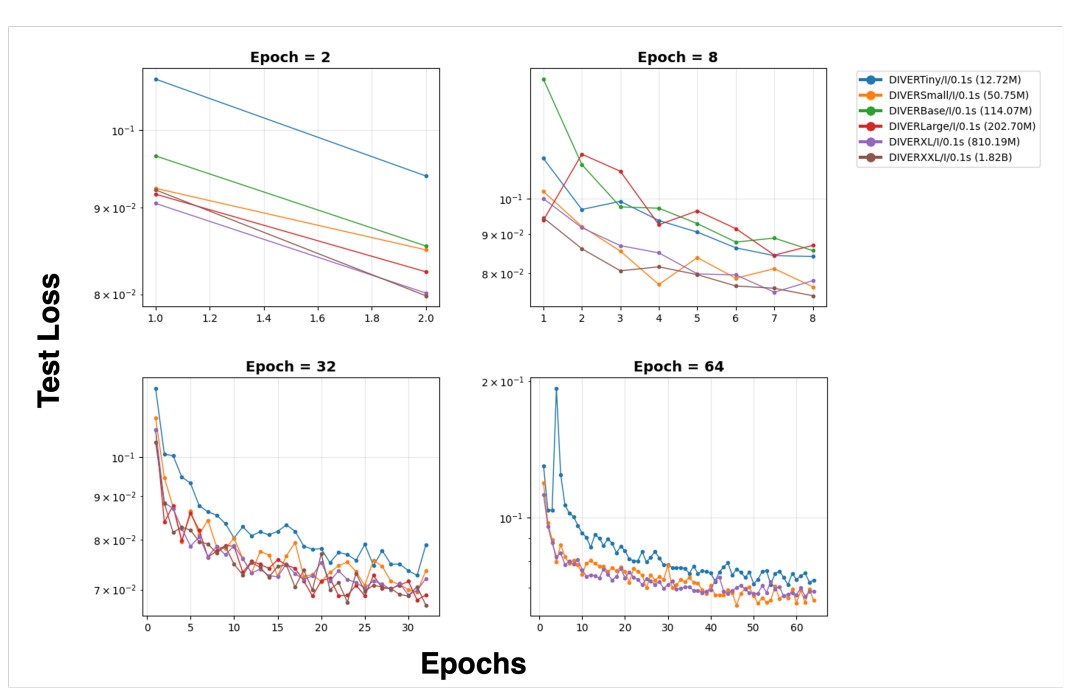

Figure 5: Loss curves of the DIVER$_{-/I/0.1s}$ model family. Test loss across epochs is shown.

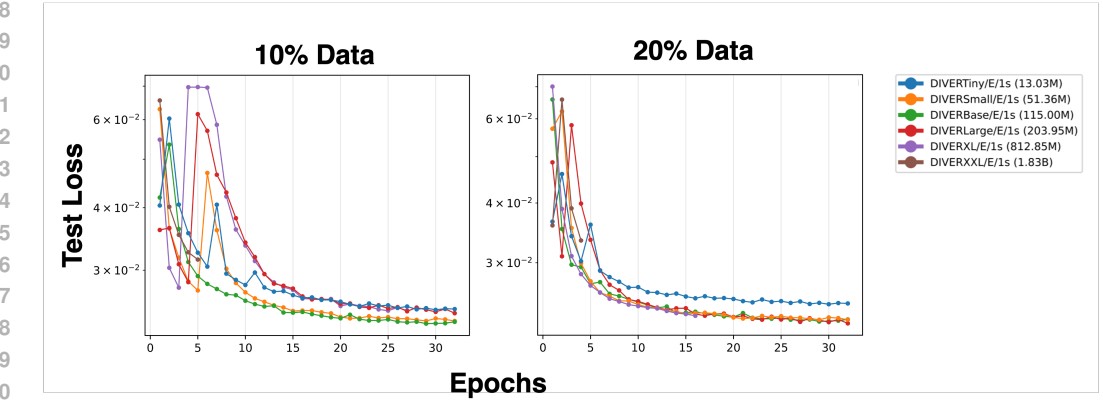

Figure 6: Loss curves of the DIVER$_{-/E/1s}$ model family for each dataset size. Test loss across epochs is shown. Unlike the iEEG experiments, where separate models were trained for each epoch, all EEG models were trained for a fixed 32-epoch schedule. Early-epoch losses were extracted from intermediate checkpoints of these longer runs. Consequently, each dataset size yields five loss curves, one for each model scale trained within the 32-epoch run.

**Unique number of tokens** First of all, the estimated total number of unique tokens is $U_D =$ number of data sample × number of tokens per sample. The number of tokens per sample can be expressed as number of channels × number of timestamps × proportion of unmasked patches.

For iEEG, the total number of data samples is 636,480. Since we randomly sampled from a 32 channel x 30 timestamp token grid using Beta(3,1) distribution for both axis, we estimate the number of tokens per sample as $32 \times 30 \times 0.75 \times 0.75 = 540$ tokens. Thus, we get $U_D = 636,480 \times 540$ for iEEG.

In the case of EEG, the approximated total number of data samples is 1,960,800, and the number of tokens per sample is same as iEEG. Therefore, the $U_D = 1,960,800 \times 540$ for EEG.

**Compute** To investigate compute scaling properties, we conducted systematic pretraining experiments across different epoch counts for the 1-second granularity models. For shorter training regimes (1, 2, 4, and 8 epochs), we pretrained all six model variants: DIVER$_{Tiny/I/1s}$, DIVER$_{Small/I/1s}$, DIVER$_{Base/I/1s}$, DIVER$_{Large/I/1s}$, DIVER$_{XL/I/1s}$, and DIVER$_{XXL/I/1s}$. For longer training regimes (16, 32, and 64 epochs), computational constraints limited our experiments to the four smaller model variants: DIVER$_{Tiny/I/1s}$, DIVER$_{Small/I/1s}$, DIVER$_{Base/I/1s}$, and DIVER$_{Large/I/1s}$.

For the DIVER$_{-/I/0.1s}$ model family with 0.1-second granularity models, we followed a similar training protocol. We trained six model variant for epochs 2, 8 and five model variants for epochs 32, excluding DIVER$_{Base/I/0.1s}$ and three model variants for epochs 64, excluding DIVER$_{Base/I/0.1s}$, DIVER$_{Large/I/0.1s}$ and DIVER$_{XXL/I/0.1s}$.

For the DIVER$_{-/E/1s}$ model family on EEG data, we adopted a different training approach due to the substantially longer training time required for EEG models. We trained all model variants with a fixed maximum of 32 epochs and extracted test loss values at epochs 2, 4, 8, 16, and 32 for scaling analysis. We conducted experiments on 10% and 20% of the EEG dataset. For the 10% subset, we trained five model variants up to 32 epochs, while for the 20% subset, computational constraints allowed us to train four variants: DIVER$_{Tiny/E/1s}$, DIVER$_{Small/E/1s}$, DIVER$_{Base/E/1s}$, DIVER$_{Large/E/1s}$. DIVER$_{XL/E/1s}$ was trained up to 16 epochs in 20% data and DIVER$_{XXL/E/1s}$ was trained up to 4 epochs in both 10% and 20% dataset setting.

**Data Size** For iEEG, data size scaling was done in 1, 3, 9, 24, 50, 90, 100% of data on DIVER$_{Small/I/1s}$ for 2 epochs with hyperparameters fixed as $lr = 6.0e - 03, wd = 1.0e - 06$. For EEG, data size scaling was done in 1, 3, 9, 10, 24, 50, 100% of data on DIVER$_{Small/E/1s}$ for 2 epochs with hyperparameters fixed as $lr = 7.70 \times 10^{-3}, wd = 2.14 \times 10^{-7}$. Number of token =

number of data sample $\times$ number of tokens per sample, while the total number of data samples was 636,480 for iEEG and 1,960,800 for EEG. Number of tokens per sample estimated as 540, given that we randomly sampled from a 32×30 token grid using Beta(3,1) distribution. A detailed explanation can be found in the aforementioned **Unique number of tokens** section.

**Model Size** We fixed the number of epochs to 2. The models varied by their width, number of layers, and patch size. The detailed experiment conditions are on table 1. We observed that the models with different number of layers or patch size show different scaling behavior, so we fitted them separately.

**Number of Subjects** Subject scaling experiments were done only for iEEG, with datasets containing 2, 4, 5, 8, 10, 15, 16 subjects respectively, while maintaining a constant dataset size.DIVER$_{\text{Small/I/0.1}s}$ trained for 2 epochs, due to compute constraint.

**Data-constrained Scaling Law** We trained a total of 31 models with varying parameter counts and numbers of training epochs, while keeping the dataset fixed. The hidden dimension was fixed to 12 layers. Since different granularity led to different scaling behavior, we experimented on two granularity conditions and fitted them separately. The empirical isoLoss contours in Figure 2(j) show less smoothness compared to the original scaling law paper(Muennighoff et al., 2023), primarily due to sampling density. While the original study used 93 model configurations with dense sampling across all loss ranges, we evaluated 30 configurations. Despite this visual difference, our empirical isoLoss contours (Figure 2(j) and Figure 7, left panel) align well with the predicted contours (Figure 2(j) and Figure 7, right panel), demonstrating that data-constrained scaling laws generalize to neural data.

### C.2 Extended Kaplan (Kaplan et al., 2020) Scaling Law Results for iEEG

We tested models at the 1-second and 0.1-second granularity. For 1-second granularity models, all six models listed in Table 1 as DIVER$_{\text{-/I/1}s}$ were tested for epochs 2, 4, 8. At epochs 16, 32, and 64, evaluation was conducted on the following four models: DIVER$_{\text{Tiny/I/1}s}$, DIVER$_{\text{Small/I/1}s}$, DIVER$_{\text{Base/I/1}s}$, and DIVER$_{\text{Large/I/1}s}$. At earlier epochs (2, 4, and 8), the general trend showed decreasing loss as model size increased. However, as observed in DIVER$_{\text{XXL/I/1}s}$, larger models exhibited substantially higher loss when trained with only a few epochs. This confirms what was also suggested by the data-constrained scaling isoplots: training very large models with insufficient updates is ineffective. Another possibility is that the aspect ratio of DIVER$_{\text{XXL/I/1}s}$ (256) places it outside the region where loss remains stable. Future work should therefore evaluate larger models within the aspect-ratio regime where stable loss behavior is maintained.

### C.3 Data-Constrained Scaling Law Fitting Results for iEEG

Table 7: **Fitted data-constrained scaling law parameters for DIVER$_{\text{-/I/1}s}$ and DIVER$_{\text{-/I/0.1}s}$ model families.**

|  | DIVER$_{\text{-/I/1}s}$ | DIVER$_{\text{-/I/0.1}s}$ |
|---|---|---|
| $A$ | 19.217 | 101.52 |
| $B$ | 57.065 | 1.1550 |
| $E$ | 0.0092 | 0.0030 |
| $\alpha$ | 0.3773 | 0.5248 |
| $\beta$ | 0.3504 | 0.1246 |
| $R_D^*$ | 9.5372 | 19.705 |
| $R_N^*$ | 3.3850 | 0.7191 |
| $R^2$ (linear) | 0.7858 | 0.7575 |
| $R^2$ (log) | 0.8152 | 0.7718 |

Table C.3 shows fitted data-constrained scaling law parameters. $A$ and $B$ describe the relative influence of parameters and dataset size on loss. In our setting, we obtain larger $A$ than $B$ in both granularity, suggesting that model size plays a more critical role than dataset size. In particular, the 0.1s patch model yields a comparatively small value of $B$ (0.3925), suggesting that variations in dataset size exert only a minor effect on the loss in this setting.

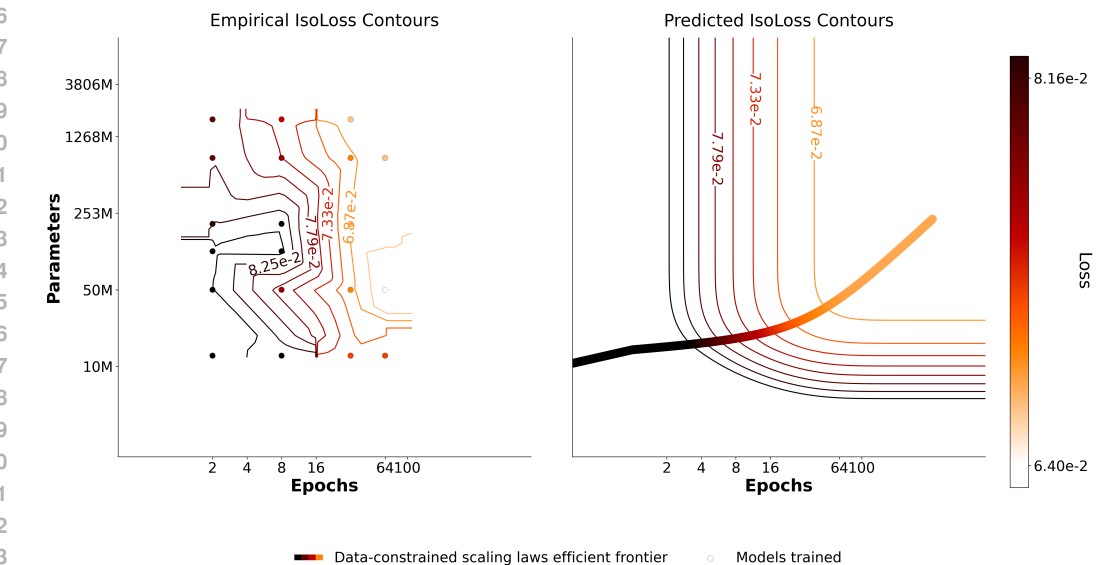

Figure 7: **IsoLoss contours for DIVER.**$_{\cdot/\mathbf{I}/0.1s}$ **model family:** (Left) Twenty models with 0.1s patches were trained across varying epochs and parameter counts. Iso-loss contours are obtained by linear interpolation between measured data points. (Right) Corresponding contours predicted by the fitted scaling law. The fading line denotes the minimum-loss configuration for each compute budget.

The exponents $\alpha$ and $\beta$ govern the marginal benefit of scaling parameters and data, respectively. Our values ($\alpha = 0.377$, $\beta = 0.350$) indicate that increasing model size and adding data yield comparable contributions to overall performance improvements. Compared to prior results in language domain ($\alpha = 0.348$, $\beta = 0.366$; (Hoffmann et al., 2022)), our fitted exponents show similar values.

Importantly, the characteristic half-lives $R_D^* = 8.9$ and $R_N^* = 3.3$ quantify diminishing returns under repeated data and excessive parameters. The relatively larger $R_D^*$ implies that repeated data remains useful for many epochs before saturation, whereas the smaller $R_N^*$ suggests that the benefit of adding parameters decays more quickly. Together, these results suggest that gains are most effectively pursued by scaling model size while maintaining moderate dataset repetition, rather than prioritizing further data collection.

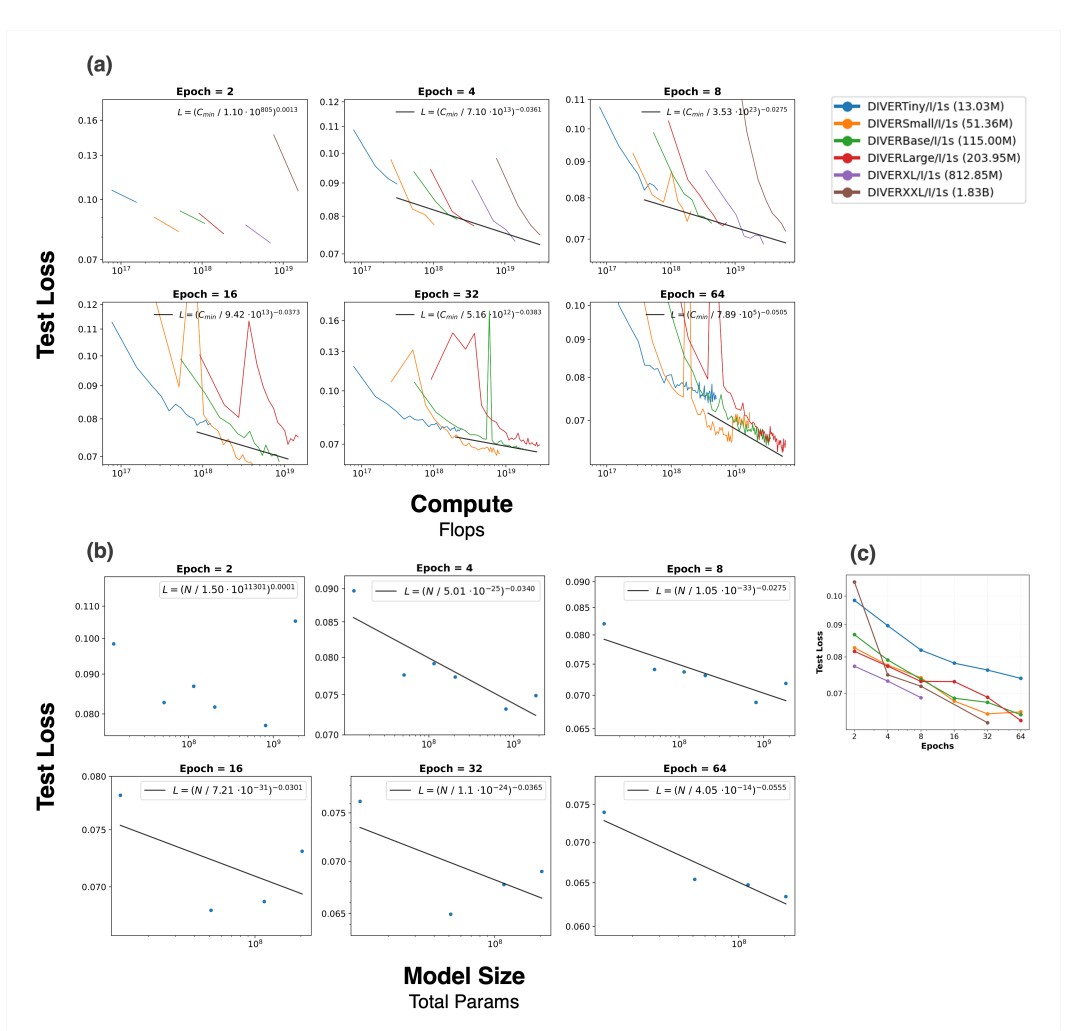

Figure 8: Scaling law extended results for the DIVER$_{-/I/1s}$ family. Compute scaling and model size scaling plots are given for models trained for 2, 4, 8, 16, 32, and 64 epochs.(a) Compute scaling and (b) model size scaling plots are given for models trained for 2, 8, 32, and 64 epochs. (c) Epoch scaling plot of the models reported in Fig. 2 (d). The same training runs are reused, with losses re-plotted against parameters and dataset size in log-log scale.

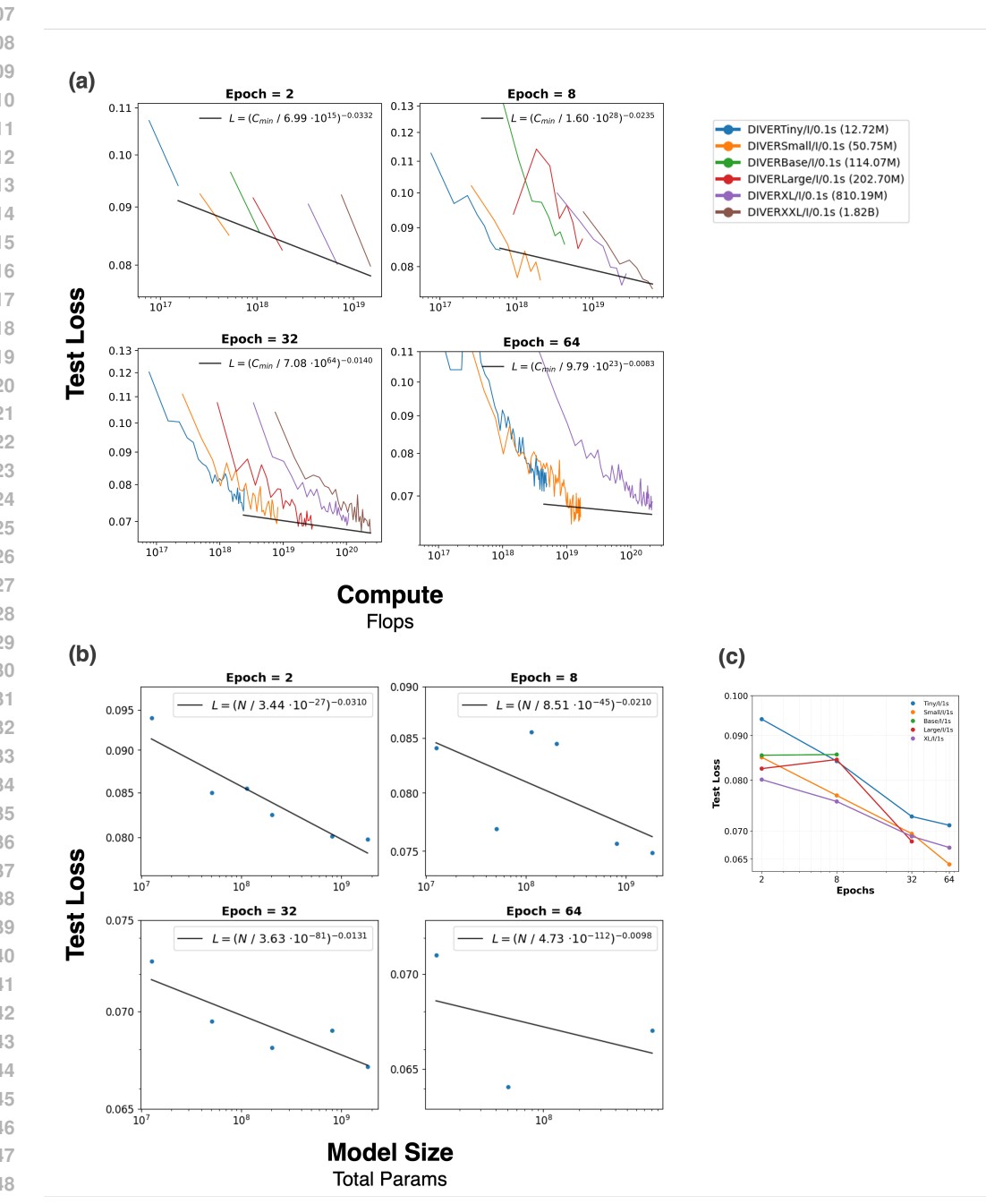

Figure 9: Scaling law extended results for the DIVER$_{-/I/0.1s}$ family. (a) Compute scaling and (b) model size scaling plots are given for models trained for 2, 8, 32, and 64 epochs. (c) Epoch scaling plot of the models reported in Fig. 7. The same training runs are reused, with losses re-plotted against parameters and dataset size in log-log scale.

## C.4 EXTENDED KAPLAN (KAPLAN ET AL., 2020) SCALING LAW RESULTS FOR EEG

We tested $\text{DIVER}_{-/E/1s}$ model family trained on 10% and 20% of the EEG dataset. Use of partial dataset was due to computational constraints. Overall, the scaling behavior across compute and model size demonstrates reasonable fit to the power law. However, notable deviations were observed for the 10% data subset at epochs 16 and 32, where the power law fit was inadequate. This is primarily attributable to instabilities in $\text{DIVER}_{XL/E/1s}$, whose loss curve exhibits sudden upward spikes and irregular behavior(Figure 6). Furthermore, when trained for shorter durations, larger models exhibit overfitting tendencies; consequently, fitting the power law across all model sizes fails to capture the expected scaling behavior in these regimes. For cases where the fitted slope approached zero, we omit the fitted line from the visualization. Additionally, it is important to note that the suboptimal learning rate schedule—arising from our use of a single training instance with fixed maximum epoch as 32 for EEG models—may contribute to these deviations from ideal scaling behavior.

## C.5 DATA-CONSTRAINED SCALING LAW FITTING RESULTS FOR EEG

We included models trained on 10% and 20% of the full dataset, as in the Kaplan scaling law experiment setup in C.4.

Table 8: **Fitted data-constrained scaling law parameters for DIVER$_{-/E/1s}$ model family.**

|  | DIVER$_{-/E/1s}$ |
| --- | --- |
| $A$ | 0.5983 |
| $B$ | 2.0633 |
| $E$ | 0.0004 |
| $\alpha$ | 3.4480 |
| $\beta$ | 0.2059 |
| $R_D^*$ | 23.860 |
| $R_N^*$ | 3.2903 |
| $R^2$ (linear) | 0.5019 |
| $R^2$ (log) | 0.6012 |

The most notable difference in the EEG scaling results appears in the model-size exponent. For EEG, we obtain an unusually large $\alpha = 3.29$, whereas the iEEG experiments produced a much smaller and value of $\alpha = 0.3773$. Such a steep exponent suggests that, within the range of model sizes we explored, performance changed very little with additional capacity. A plausible explanation is the behavior of the XXL model: its loss is higher than that of the smaller models, probably due to overfitting. Similar behavior has been reported in very large models trained on fixed data in the Data-constrained scaling law literature(Muennighoff et al., 2023). In addition, we observed that losses often rise around epochs 4–8; because our analysis used the midpoint checkpoint of the 32-epoch runs, this choice may also have distorted the fit.

This instability is also reflected in the overall goodness-of-fit. The EEG scaling law yields an $R^2$ of 0.5019, substantially lower than the 0.8152 observed for iEEG, indicating that EEG does not follow the expected power-law pattern nearly as well. Some of this gap may arise from fundamental differences between EEG and iEEG: EEG has lower signal quality and greater trial-to-trial variability, which can obscure systematic trends. At the same time, aspects of our experimental design may also have contributed. In particular, training all models for a fixed 32 epochs and sampling intermediate losses, rather than evaluating models at comparable levels of convergence, may have introduced additional noise into the fit.

This interpretation is reinforced by the estimate of the irreducible-loss term $E$, which drops to an unusually small value ($E = 0.0004$) for EEG, far below the iEEG estimate of $E = 0.0092$. Such a low value is difficult to justify on theoretical grounds and likely reflects a compensatory effect of the fitting procedure rather than a meaningful property of the data. Even so, the EEG models follow the general direction of the expected scaling behavior, albeit in a much noisier and less stable form than iEEG.

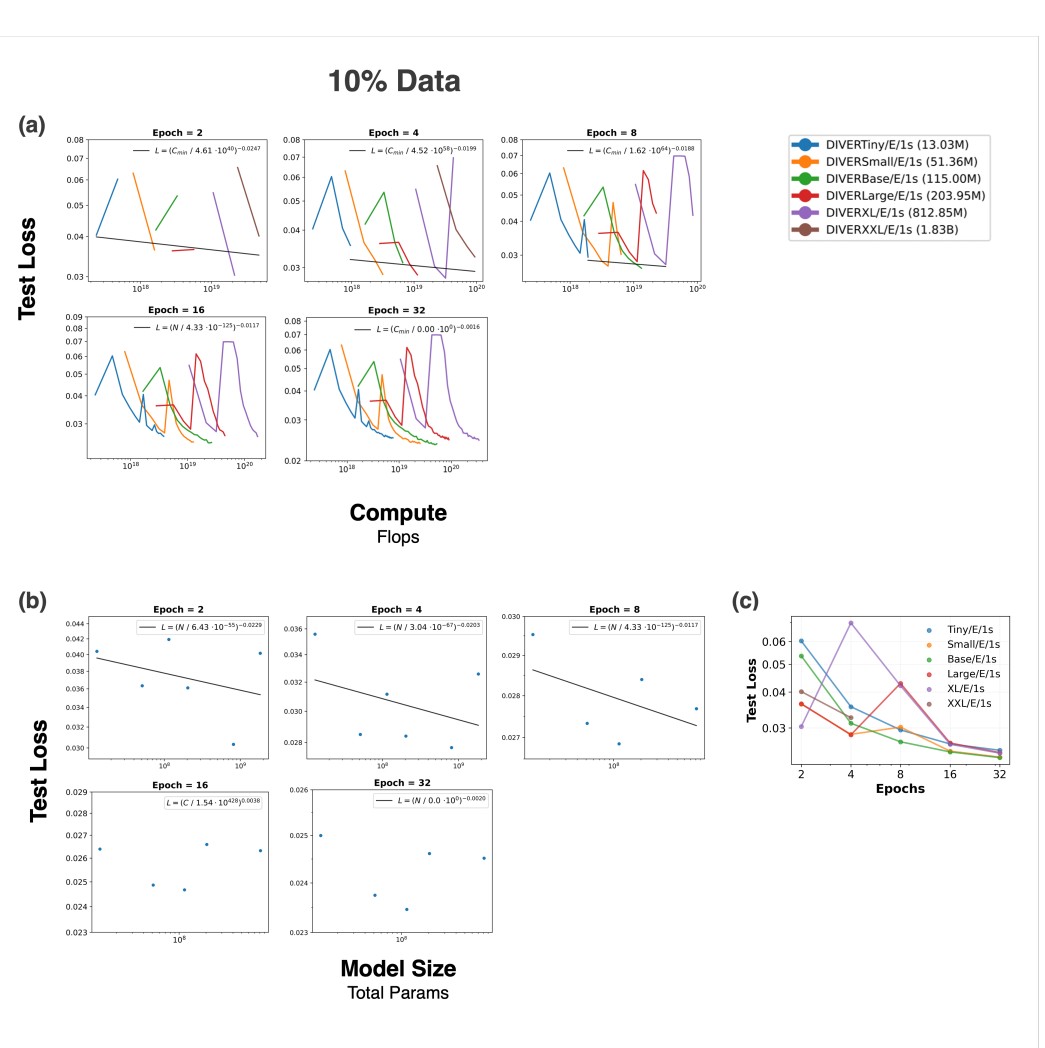

Figure 10: Scaling law extended results for the DIVER$_{-/E/1s}$ family on 10% of EEG dataset. (a) Compute scaling and (b) model size scaling plots are given for test loss values extracted at epochs 2, 4, 8, 16, and 32 from models trained with a maximum of 32 epochs. For epochs 16 and 32, the fitted slope was effectively zero, so the corresponding fitted lines were omitted from the visualization. (c) Epoch scaling plot of the models in log-log scale.

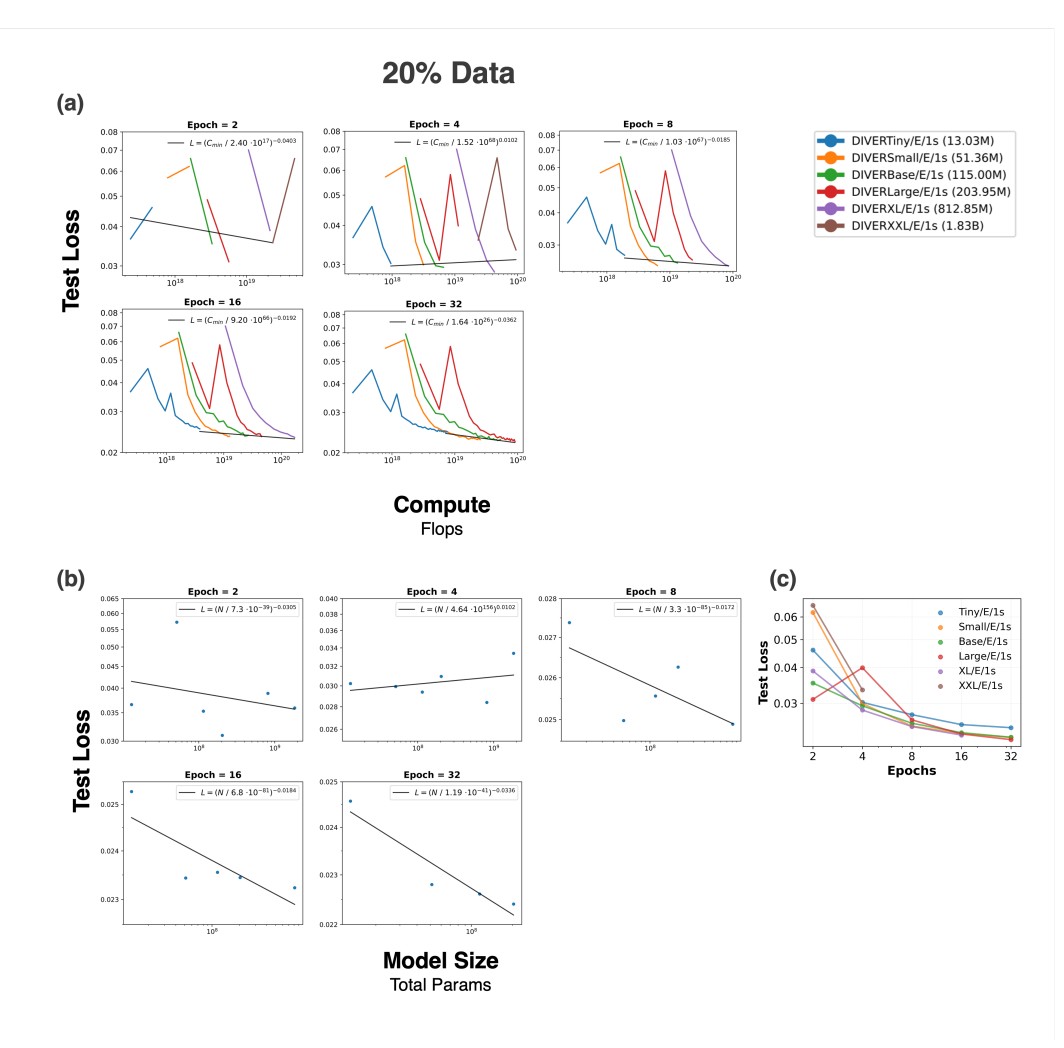

Figure 11: Scaling law extended results for the DIVER$_{-/E/1s}$ family on 20% of EEG dataset. (a) Compute scaling and (b) model size scaling plots are given for test loss values extracted at epochs 2, 4, 8, 16, and 32 from models trained with a maximum of 32 epochs. (c) Epoch scaling plot of the models reported in reported in Fig. 2(h). The same training runs are reused, with losses re-plotted against parameters and dataset size in log-log scale.

## C.6 EXTENDED EEG DOWNSTREAM SCALING RESULTS

**Data size scaling (Table 9).** We evaluate how pretraining data size affects downstream performance by training $\text{DIVER}_{\text{Small}/E/1s}$ with $d_{\text{model}} = 512$ for 2 epochs on varying fractions of the EEG pretraining dataset (1%, 3%, 9%, 24%, 50%, and 100%). Performance generally improves as more pretraining data is used for FACED dataset. While the performance peaked at 24% of the pretraining dataset for PhysioNet-MI, this difference is not significant when considering the standard deviation; performance at 100% is comparable. Overall, larger pretraining datasets lead to better downstream task performance. The improvements are most pronounced when scaling from small fractions (1-10%) to larger portions of the dataset, with diminishing but still positive returns at the largest scales.

**Model size scaling (Table 10).** We examine the effect of model capacity by evaluating the $\text{DIVER}_{-/E/1s}$ model family with varying hidden dimensions: $d_{\text{model}} \in \{256, 512, 768, 1024, 2048\}$ (corresponding to 12M, 48M, 114M, 204M, and 813M parameters respectively). All models were pretrained for 2 epochs on the full EEG dataset. Performance generally improves with model size, though not always monotonically. The $\text{DIVER}_{XL/E/1s}(d_{model} = 2048)$ achieves the best performance for PhysioNet-MI dataset while $\text{DIVER}_{Small/E/1s}$ ($d_{model} = 512$) achieved best performance for FACED dataset. This could be possible due to the limited number of epochs used to pretrain the model, but further work is needed to clarify this.

**Epoch scaling (Table 11).** We investigate how pretraining epochs affects downstream performance by training models across different sizes ($d_{\text{model}} \in \{256, 512, 768, 1024, 2048\}$) for varying numbers of epochs (2, 4, 8, 16, 32). All models were pretrained on 10% of the EEG dataset due to computational constraints. Performance generally improves with more training epochs, though the optimal number of epochs varies by model size. Larger models tend to benefit more from extended training, while smaller models may plateau earlier. These results demonstrate that both model size and training duration are important factors in achieving optimal downstream performance.

Together, these scaling experiments demonstrate that EEG foundation model performance improves predictably across multiple dimensions—data size, model size, and training duration—providing practical guidance for efficient model development.

Table 9: **Data size scaling on EEG downstream tasks.** $\text{DIVER}_{Small/E/1s}$ with $d_{\text{model}} = 512$ was trained for 2 epochs on a different fraction (1%, 3%, 9%, ..., 100%) of the EEG pretraining dataset.

| Pretraining Fraction | FACED | | | PhysioNet-MI | | |
|---|---|---|---|---|---|---|
| | ACC | kappa | F1 | ACC | kappa | F1 |
| 1% | $0.111 \pm 0.000$ | $0.000 \pm 0.000$ | $0.036 \pm 0.000$ | $0.581 \pm 0.007$ | $0.441 \pm 0.009$ | $0.582 \pm 0.006$ |
| 3% | $0.467 \pm 0.012$ | $0.400 \pm 0.013$ | $0.474 \pm 0.012$ | $0.624 \pm 0.005$ | $0.498 \pm 0.007$ | $0.626 \pm 0.005$ |
| 9% | $0.451 \pm 0.017$ | $0.380 \pm 0.020$ | $0.456 \pm 0.017$ | $0.600 \pm 0.005$ | $0.467 \pm 0.007$ | $0.601 \pm 0.005$ |
| 10% | $0.509 \pm 0.013$ | $0.448 \pm 0.014$ | $0.520 \pm 0.012$ | $0.640 \pm 0.006$ | $0.519 \pm 0.009$ | $0.641 \pm 0.006$ |
| 24% | $0.525 \pm 0.007$ | $0.464 \pm 0.008$ | $0.529 \pm 0.006$ | $\mathbf{0.650 \pm 0.004}$ | $\mathbf{0.533 \pm 0.005}$ | $\mathbf{0.651 \pm 0.004}$ |
| 50% | $0.493 \pm 0.007$ | $0.426 \pm 0.007$ | $0.491 \pm 0.006$ | $0.628 \pm 0.006$ | $0.504 \pm 0.008$ | $0.628 \pm 0.006$ |
| 100% | $\mathbf{0.568 \pm 0.011}$ | $\mathbf{0.513 \pm 0.013}$ | $\mathbf{0.579 \pm 0.009}$ | $0.647 \pm 0.006$ | $0.529 \pm 0.008$ | $0.649 \pm 0.006$ |

| Pretraining Fraction | MentalArithmetic | | |
|---|---|---|---|
| | ACC | AUC-PR | AUROC |
| 1% | $0.500 \pm 0.000$ | $0.625 \pm 0.000$ | $0.500 \pm 0.000$ |
| 3% | $0.597 \pm 0.090$ | $0.350 \pm 0.053$ | $0.611 \pm 0.069$ |
| 9% | $0.626 \pm 0.058$ | $0.520 \pm 0.087$ | $0.690 \pm 0.065$ |
| 10% | $\mathbf{0.635 \pm 0.041}$ | $0.547 \pm 0.045$ | $\mathbf{0.765 \pm 0.020}$ |
| 24% | $0.617 \pm 0.067$ | $0.370 \pm 0.054$ | $0.701 \pm 0.033$ |
| 50% | $0.572 \pm 0.088$ | $\mathbf{0.563 \pm 0.234}$ | $0.752 \pm 0.115$ |
| 100% | $0.619 \pm 0.046$ | $0.490 \pm 0.069$ | $0.765 \pm 0.049$ |

Table 10: **Model size scaling on EEG downstream tasks.** $\text{DIVER}_{-/E/1s}$ model family across different model sizes were evaluated. All models were pretrained for 2 epochs using the EEG dataset.

| $d_{\text{model}}$ (Params) | FACED | | | PhysioNet-MI | | |
|---|---|---|---|---|---|---|
| | ACC | kappa | F1 | ACC | kappa | F1 |
| 256 (13M) | $0.488 \pm 0.006$ | $0.421 \pm 0.006$ | $0.485 \pm 0.005$ | $0.630 \pm 0.006$ | $0.506 \pm 0.008$ | $0.631 \pm 0.007$ |
| 512 (51M) | $\mathbf{0.568 \pm 0.011}$ | $\mathbf{0.513 \pm 0.013}$ | $\mathbf{0.579 \pm 0.009}$ | $0.647 \pm 0.006$ | $0.529 \pm 0.008$ | $0.649 \pm 0.006$ |
| 768 (114M) | $0.477 \pm 0.006$ | $0.411 \pm 0.006$ | $0.477 \pm 0.006$ | $0.659 \pm 0.005$ | $0.546 \pm 0.007$ | $0.660 \pm 0.005$ |
| 1024 (204M) | $0.524 \pm 0.005$ | $0.462 \pm 0.005$ | $0.523 \pm 0.005$ | $\mathbf{0.691 \pm 0.006}$ | $\mathbf{0.588 \pm 0.008}$ | $\mathbf{0.692 \pm 0.006}$ |
| 2048 (813M) | $0.525 \pm 0.012$ | $0.462 \pm 0.012$ | $0.524 \pm 0.010$ | $0.669 \pm 0.011$ | $0.560 \pm 0.014$ | $0.672 \pm 0.010$ |

| $d_{\text{model}}$ (Params) | MentalArithmetic | | |
|---|---|---|---|
| | ACC | AUC-PR | AUROC |
| 256 (13M) | $0.626 \pm 0.032$ | $0.480 \pm 0.072$ | $0.751 \pm 0.036$ |
| 512 (51M) | $0.619 \pm 0.046$ | $0.490 \pm 0.069$ | $0.765 \pm 0.049$ |
| 768 (114M) | $0.697 \pm 0.012$ | $0.631 \pm 0.045$ | $0.800 \pm 0.029$ |
| 1024 (204M) | $\mathbf{0.722 \pm 0.028}$ | $\mathbf{0.657 \pm 0.063}$ | $\mathbf{0.806 \pm 0.033}$ |
| 2048 (813M) | $0.674 \pm 0.031$ | $0.601 \pm 0.053$ | $0.780 \pm 0.031$ |

Table 11: **Epoch scaling on EEG downstream tasks.** We evaluate the $\text{DIVER}_{-/E/1s}$ model family across different model sizes, each pretrained for varying numbers of epochs. All models were pre-trained using 10% of the EEG dataset.

| $d_{\text{model}}$ | Epochs | FACED | | | PhysioNet-MI | | |
|---|---|---|---|---|---|---|---|
| | | ACC | kappa | F1 | ACC | kappa | F1 |
| 256 | 2 | $0.153 \pm 0.007$ | $0.050 \pm 0.007$ | $0.103 \pm 0.015$ | $0.580 \pm 0.006$ | $0.440 \pm 0.007$ | $0.582 \pm 0.006$ |
| | 4 | $0.463 \pm 0.013$ | $0.392 \pm 0.015$ | $0.461 \pm 0.014$ | $0.604 \pm 0.008$ | $0.472 \pm 0.011$ | $0.605 \pm 0.008$ |
| | 8 | $0.484 \pm 0.005$ | $0.417 \pm 0.006$ | $0.484 \pm 0.006$ | $0.614 \pm 0.003$ | $0.485 \pm 0.004$ | $0.614 \pm 0.003$ |
| | 16 | $0.447 \pm 0.003$ | $0.375 \pm 0.003$ | $0.443 \pm 0.002$ | $0.609 \pm 0.004$ | $0.479 \pm 0.005$ | $0.610 \pm 0.004$ |
| | 32 | $0.488 \pm 0.006$ | $0.421 \pm 0.006$ | $0.485 \pm 0.005$ | $0.615 \pm 0.003$ | $0.487 \pm 0.005$ | $0.615 \pm 0.004$ |
| 512 | 2 | $0.510 \pm 0.012$ | $0.448 \pm 0.013$ | $0.517 \pm 0.012$ | $0.624 \pm 0.005$ | $0.498 \pm 0.007$ | $0.625 \pm 0.005$ |
| | 4 | $0.539 \pm 0.012$ | $0.479 \pm 0.013$ | $0.543 \pm 0.010$ | $0.654 \pm 0.005$ | $0.538 \pm 0.007$ | $0.655 \pm 0.005$ |
| | 8 | $0.445 \pm 0.008$ | $0.373 \pm 0.008$ | $0.444 \pm 0.008$ | $0.619 \pm 0.004$ | $0.491 \pm 0.005$ | $0.619 \pm 0.004$ |
| | 16 | $0.476 \pm 0.004$ | $0.410 \pm 0.004$ | $0.477 \pm 0.004$ | $0.668 \pm 0.002$ | $0.557 \pm 0.003$ | $0.669 \pm 0.002$ |
| | 32 | $0.493 \pm 0.005$ | $0.429 \pm 0.005$ | $0.495 \pm 0.005$ | $0.680 \pm 0.004$ | $0.573 \pm 0.006$ | $0.681 \pm 0.004$ |
| 768 | 2 | $0.387 \pm 0.024$ | $0.309 \pm 0.027$ | $0.386 \pm 0.025$ | $0.605 \pm 0.004$ | $0.473 \pm 0.005$ | $0.606 \pm 0.003$ |
| | 4 | $0.511 \pm 0.009$ | $0.448 \pm 0.010$ | $0.516 \pm 0.010$ | $0.636 \pm 0.005$ | $0.515 \pm 0.006$ | $0.638 \pm 0.005$ |
| | 8 | $0.531 \pm 0.004$ | $0.470 \pm 0.005$ | $0.534 \pm 0.005$ | $0.663 \pm 0.004$ | $0.550 \pm 0.006$ | $0.664 \pm 0.004$ |
| | 16 | $0.519 \pm 0.011$ | $0.458 \pm 0.012$ | $0.518 \pm 0.012$ | $0.682 \pm 0.004$ | $0.576 \pm 0.005$ | $0.683 \pm 0.004$ |
| | 32 | $0.526 \pm 0.005$ | $0.467 \pm 0.006$ | $0.527 \pm 0.005$ | $\mathbf{0.690 \pm 0.002}$ | $\mathbf{0.587 \pm 0.002}$ | $\mathbf{0.691 \pm 0.002}$ |
| 1024 | 2 | $0.510 \pm 0.010$ | $0.449 \pm 0.010$ | $0.518 \pm 0.008$ | $0.638 \pm 0.005$ | $0.518 \pm 0.007$ | $0.640 \pm 0.005$ |
| | 4 | $\mathbf{0.544 \pm 0.011}$ | $\mathbf{0.486 \pm 0.013}$ | $\mathbf{0.554 \pm 0.010}$ | $0.656 \pm 0.004$ | $0.542 \pm 0.005$ | $0.658 \pm 0.004$ |
| | 8 | $0.462 \pm 0.012$ | $0.393 \pm 0.013$ | $0.464 \pm 0.012$ | $0.591 \pm 0.006$ | $0.454 \pm 0.008$ | $0.592 \pm 0.007$ |
| | 16 | $0.461 \pm 0.007$ | $0.390 \pm 0.008$ | $0.458 \pm 0.007$ | $0.630 \pm 0.008$ | $0.507 \pm 0.011$ | $0.632 \pm 0.008$ |
| | 32 | $0.451 \pm 0.030$ | $0.380 \pm 0.033$ | $0.449 \pm 0.030$ | $0.644 \pm 0.016$ | $0.526 \pm 0.022$ | $0.646 \pm 0.016$ |
| 2048 | 2 | $0.380 \pm 0.220$ | $0.301 \pm 0.246$ | $0.355 \pm 0.261$ | $0.620 \pm 0.016$ | $0.495 \pm 0.014$ | $0.624 \pm 0.009$ |
| | 4 | $0.111 \pm 0.000$ | $0.000 \pm 0.000$ | $0.036 \pm 0.000$ | $0.512 \pm 0.131$ | $0.351 \pm 0.176$ | $0.485 \pm 0.193$ |
| | 8 | $0.450 \pm 0.018$ | $0.379 \pm 0.021$ | $0.450 \pm 0.019$ | $0.620 \pm 0.006$ | $0.495 \pm 0.007$ | $0.623 \pm 0.005$ |
| | 16 | $0.514 \pm 0.008$ | $0.449 \pm 0.008$ | $0.512 \pm 0.007$ | $0.658 \pm 0.004$ | $0.547 \pm 0.005$ | $0.662 \pm 0.004$ |
| | 32 | $0.531 \pm 0.014$ | $0.470 \pm 0.015$ | $0.514 \pm 0.013$ | $0.661 \pm 0.009$ | $0.550 \pm 0.012$ | $0.665 \pm 0.009$ |

| $d_{\text{model}}$ | Epochs | MentalArithmetic | | |
|---|---|---|---|---|
| | | ACC | AUC-PR | AUROC |
| 256 | 2 | $0.530 \pm 0.071$ | $0.325 \pm 0.079$ | $0.625 \pm 0.078$ |
| | 4 | $0.537 \pm 0.066$ | $0.333 \pm 0.051$ | $0.620 \pm 0.046$ |
| | 8 | $0.710 \pm 0.086$ | $0.620 \pm 0.103$ | $0.811 \pm 0.054$ |
| | 16 | $0.567 \pm 0.063$ | $0.516 \pm 0.102$ | $0.767 \pm 0.042$ |
| | 32 | $0.647 \pm 0.025$ | $0.504 \pm 0.069$ | $0.734 \pm 0.033$ |
| 512 | 2 | $0.586 \pm 0.051$ | $0.457 \pm 0.069$ | $0.700 \pm 0.042$ |
| | 4 | $0.623 \pm 0.043$ | $0.505 \pm 0.061$ | $0.766 \pm 0.022$ |
| | 8 | $0.639 \pm 0.057$ | $0.466 \pm 0.149$ | $0.728 \pm 0.074$ |
| | 16 | $0.712 \pm 0.014$ | $0.643 \pm 0.049$ | $0.803 \pm 0.019$ |
| | 32 | $0.728 \pm 0.030$ | $0.672 \pm 0.030$ | $0.805 \pm 0.028$ |
| 768 | 2 | $0.569 \pm 0.033$ | $0.367 \pm 0.067$ | $0.610 \pm 0.044$ |
| | 4 | $0.577 \pm 0.063$ | $0.433 \pm 0.052$ | $0.693 \pm 0.032$ |
| | 8 | $\mathbf{0.738 \pm 0.034}$ | $0.615 \pm 0.046$ | $0.811 \pm 0.027$ |
| | 16 | $0.692 \pm 0.033$ | $\mathbf{0.709 \pm 0.025}$ | $0.819 \pm 0.019$ |
| | 32 | $0.710 \pm 0.025$ | $0.705 \pm 0.040$ | $0.824 \pm 0.029$ |
| 1024 | 2 | $0.576 \pm 0.032$ | $0.393 \pm 0.065$ | $0.648 \pm 0.053$ |
| | 4 | $0.704 \pm 0.044$ | $0.601 \pm 0.064$ | $0.800 \pm 0.031$ |
| | 8 | $0.574 \pm 0.016$ | $0.395 \pm 0.017$ | $0.660 \pm 0.020$ |
| | 16 | $0.626 \pm 0.032$ | $0.470 \pm 0.066$ | $0.770 \pm 0.050$ |
| | 32 | $0.693 \pm 0.020$ | $0.561 \pm 0.034$ | $0.792 \pm 0.015$ |
| 2048 | 2 | $0.630 \pm 0.076$ | $0.610 \pm 0.073$ | $\mathbf{0.848 \pm 0.028}$ |
| | 4 | $0.500 \pm 0.000$ | $0.596 \pm 0.065$ | $0.501 \pm 0.003$ |
| | 8 | $0.554 \pm 0.008$ | $0.352 \pm 0.057$ | $0.621 \pm 0.047$ |
| | 16 | $0.717 \pm 0.038$ | $0.618 \pm 0.073$ | $0.816 \pm 0.028$ |
| | 32 | $0.690 \pm 0.025$ | $0.644 \pm 0.059$ | $0.795 \pm 0.028$ |

# D EXTENDED RESULTS

## D.1 COMPREHENSIVE DIVER DOWNSTREAM TASK RESULTS

We present the detailed numerical values. The EEG performance table is provided in Table 14. The iEEG performance table is provided in Table 12 and Table 13. For Neuroprobe binary-label (iEEG task), we evaluated $\text{DIVER}_{Tiny/I/0.1s}$ (with $d_{\text{model}} = 256$ and patch size 0.1s, pretrained on iEEG for 32 epochs) in both frozen (linear probing, red) and fine-tuned (blue) configurations against Linear Laplacian STFT, BrainBERT and PopT baselines on Figure 2. Table 13 presents comprehensive results comparing DIVER with baseline models across all 15 Neuroprobe tasks. For Neuroprobe multi-label, our model are still overperform the linear baseline. Results are reported as mean AUROC $\pm$ SEM across subjects, trials, and cross-validation folds. DIVER consistently outperformed baseline models across the majority of tasks in both evaluation settings. The model demonstrated particularly strong performance on language-related tasks (speech decoding, word prediction, onset detection) and auditory tasks (volume, pitch), with finetuning providing additional gains over frozen features. These results validate that self-supervised pretraining on iEEG data produces representations that transfer effectively to diverse downstream neural decoding tasks. In the multi-label setting, DIVER again outperformed the linear baseline for both frozen and fully fine-tuned models.[4]

For MAYO (iEEG task), we evaluated $\text{DIVER}_{Tiny/I/1s}$ (with $d_{\text{model}} = 256$ and 1 s patch size, pretrained on iEEG for 32 epochs) in both frozen (linear probing) and full-finetuning settings. We could not evaluate PopT because the dataset does not contain any coordinates. By contrast, although our model is trained with 3D positional embeddings, it can handle missing position information by replacing the positional embedding with a zero vector for electrodes with unknown location. Among the different baselines (BrainBERT frozen, Brant frozen, and Brant full-finetuning), $\text{DIVER}_{Tiny/I/1s}$ achieved the best performance. Brant exhibited a substantial performance drop under full finetuning, likely because we deviated from its original pretraining configuration by using only a single patch, which can impair optimization when updating all parameters with original context windows (90s), whereas the frozen setting simply uses the fixed embedding.

Table 12: **Comparison of the DIVER-1 iEEG model with other baseline models.** We evaluated $\text{DIVER}_{Tiny/I}$ models that were pretrained for 32 epochs on 100% of the iEEG pretraining dataset. For Neuroprobe (1 s), we compare $\text{DIVER}_{Tiny/I/0.1}$ with other baselines using an overall score defined as the mean AUROC averaged over all 15 tasks, subjects, trials, and folds. For MAYO (6 s), we compare $\text{DIVER}_{Tiny/I/1s}$ with Brant and BrainBERT; scores are reported as mean AUROC $\pm$ SEM across 8 subjects and 3 folds.

| | Neuroprobe (overall) | | MAYO |
|---|---|---|---|
| | binary-label | multi-label | |
| linear (stft-laplacian) | $0.660 \pm 0.005$ | $0.617 \pm 0.007$ | - |
| PopT | $0.545 \pm 0.006$ | - | - |
| BrainBERT (frozen) | $0.586 \pm 0.004$ | - | $0.748 \pm 0.038$ |
| Brant | - | - | $0.551 \pm 0.023$ |
| Brant (frozen) | - | - | $0.757 \pm 0.042$ |
| $\text{DIVER}_{Tiny/I/0.1or1s}$ | $0.662 \pm 0.008$ | $0.621 \pm 0.007$ | $\mathbf{0.961 \pm 0.011}$ |
| $\text{DIVER}_{Tiny/I/0.1or1s}$ (frozen) | $\mathbf{0.676 \pm 0.007}$ | $\mathbf{0.631 \pm 0.007}$ | $0.935 \pm 0.012$ |

---

[4]A multi-label option is available in the most recent release of the Neuroprobe code, but it has not yet been documented in the paper. Consequently, we compare only against the linear (STFT–Laplacian) baseline, witch can be easily compute by running their released code.

Table 13: **Downstream performance of each task in Neuroprobe.** We compare DIVER with existing models on comprehensive iEEG downstream tasks. We evaluated both the fine-tuned and frozen configurations of $DIVER_{Tiny/I/0.1s}$ (pretrained on iEEG for 32 epochs) against Linear Laplacian STFT, BrainBERT, and popT. Results are reported as mean AUROC $\pm$ SEM across subjects, trials, and folds. Overall, DIVER consistently outperformed baselines across the majority of tasks.

| Models | Overall | Sentence Onset | Speech | Volume |
|---|---|---|---|---|
| Linear Laplacian STFT | $0.660 \pm 0.005$ | $0.891 \pm 0.018$ | $0.883 \pm 0.018$ | $0.717 \pm 0.032$ |
| BrainBERT (frozen) | $0.586 \pm 0.004$ | $0.757 \pm 0.027$ | $0.611 \pm 0.022$ | $0.583 \pm 0.010$ |
| PopT | $0.545 \pm 0.006$ | $0.689 \pm 0.050$ | $0.677 \pm 0.044$ | $0.576 \pm 0.018$ |
| $DIVER_{Tiny/I/0.1s}$ | $0.662 \pm 0.008$ | $0.924 \pm 0.009$ | $\mathbf{0.900 \pm 0.011}$ | $0.699 \pm 0.020$ |
| $DIVER_{Tiny/I/0.1s}$(frozen) | $\mathbf{0.676 \pm 0.007}$ | $\mathbf{0.930 \pm 0.008}$ | $0.896 \pm 0.012$ | $\mathbf{0.717 \pm 0.018}$ |

| Models | Delta Volume | Voice Pitch | Word position | Inter-word Gap |
|---|---|---|---|---|
| Linear Laplacian STFT | $0.762 \pm 0.026$ | $0.578 \pm 0.016$ | $0.740 \pm 0.028$ | $0.612 \pm 0.014$ |
| BrainBERT (frozen) | $0.706 \pm 0.021$ | $0.524 \pm 0.007$ | $0.685 \pm 0.027$ | $0.584 \pm 0.017$ |
| PopT | $0.628 \pm 0.025$ | $0.509 \pm 0.008$ | $0.519 \pm 0.023$ | $0.509 \pm 0.009$ |
| $DIVER_{Tiny/I/0.1s}$ | $\mathbf{0.812 \pm 0.017}$ | $0.563 \pm 0.007$ | $0.777 \pm 0.016$ | $0.623 \pm 0.014$ |
| $DIVER_{Tiny/I/0.1s}$ (frozen) | $0.809 \pm 0.016$ | $\mathbf{0.589 \pm 0.007}$ | $\mathbf{0.791 \pm 0.014}$ | $\mathbf{0.628 \pm 0.011}$ |

| Models | GPT-2 Surprisal | Head Word Pos | Part of Speech | Word Length |
|---|---|---|---|---|
| Linear Laplacian STFT | $0.613 \pm 0.017$ | $0.602 \pm 0.012$ | $0.605 \pm 0.012$ | $0.618 \pm 0.015$ |
| BrainBERT (frozen) | $0.580 \pm 0.015$ | $0.585 \pm 0.013$ | $0.556 \pm 0.012$ | $0.571 \pm 0.012$ |
| PopT | $0.523 \pm 0.014$ | $0.519 \pm 0.008$ | $0.513 \pm 0.004$ | $0.505 \pm 0.005$ |
| $DIVER_{Tiny/I/0.1s}$ | $0.617 \pm 0.009$ | $0.613 \pm 0.009$ | $0.597 \pm 0.011$ | $0.638 \pm 0.011$ |
| $DIVER_{Tiny/I/0.1s}$ (frozen) | $\mathbf{0.628 \pm 0.009}$ | $\mathbf{0.622 \pm 0.009}$ | $\mathbf{0.624 \pm 0.011}$ | $\mathbf{0.642 \pm 0.013}$ |

| Models | Global Flow | Local Flow | Frame Brightness | Num of Faces |
|---|---|---|---|---|
| Linear Laplacian STFT | $\mathbf{0.625 \pm 0.054}$ | $0.607 \pm 0.017$ | $\mathbf{0.521 \pm 0.025}$ | $\mathbf{0.530 \pm 0.014}$ |
| BrainBERT (frozen) | $0.521 \pm 0.006$ | $0.525 \pm 0.003$ | $0.508 \pm 0.012$ | $0.503 \pm 0.007$ |
| PopT | $0.509 \pm 0.008$ | $0.508 \pm 0.014$ | $0.499 \pm 0.019$ | $0.492 \pm 0.010$ |
| $DIVER_{Tiny/I/0.1s}$ | $0.587 \pm 0.010$ | $0.586 \pm 0.012$ | $0.492 \pm 0.015$ | $0.509 \pm 0.007$ |
| $DIVER_{Tiny/I/0.1s}$ (frozen) | $0.620 \pm 0.009$ | $\mathbf{0.614 \pm 0.012}$ | $0.502 \pm 0.012$ | $0.523 \pm 0.010$ |

Table 14: **Comparison of DIVER-1 EEG model with other baseline models.** We compare DIVER with existing state-of-the-art models on three EEG downstream tasks using their reported values from the original papers. DIVER consistently outperformed baseline models across all metrics.

| Models | FACED (9-class) | | |
|---|---|---|---|
| | ACC | kappa | F1 |
| LaBraM | $0.527 \pm 0.011$ | $0.470 \pm 0.019$ | $0.529 \pm 0.010$ |
| CBraMod | $0.551 \pm 0.009$ | $0.504 \pm 0.012$ | $0.562 \pm 0.009$ |
| **DIVER (Ours)** | $\mathbf{0.601 \pm 0.008}$ | $\mathbf{0.550 \pm 0.009}$ | $\mathbf{0.607 \pm 0.009}$ |

| Models | PhysioNet-MI (4-class) | | |
|---|---|---|---|
| | ACC | kappa | F1 |
| LaBraM | $0.617 \pm 0.012$ | $0.491 \pm 0.019$ | $0.618 \pm 0.014$ |
| CBraMod | $0.642 \pm 0.009$ | $0.522 \pm 0.017$ | $0.643 \pm 0.010$ |
| **DIVER (Ours)** | $\mathbf{0.676 \pm 0.003}$ | $\mathbf{0.567 \pm 0.004}$ | $\mathbf{0.678 \pm 0.004}$ |

| Models | MentalArithmetic (2-class) | | |
|---|---|---|---|
| | ACC | AUC-PR | AUROC |
| LaBraM | $0.691 \pm 0.013$ | $0.600 \pm 0.016$ | $0.772 \pm 0.009$ |
| CBraMod | $0.726 \pm 0.013$ | $0.627 \pm 0.010$ | $0.791 \pm 0.007$ |
| **DIVER (Ours)** | $\mathbf{0.727 \pm 0.018}$ | $\mathbf{0.676 \pm 0.046}$ | $\mathbf{0.814 \pm 0.026}$ |

## D.2 PERFORMANCE EVALUATION ACROSS DIVER-1 MODEL CONFIGURATIONS

We first evaluated $DIVER_{Tiny/I/.}$'s performance in Neuroprobe, based on different model configurations including the patch size, Laplacian re-referencing and training settings. The 0.1s model outperformed the 1s model (Table 15) in 4 tasks in Neuroprobe. Considering that the Neuroprobe

dataset consists of 1-second samples and its tasks require classifying short-timescale features such as speech and onset, the effectiveness of 0.1s model may be explained. Additionally, the model without Laplacian re-referencing generally showed degraded performance, indicating the effectiveness of the pre-processing method . Further, we examined the effect of pretraining by comparing the performance of DIVER$_{Tiny/I/0.1s}$ with diverse training settings. The model trained from scratch showed significantly degraded performance than the full-finetuned and backbone-frozen models. Such results indicate the efficacy of pretraining. Specifically, the model with a frozen backbone showed the highest performance, except for the speech task.

Table 15: **Performance evaluation between various DIVER model configurations in Neuroprobe (iEEG tasks).** We first compare DIVER$_{Tiny/I}$ models by patch size, Laplacian re-referencing, and training settings. The model from scratch was trained on only four tasks (speech, onset, volume, pitch) due to computational constraints, consequently; model evaluation is limited to these four tasks. For the backbone-frozen models, 0.1s variants with different sizes were trained on four tasks, whereas the tiny model was trained on all Neuroprobe tasks. The results are reported as mean AUROC ± SEM across multiple subjects, trials, and folds. All models, except the one trained from scratch, were pretrained for 32 epochs on 100% of the iEEG pretraining dataset.

| | speech | onset | volume | pitch | neuroprobe total |
|---|---|---|---|---|---|
| $Tiny/I/1s$ | $0.828 \pm 0.016$ | $0.885 \pm 0.012$ | $0.634 \pm 0.018$ | $0.551 \pm 0.009$ | $0.645 \pm 0.007$ |
| $Tiny/I/0.1s$ | $\mathbf{0.900 \pm 0.011}$ | $0.924 \pm 0.009$ | $0.699 \pm 0.020$ | $0.563 \pm 0.007$ | $0.662 \pm 0.008$ |
| $Tiny/I/0.1s$ (w.o. laplacian) | $0.862 \pm 0.018$ | $0.901 \pm 0.013$ | $0.662 \pm 0.018$ | $0.533 \pm 0.004$ | $0.642 \pm 0.007$ |
| $Tiny/I/0.1s$ (from scratch) | $0.832 \pm 0.014$ | $0.872 \pm 0.010$ | $0.622 \pm 0.016$ | $0.554 \pm 0.006$ | - |
| $Tiny/I/0.1s$ (frozen) | $0.896 \pm 0.012$ | $\mathbf{0.930 \pm 0.008}$ | $\mathbf{0.717 \pm 0.018}$ | $\mathbf{0.589 \pm 0.007}$ | $\mathbf{0.676 \pm 0.007}$ |
| $Small/I/0.1s$ (frozen) | $0.888 \pm 0.012$ | $0.926 \pm 0.009$ | $0.705 \pm 0.016$ | $0.578 \pm 0.006$ | - |
| $Large/I/0.1s$ (frozen) | $0.890 \pm 0.0126$ | $0.928 \pm 0.009$ | $0.710 \pm 0.017$ | $0.581 \pm 0.007$ | - |
| $XL/I/0.1s$ (frozen) | $0.893 \pm 0.012$ | $0.930 \pm 0.009$ | $0.713 \pm 0.017$ | $0.582 \pm 0.007$ | - |

And for MAYO (seizure detection), we chosed only 1 sec model, because MAYO has 6 s window that is too long for 0.1 s model's context. We compared the frozen and full finetuning model in each model size in Table 16. In contrast to the Neuroprobe results, we found that full fine-tuning outperformed the frozen for the Tiny, Small and XL model, even though it was slightly worse for the Large and Base models. Since our preprocessing clips signal amplitudes above a certain threshold (200 μV), so pretrained-dataset's distribution can differ from seizure data, which contains spikes with much larger amplitudes; under this distribution shift, fully fine-tuned models may therefore tend to achieve better performance. For the linear baseline, the highest performance was obtained with the Large model, whereas under full-finetuning the Tiny model achieved the best performance.

Table 16: **Performance evaluation between various DIVER model configurations in MAYO (iEEG task).** DIVER 1 s models were trained for each model size, and we compared the frozen and full fine-tuning variants. The results are reported as mean AUROC ± SEM across multiple subjects and folds. All models, except were pretrained for 32 epochs on 100% of the iEEG pretraining dataset.

| | Tiny | Small | Base | Large | XL |
|---|---|---|---|---|---|
| frozen | $0.935 \pm 0.012$ | $0.904 \pm 0.019$ | $0.911 \pm 0.017$ | $0.937 \pm 0.015$ | $0.914 \pm 0.018$ |
| full-finetuned | $\mathbf{0.961 \pm 0.011}$ | $0.927 \pm 0.020$ | $0.905 \pm 0.026$ | $0.934 \pm 0.014$ | $0.947 \pm 0.019$ |

### D.3 ABLATIONS

**Input ablations** Previously in Table15 we confirmed that the iEEG models' downstream performance improve when Laplacian re-referencing is used. PopT and BrainBERT were pretrained on Laplacian re-referenced signals, and their downstream performances were also derived under that setting. Even though our model was trained on raw signals (not referenced), it was better with Laplacian referencing (Table 15). Therefore, we use Laplacian re-referencing as the default setting for finetuning on iEEG downstream tasks.

**Architecture ablations** To assess whether encoder components (RoPE and any variate attention), embedding components, and multi-domain reconstruction task are effective, we removed each ele-

Table 17: **Architecture ablation for iEEG downstream tasks.** Tasks include speech decoding, onset detection, volume prediction, and pitch estimation tasks. DIVER$_{Tiny/I/0.1s}$ (bottom row) represents the full model with all components. Each row indicates the model's performance when removing a specific component. All models use 12 layers with $d_{model} = 256$ and were pretrained for 8 epochs. Results are reported as mean AUROC $\pm$ SEM across multiple subjects, trials and folds.

| | speech | onset | volume | pitch |
|---|---|---|---|---|
| w.o. RoPE | $0.886 \pm 0.013$ | $0.916 \pm 0.011$ | $0.693 \pm 0.019$ | $0.579 \pm 0.008$ |
| w.o anyV attention | $0.889 \pm 0.013$ | $0.919 \pm 0.009$ | $0.699 \pm 0.018$ | $0.579 \pm 0.007$ |
| w.o RoPE and anyV attention | $0.870 \pm 0.014$ | $0.898 \pm 0.012$ | $0.669 \pm 0.018$ | $0.560 \pm 0.006$ |
| w.o. STCPE | $0.879 \pm 0.014$ | $0.911 \pm 0.013$ | $0.686 \pm 0.019$ | $0.572 \pm 0.007$ |
| w.o Channel modality + subtype emb. | $0.892 \pm 0.011$ | $0.919 \pm 0.009$ | $0.690 \pm 0.017$ | $0.577 \pm 0.006$ |
| w.o Channel 3d position emb. | $\mathbf{0.900 \pm 0.011}$ | $\mathbf{0.927 \pm 0.009}$ | $\mathbf{0.710 \pm 0.018}$ | $\mathbf{0.584 \pm 0.009}$ |
| w.o Spectral feature emb. | $0.885 \pm 0.013$ | $0.919 \pm 0.010$ | $0.694 \pm 0.019$ | $0.571 \pm 0.010$ |
| w.o Multi-domain reconstruction (only raw) | $0.875 \pm 0.014$ | $0.916 \pm 0.010$ | $0.680 \pm 0.017$ | $0.569 \pm 0.006$ |
| DIVER$_{Tiny/I/0.1s}$ | $0.890 \pm 0.013$ | $0.922 \pm 0.009$ | $0.698 \pm 0.018$ | $0.572 \pm 0.008$ |

ment and evaluated the corresponding performance. Ablation studies on iEEG were conducted on DIVER$_{Tiny/I/0.1s}$ trained for 8 epochs with full pretraining dataset, and for EEG, DIVER$_{Tiny/E/1s}$ trained for 2 epochs with 10% of the data were used, due to computational constraints and time limitations. Architecture ablation results for iEEG are given in Table 17. When each encoder component was removed individually, the performance varied across tasks, but dropped noticeably when both were excluded. An ablation of STCPE and multi-domain reconstruction task each induced performance degradation, whereas the ablation of other embedding components did not yield significant changes. Since the Neuroprobe dataset includes only depth electrodes (SEEG), the effect of channel modality and subtype embedding may be minimal. Moreover, as the dataset utilizes only child and adolescent subjects, whose brain volumes differ by age, the effect of the channel 3D positional embedding may be attenuated.

For EEG downstream tasks, detailed results are described in Table 18. The ablation of encoder components yielded a significant performance decline in both EEG tasks, indicating that the encoder components are crucial contributors to overall performance. Removing the multi-domain reconstruction task, STCPE and spectral feature embedding resulted in a notable performance degradation as well. Ablation of channel-wise patch embedding components induced inconsistent results across tasks; for FACED, performance dropped while for PhysioNet-MI, performance slightly improved.

Incorporating encoder components and multi-domain reconstruction task significantly improved the model's performance in both iEEG and EEG downstream tasks. Specifically, in multi-domain reconstruction, removing each component for STFT and FFT also degrades performance, which shows that each element of multi-domain reconstruction is important. Channel-wise embedding components however differed in their effects depending on the modality and the type of downstream tasks. Since informative features in Ephys signals can vary across modalities and tasks, holding these various components may help improve generalization across a range of tasks.

**Dataset ablations and dataset size-based comparisons** The other iEEG models (PopT (Chau et al., 2025) and BrainBERT (Wang et al., 2023)) are pretrained on the BrainTreebank (BTB) (Wang et al., 2024a) datasets—the precursor to Neuroprobe. Therefore, we compare downstream performance when we use BTB exclusively (Figure 12). We trained DIVER$_{Tiny/I/0.1s}$ models with BTB and size-variations of our self-collected iEEG datasets. For the model trained on our self-collected data of the same size as BTB, the linear-probing results were lower than those trained with BTB only. However, when we increased the size of the self-collected dataset (approximately ×16 and ×64 size of BTB), performance surpassed the BTB-only setting. This shows that, with sufficient data, it is possible to achieve higher performance even if the distribution of the pretraining dataset differs from that of the downstream tasks.

**Comparison of DIVER-1 iEEG model with other baseline models on a shared dataset** Since the pretraining dataset between iEEG baseline models and our model differentiated, we additionally trained the DIVER model on the pretraining dataset of the baseline models. The corresponding results are shown in Table 19. The model trained under our own early-stopping strategy is denoted as "ours." Given the same pretraining dataset, the DIVER model achieved the highest performance compared to the two baseline models.

Table 18: **Architecture ablation for EEG downstream tasks.** Tasks include FACED and PhysioNet-MI dataset. $\text{DIVER}_{Tiny/E/1s}$ represents the full model with all components. Each row shows performance when removing a specific component. All models use 12 layers with $d_{\text{model}} = 256$ and were pretrained for 2 epochs. Results are reported as mean $\pm$ standard deviation across 5 random seeds.

| | FACED (9-class) | | |
|---|---|---|---|
| | ACC | kappa | F1 |
| w.o. RoPE | $0.408 \pm 0.025$ | $0.333 \pm 0.027$ | $0.414 \pm 0.024$ |
| w.o anyV attention | $0.414 \pm 0.012$ | $0.339 \pm 0.014$ | $0.417 \pm 0.012$ |
| w.o RoPE and anyV attention | $0.446 \pm 0.007$ | $0.376 \pm 0.007$ | $0.450 \pm 0.005$ |
| w.o. STCPE | $0.463 \pm 0.024$ | $0.395 \pm 0.027$ | $0.471 \pm 0.026$ |
| w.o Channel modality + subtype emb. | $0.474 \pm 0.018$ | $0.406 \pm 0.021$ | $0.482 \pm 0.019$ |
| w.o Channel 3d position emb. | $0.481 \pm 0.016$ | $0.415 \pm 0.018$ | $0.487 \pm 0.016$ |
| w.o Spectral feature emb. | $0.454 \pm 0.015$ | $0.386 \pm 0.016$ | $0.462 \pm 0.013$ |
| w.o Multi-domain reconstruction (only raw) | $0.435 \pm 0.008$ | $0.364 \pm 0.009$ | $0.437 \pm 0.006$ |
| w.o FFT reconstruction (raw and stft) | $0.468 \pm 0.007$ | $0.401 \pm 0.008$ | $0.479 \pm 0.008$ |
| w.o STFT reconstruction (raw and fft) | $0.485 \pm 0.014$ | $0.418 \pm 0.015$ | $0.487 \pm 0.013$ |
| $\text{DIVER}_{Tiny/E/1s}$ | $\mathbf{0.491 \pm 0.023}$ | $\mathbf{0.428 \pm 0.025}$ | $\mathbf{0.502 \pm 0.023}$ |

| | PhysioNet-MI (4-class) | | |
|---|---|---|---|
| | ACC | kappa | F1 |
| w.o. RoPE | $0.614 \pm 0.005$ | $0.485 \pm 0.006$ | $0.615 \pm 0.005$ |
| w.o anyV attention | $0.611 \pm 0.003$ | $0.481 \pm 0.004$ | $0.612 \pm 0.004$ |
| w.o RoPE and anyV attention | $0.591 \pm 0.005$ | $0.454 \pm 0.006$ | $0.593 \pm 0.005$ |
| w.o. STCPE | $0.626 \pm 0.006$ | $0.502 \pm 0.008$ | $0.627 \pm 0.006$ |
| w.o Channel modality + subtype emb. | $0.629 \pm 0.006$ | $0.505 \pm 0.007$ | $\mathbf{0.632 \pm 0.005}$ |
| w.o Channel 3d position emb. | $\mathbf{0.629 \pm 0.008}$ | $\mathbf{0.506 \pm 0.010}$ | $0.631 \pm 0.007$ |
| w.o Spectral feature emb. | $0.626 \pm 0.005$ | $0.501 \pm 0.006$ | $0.627 \pm 0.004$ |
| w.o Multi-domain reconstruction (only raw) | $0.614 \pm 0.005$ | $0.485 \pm 0.006$ | $0.616 \pm 0.004$ |
| w.o FFT reconstruction (raw and stft) | $0.626 \pm 0.006$ | $0.502 \pm 0.008$ | $0.628 \pm 0.006$ |
| w.o STFT reconstruction (raw and fft) | $0.615 \pm 0.005$ | $0.487 \pm 0.007$ | $0.617 \pm 0.005$ |
| $\text{DIVER}_{Tiny/E/1s}$ | $0.628 \pm 0.005$ | $0.504 \pm 0.007$ | $0.630 \pm 0.005$ |

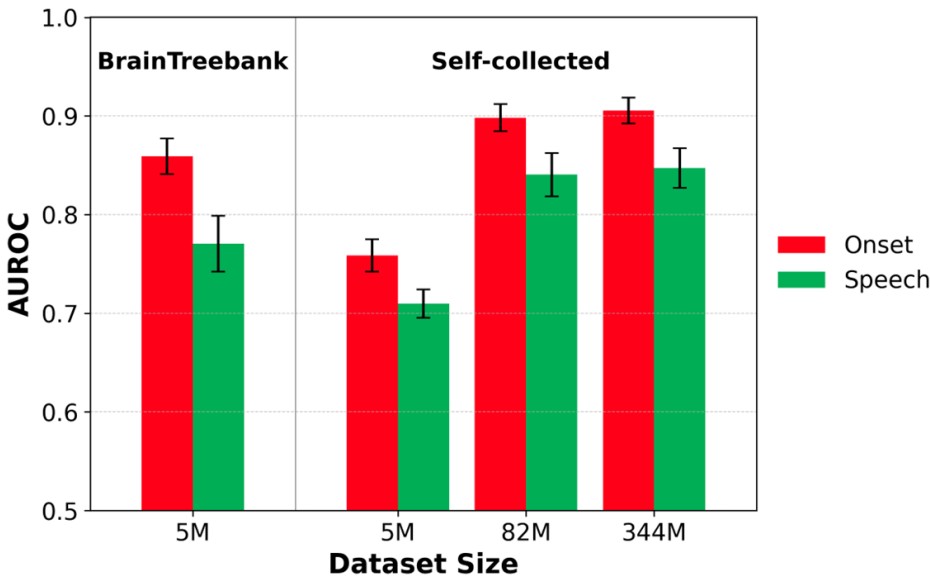

Figure 12: Pretraining dataset and size effects on performance: BrainTreebank vs. Self-Collected

Table 19: **Model comparison with a shared pretraining dataset.** Each row indicates a model architecture with its pretraining dataset in parentheses. Models pretrained on the same BTB dataset are compared to isolate architectural effects. The results are reported as mean AUROC ± SEM across multiple subjects, trials and folds. The DIVER model was pretrained for 32 epochs.

|  | speech | onset |
|---|---|---|
| BrainBERT (frozen) ((Zahorodnii et al., 2025) | $0.611 \pm 0.022$ | $0.757 \pm 0.027$ |
| BrainBERT (our training code) | $0.575 \pm 0.018$ | $0.659 \pm 0.026$ |
| PopT (our training code) | $0.702 \pm 0.029$ | $0.780 \pm 0.025$ |
| PopT ((Zahorodnii et al., 2025) | $0.677 \pm 0.044$ | $0.689 \pm 0.050$ |
| DIVER$_{Tiny/I/0.1s}$ (frozen) | $\mathbf{0.770 \pm 0.028}$ | $\mathbf{0.859 \pm 0.018}$ |

### D.4 ANALYSIS OF JOINT MODALITY (INTRACRANIAL/SCALP-EEG) PRETRAINING

We examined the effects of joint training on both iEEG and EEG downstream tasks, with detailed results presented in Table 20. Joint training showed contrasting effects on the two modalities: it decreased performance on iEEG benchmarks but improved performance on EEG tasks.

This difference may stem from the signal quality disparity between the two modalities. Since EEG signals are inherently noisier than iEEG signals, incorporating EEG data during joint training may introduce noise that degrades the model's ability to process high-quality iEEG signals. Conversely, for EEG tasks, joint training provides EEG specific information that iEEG cannot provide. Additionally, the data imbalance between modalities may contribute to these results. Since EEG data outnumbers iEEG data, the model may become biased toward EEG-specific patterns during joint training. As detailed in Table 20, training exclusively on curated EEG datasets achieves the best performance on EEG downstream tasks, outperforming models that include iEEG data.

Table 20: **Joint pretraining results on downstream tasks.** Performance compared between $\text{DIVER}_{Tiny/I/1s}$ (trained on iEEG only) and $\text{DIVER}_{Tiny/IE/1s}$ (trained on iEEG and EEG), both with $d_{model} = 256$. Performance across iEEG tasks (speech and onset, measured by AUROC) and EEG task (FACED, measured by ACC) are shown. Joint training with EEG data improves performance on EEG tasks but slightly decreases performance on iEEG tasks.

|  | speech (iEEG, AUROC) | onset (iEEG, AUROC) | FACED (EEG, ACC) |
|---|---|---|---|
| $Tiny/I/1s$ (frozen) | $\mathbf{0.854 \pm 0.011}$ | $\mathbf{0.906 \pm 0.008}$ | $0.328 \pm 0.003$ |
| $Tiny/IE/1s$ (frozen) | $0.817 \pm 0.018$ | $0.891 \pm 0.012$ | $\mathbf{0.359 \pm 0.004}$ |

Table 21: **Ablation on pretraining data composition for EEG downstream tasks.** We evaluated how different pretraining dataset combinations affect performance on EEG downstream tasks. We compared models pretrained on: TUEG (Obeid & Picone, 2016)-only (largest single EEG dataset), TUEG+iEEG, all EEG datasets, and EEG+iEEG (all available data). All models use $\text{DIVER}_{Small/E/1s}$ architecture with $d_{\text{model}} = 512$, pretrained for 2 epochs. The model trained exclusively on curated EEG datasets achieves the best performance, outperforming models that include iEEG data.

| Pretraining Data | FACED (9-class) | | | PhysioNet-MI (4-class) | | |
|---|---|---|---|---|---|---|
|  | ACC | kappa | F1 | ACC | kappa | F1 |
| TUEG-only | $0.519 \pm 0.004$ | $0.456 \pm 0.005$ | $0.518 \pm 0.005$ | $0.623 \pm 0.010$ | $0.497 \pm 0.013$ | $0.625 \pm 0.009$ |
| TUEG + iEEG | $0.461 \pm 0.011$ | $0.394 \pm 0.013$ | $0.471 \pm 0.012$ | $0.599 \pm 0.008$ | $0.465 \pm 0.010$ | $0.600 \pm 0.008$ |
| EEG + iEEG | $0.540 \pm 0.013$ | $0.482 \pm 0.015$ | $0.550 \pm 0.014$ | $0.638 \pm 0.005$ | $0.517 \pm 0.006$ | $0.639 \pm 0.004$ |
| EEG | $\mathbf{0.570 \pm 0.009}$ | $\mathbf{0.515 \pm 0.010}$ | $\mathbf{0.579 \pm 0.008}$ | $\mathbf{0.644 \pm 0.005}$ | $\mathbf{0.526 \pm 0.006}$ | $\mathbf{0.646 \pm 0.005}$ |

## D.5 INTERPRETATION RESULTS

Figure 13 shows a visualization of representation analysis on downstream datasets using UMAP (McInnes et al., 2018). We examine the embeddings obtained from the pretrained $\text{DIVER}_{Tiny/I/0.1s}$ model on the test set of speech, onset, and volume tasks in neuroprobe dataset without any finetuning. The results indicate that DIVER learned meaningful iEEG representations from pretraining, thereby capturing label-relevant structure in downstream datasets even in the absence of finetuning.

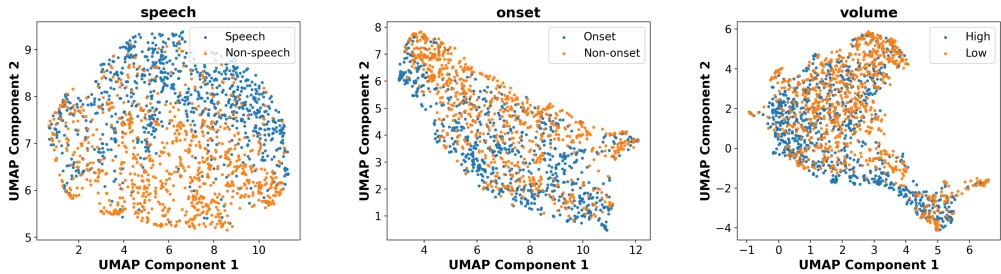

Figure 13: Visualizations of representations on neuroprobe downstream tasks. Each plot shows test set embeddings from one fold of a single trial from a single subject.

Table 22: **Comparison of finetuning architectures in EEG downstream tasks.** Deeper heads generally improve performance up to 3-4 layers.

| Head Depth | FACED (9-class) | | | PhysioNet-MI (4-class) | | |
|---|---|---|---|---|---|---|
| | ACC | kappa | F1 | ACC | kappa | F1 |
| Linear | $0.523 \pm 0.018$ | $0.461 \pm 0.020$ | $0.521 \pm 0.018$ | $0.653 \pm 0.013$ | $0.538 \pm 0.017$ | $0.654 \pm 0.013$ |
| 2-Layer | $0.584 \pm 0.005$ | $0.530 \pm 0.005$ | $0.584 \pm 0.005$ | $0.674 \pm 0.003$ | $0.565 \pm 0.004$ | $0.676 \pm 0.003$ |
| 3-Layer | $0.601 \pm 0.008$ | $0.550 \pm 0.009$ | $0.607 \pm 0.009$ | $\mathbf{0.676 \pm 0.003}$ | $\mathbf{0.567 \pm 0.004}$ | $\mathbf{0.678 \pm 0.004}$ |
| 4-Layer | $\mathbf{0.603 \pm 0.007}$ | $\mathbf{0.552 \pm 0.008}$ | $\mathbf{0.609 \pm 0.007}$ | $0.672 \pm 0.008$ | $0.573 \pm 0.011$ | $0.674 \pm 0.008$ |
| 5-Layer | $0.594 \pm 0.021$ | $0.543 \pm 0.023$ | $0.601 \pm 0.019$ | $0.660 \pm 0.007$ | $0.547 \pm 0.009$ | $0.662 \pm 0.006$ |

## E    EXPLORATION OF FINETUNING METHOD

### E.1    OVERVIEW:IMPACT OF FINETUNING METHODOLOGY ON EEG DOWNSTREAM TASK PERFORMANCE

We conduct comprehensive experiments to investigate the impact of different finetuning methods on downstream task performance. Our analysis includes: (1) systematic exploration of MLP classifier depth for DIVER(appendix subsection E.2), (2) reproduction of CBraMod (Wang et al., 2024c) performance using publicly available code and weights(appendix subsection E.3), and (3) comparative analysis of various finetuning configurations across both models E.4. These experiments reveal that finetuning methodology significantly affects model performance, and optimal configurations vary across tasks and models. Importantly, **when comparing DIVER and CBraMod under identical finetuning configurations (one-to-one comparison), DIVER demonstrates competitive or superior performance to CBraMod across multiple tasks, achieving overall state-of-the-art results** despite CBraMod's higher reported in-paper performance on some tasks.

### E.2    IMPACT OF MLP CLASSIFIER DEPTH ON DIVER PERFORMANCE

Initially, we employed a linear classifier head for finetuning but observed suboptimal performance. Upon examining the publicly available code of CBraMod (Wang et al., 2024c), the previous state-of-the-art model, we found that it uses an MLP classifier head. To ensure fair comparison, we replaced the linear head with an MLP classifier, which resulted in substantial performance improvements for DIVER-1. This motivated us to conduct systematic experiments investigating how MLP depth affects performance. Table 22 compares the performance across the depths of MLP classifier for downstream EEG tasks. The original 3-layer MLP classifier is varied between 1 to 5 layers. For both FACED and PhysioNet-MI tasks, performance improved with the increase of MLP depth, achieving peak performance at 4-layer for FACED (balanced accuracy of 0.603) and 3-layer for PhysioNet-MI (balanced accuracy of 0.676). Beyond the optimal depth, we observed performance saturation or slight degradation, particularly notable in the 5-layer MLP for both datasets. These results indicate that a moderate depth (3-4 layers) suffices model's effectiveness across different EEG downstream tasks.

### E.3    CHALLENGES IN REPRODUCING CBRAMOD BASELINE PERFORMANCE

While we found that finetuning method significantly impacts downstream task performance, the CBraMod paper does not specify which finetuning methods were used for each downstream task. Therefore, we conducted experiments to reproduce CBraMod's performance using the default configuration from their publicly released code and weights. To ensure faithful reproduction, we used CBraMod's preprocessing pipeline, pretrained weights, and finetuning code without modification.

The experimental results revealed substantial performance gaps on several tasks (Table 23). On MentalArithmetic(Zyma et al., 2019), reproduced accuracy (0.619) fell short of reported performance (0.726); on TUEV(Obeid & Picone, 2016), accuracy decreased from 0.667 to 0.605; on Mumtaz2016(Zyma et al., 2019), from 0.956 to 0.882. Only FACED(Chen et al., 2023) and PhysioNet-MI(Goldberger et al., 2000; Schalk et al., 2004) showed relatively successful reproduction. Notably, similar reproduction difficulties with CBraMod have been reported in recent work (Wang et al.,

Table 23: **Comparison of DIVER-1 EEG model with CBraMod.** CBraMod (in paper) refers to the performance reported in the original CBraMod paper, while CBraMod (reproduction) represents our reproduction using the default configuration from their publicly released code and weights.

| Models | FACED (9-class) | | |
|---|---|---|---|
| | ACC | kappa | F1 |
| CBraMod(in paper) | $0.551 \pm 0.009$ | $0.504 \pm 0.012$ | $0.562 \pm 0.009$ |
| CBraMod(reproduction) | $0.570 \pm 0.005$ | $0.514 \pm 0.006$ | $0.574 \pm 0.006$ |
| **DIVER (Ours)** | $\mathbf{0.601 \pm 0.008}$ | $\mathbf{0.550 \pm 0.009}$ | $\mathbf{0.607 \pm 0.009}$ |

| Models | PhysioNet-MI (4-class) | | |
|---|---|---|---|
| | ACC | kappa | F1 |
| CBraMod(in paper) | $0.642 \pm 0.009$ | $0.522 \pm 0.017$ | $0.643 \pm 0.010$ |
| CBraMod(reproduction) | $0.621 \pm 0.002$ | $0.495 \pm 0.003$ | $0.622 \pm 0.003$ |
| **DIVER (Ours)** | $\mathbf{0.676 \pm 0.003}$ | $\mathbf{0.567 \pm 0.004}$ | $\mathbf{0.678 \pm 0.004}$ |

| Models | MentalArithmetic (2-class) | | |
|---|---|---|---|
| | ACC | AUC-PR | AUROC |
| CBraMod(in paper) | $0.726 \pm 0.013$ | $0.627 \pm 0.010$ | $0.791 \pm 0.007$ |
| CBraMod(reproduction) | $0.619 \pm 0.035$ | $0.533 \pm 0.064$ | $0.749 \pm 0.031$ |
| **DIVER (Ours)** | $\mathbf{0.727 \pm 0.018}$ | $\mathbf{0.676 \pm 0.046}$ | $\mathbf{0.814 \pm 0.026}$ |

| Models | Mumtaz2016 (2-class) | | |
|---|---|---|---|
| | ACC | AUC-PR | AUROC |
| CBraMod(in paper) | $\mathbf{0.956 \pm 0.006}$ | $\mathbf{0.992 \pm 0.003}$ | $\mathbf{0.992 \pm 0.003}$ |
| CBraMod(reproduction) | $0.882 \pm 0.019$ | $0.976 \pm 0.007$ | $0.974 \pm 0.009$ |
| **DIVER (Ours)** | $0.894 \pm 0.006$ | $0.971 \pm 0.003$ | $0.968 \pm 0.005$ |

| Models | TUEV (6-class) | | |
|---|---|---|---|
| | ACC | kappa | F1 |
| CBraMod(in paper) | $\mathbf{0.667 \pm 0.011}$ | $\mathbf{0.677 \pm 0.010}$ | $\mathbf{0.834 \pm 0.006}$ |
| CBraMod(reproduction) | $0.605 \pm 0.024$ | $0.623 \pm 0.016$ | $0.802 \pm 0.009$ |
| **DIVER (Ours)** | $0.630 \pm 0.029$ | $0.527 \pm 0.039$ | $0.747 \pm 0.019$ |

2025), which also observed performance gaps between reported and reproduced results on certain tasks.

These reproduction challenges highlight the sensitivity of EEG foundation models to finetuning configurations. When comparing DIVER to the reproduced CBraMod baselines, the performance gaps narrow considerably: on TUEV, DIVER achieves 0.630 compared to reproduced CBraMod's 0.605; on MentalArithmetic, 0.727 vs 0.619. This underscores the importance of transparent reporting of finetuning protocols for fair model comparison.

### E.4 TASK-SPECIFIC OPTIMAL FINETUNING CONFIGURATIONS FOR CBRAMOD AND DIVER

As mentioned above, the default configuration from CBraMod's public code was insufficient to fully reproduce their reported performance. To maximize performance recovery, we experimented with various finetuning methods available in their codebase. We systematically explored five different configurations: (1) *no multi lr*: using a single learning rate for both backbone and head, (2) *multi lr multiplier 3/7*: setting the head learning rate to 3× or 7× the backbone learning rate, (3) *linear classifier*: full finetuning with a 1-layer linear head, (4) *linear probing*: freezing the backbone while training only a 1-layer linear head, and (5) *CBraMod finetuning method*: using a 3-layer MLP head with 5× learning rate multiplier as the default configuration.

As shown in Table 24, the optimal finetuning method for CBraMod varied across tasks. Specifically, for Mumtaz2016, the "no multi lr" configuration achieved the best performance (ACC: 0.920), while for TUEV, the "linear classifier" method performed best (ACC: 0.635). For MentalArithmetic, both "no multi lr" and the default CBraMod method showed comparable performance.

Table 24: **One-to-One Comparison between CBraMod and DIVER using fixed finetuning method.** Both models evaluated with identical finetuning configurations. CBraMod uses publicly released weights and code. DIVER model uses 12 layers with $d_{model} = 512$ and were pretrained for 32 epochs. Results are reported as mean $\pm$ standard deviation across 5 random seeds.

| Mumtaz2016 (2-class) | DIVER | | | CBraMod | | |
|---|---|---|---|---|---|---|
| | ACC | AUC-PR | AUROC | ACC | AUC-PR | AUROC |
| no multi lr | $0.894 \pm 0.006$ | $0.971 \pm 0.003$ | $0.968 \pm 0.005$ | $\mathbf{0.920 \pm 0.027}$ | $\mathbf{0.985 \pm 0.010}$ | $\mathbf{0.984 \pm 0.012}$ |
| multi lr multiplier 3 | $0.896 \pm 0.003$ | $0.980 \pm 0.001$ | $0.979 \pm 0.002$ | $0.892 \pm 0.010$ | $0.977 \pm 0.006$ | $0.974 \pm 0.008$ |
| multi lr multiplier 7 | $0.882 \pm 0.044$ | $0.981 \pm 0.008$ | $0.980 \pm 0.007$ | $0.888 \pm 0.012$ | $0.977 \pm 0.007$ | $0.975 \pm 0.008$ |
| linear classifier | $0.902 \pm 0.004$ | $0.986 \pm 0.005$ | $0.985 \pm 0.006$ | $0.872 \pm 0.059$ | $0.975 \pm 0.012$ | $0.973 \pm 0.014$ |
| linear probing | $\mathbf{0.944 \pm 0.003}$ | $\mathbf{0.990 \pm 0.000}$ | $\mathbf{0.990 \pm 0.000}$ | $0.515 \pm 0.003$ | $0.962 \pm 0.003$ | $0.972 \pm 0.001$ |
| CBraMod finetuning method[*] | $0.901 \pm 0.011$ | $0.985 \pm 0.006$ | $0.985 \pm 0.006$ | $0.882 \pm 0.019$ | $0.976 \pm 0.007$ | $0.974 \pm 0.009$ |

| TUEV (6-class) | DIVER | | | CBraMod | | |
|---|---|---|---|---|---|---|
| | ACC | kappa | F1 | ACC | kappa | F1 |
| no multi lr | $0.630 \pm 0.029$ | $0.527 \pm 0.039$ | $0.747 \pm 0.019$ | $0.609 \pm 0.021$ | $0.618 \pm 0.023$ | $0.801 \pm 0.012$ |
| multi lr multiplier 3 | $\mathbf{0.649 \pm 0.030}$ | $\mathbf{0.563 \pm 0.026}$ | $\mathbf{0.765 \pm 0.018}$ | $0.611 \pm 0.021$ | $\mathbf{0.628 \pm 0.027}$ | $\mathbf{0.805 \pm 0.013}$ |
| multi lr multiplier 7 | $0.648 \pm 0.036$ | $0.555 \pm 0.050$ | $0.761 \pm 0.025$ | $0.602 \pm 0.057$ | $0.624 \pm 0.031$ | $0.802 \pm 0.017$ |
| linear classifier | $0.611 \pm 0.024$ | $0.524 \pm 0.031$ | $0.753 \pm 0.016$ | $\mathbf{0.635 \pm 0.024}$ | $0.625 \pm 0.047$ | $0.804 \pm 0.024$ |
| linear probing | $0.559 \pm 0.025$ | $0.397 \pm 0.037$ | $0.624 \pm 0.041$ | $0.314 \pm 0.011$ | $0.307 \pm 0.014$ | $0.574 \pm 0.012$ |
| CBraMod finetuning method[*] | $0.612 \pm 0.014$ | $0.414 \pm 0.021$ | $0.644 \pm 0.017$ | $0.605 \pm 0.024$ | $0.623 \pm 0.016$ | $0.802 \pm 0.009$ |

| MentalArithmetic (2-class) | DIVER | | | CBraMod | | |
|---|---|---|---|---|---|---|
| | ACC | AUC-PR | AUROC | ACC | AUC-PR | AUROC |
| no multi lr | $0.727 \pm 0.018$ | $0.676 \pm 0.046$ | $0.814 \pm 0.026$ | $\mathbf{0.637 \pm 0.038}$ | $0.494 \pm 0.022$ | $0.747 \pm 0.031$ |
| multi lr multiplier 3 | $0.654 \pm 0.091$ | $0.666 \pm 0.070$ | $0.815 \pm 0.027$ | $0.629 \pm 0.035$ | $0.493 \pm 0.042$ | $0.734 \pm 0.025$ |
| multi lr multiplier 7 | $0.669 \pm 0.120$ | $\mathbf{0.710 \pm 0.113}$ | $0.852 \pm 0.052$ | $0.584 \pm 0.030$ | $0.459 \pm 0.051$ | $0.704 \pm 0.033$ |
| linear classifier | $0.724 \pm 0.040$ | $0.705 \pm 0.035$ | $\mathbf{0.855 \pm 0.021}$ | $0.621 \pm 0.084$ | $0.453 \pm 0.079$ | $0.720 \pm 0.072$ |
| linear probing | $0.608 \pm 0.037$ | $0.667 \pm 0.021$ | $0.791 \pm 0.011$ | $0.515 \pm 0.008$ | $0.522 \pm 0.017$ | $0.668 \pm 0.008$ |
| CBraMod finetuning method[*] | $\mathbf{0.735 \pm 0.045}$ | $0.707 \pm 0.069$ | $0.839 \pm 0.022$ | $0.619 \pm 0.035$ | $\mathbf{0.533 \pm 0.064}$ | $\mathbf{0.749 \pm 0.031}$ |

[*] Finetuning methods: (1) *no multi lr*: single LR for backbone and head (2) *multi lr multiplier X*: head LR = X × backbone LR (3) *linear classifier*: full finetuning with 1-layer head (4) *linear probing*: frozen backbone, trainable 1-layer head (5) *CBraMod finetuning method*: 3-layer MLP head, multi lr multiplier 5.

Interestingly, when applying the same finetuning methods to DIVER, different configurations yielded superior performance compared to what worked best for CBraMod. For instance, on Mumtaz2016, DIVER achieved its best performance with "linear probing" (ACC: 0.944), which performed poorly for CBraMod (ACC: 0.515). On TUEV, DIVER performed best with "multi lr multiplier 3" (ACC: 0.649), whereas CBraMod favored the "linear classifier" approach.

Crucially, when comparing both models under identical finetuning configurations (one-to-one comparison), DIVER demonstrates competitive performance across tasks. DIVER achieves superior performance on multiple configurations for Mumtaz2016 and TUEV, and shows strong results on MentalArithmetic. This head-to-head comparison under controlled conditions reveals that DIVER achieves overall state-of-the-art performance when evaluation methodology is held constant, even though CBraMod's reported in-paper results appear higher on some tasks.

These findings underscore that finetuning methodology is critical for evaluating EFMs, and optimal configurations can be model-dependent. Given this importance, we provide detailed specifications of the finetuning methods used for DIVER (Appendix B.8) to promote transparency and reproducibility in the EFM research community. We hope this contributes positively to establishing standardized evaluation protocols for electrophysiology foundation models.

## F  DATA DETAILS

### F.1  PRETRAINING DATASET DESCRIPTION

The following datasets were utilized for the pretraining of our DIVER models. The total pretraining time for the $DIVER_I$ dataset is 5,310 hours, and for the $DIVER_{IE}$ dataset, it is 59,613 hours.

- **AJILE12 (Anootated Joints in Long-term Electrocorticography)** (Peterson et al., 2022): An ECoG dataset from 12 epilepsy patients, recorded semi-continuously over 55 days. Signals were collected from $\geq$ 64 electrodes at 1 kHz sampling rate and paired with synchronized video-vased 3D human pose estimation and annotated wrist-movement events.

- **Self-collected iEEG dataset**: An intracranial EEG dataset from 25 drug-resistant epilepsy patients ($\sim$7 days, $\sim$168 h per subject) with long-term ECoG and sEEG recordings (mean $56.4 \pm 3.38$ channels, sampled at 2 kHz) during naturalistic hospital behaviors.

- **TUEG (Temple University Hospital EEG Corpus)** (Obeid & Picone, 2016): A large-scale clinical EEG dataset comprising 16,986 recording sessions from 10,874 subjects with heterogeneous diagnoses. EEG signals were recorded using 20–31+ channels, predominanty at sampling rate between 250–512 Hz, and are linked with de-identified clinical reports.

- **HBN-EEG (Healthy Brain Network)** (Shirazi et al., 2024): An developmental EEG dataset from 2,782 participants aged 5–21. Each participant underwent approximately 60 minutes of high-density (128-channel) EEG and eye-tracking recordings across six distinct tasks, including resting-state and movie watching.

- **NCHSDB (Nationwide Children's Hospital Sleep DataBank)** (Lee et al., 2022): An pediatric sleep EEG dataset of 3,673 patients. Each record includes 8–12 hours of EEG data (26–29 channels, sampled at 256–512 Hz) manually scored for sleep stages and events. While the dataset contains multimodal PSG signals (EOG, EMG, ECG, respiration, etc.), we used only the EEG channels.

- **PEERS (Penn Electrophysiology of Encoding and Retrieval Study)** (Kahana et al., 2023): An EEG dataset from 364 subjects who participated in multiple sessions of free recall, recognitionm and distractor tasks. EEG signals were recorded with 125 channels at 500 Hz sampling rate.

### F.2  FINETUNING DATASET DESCRIPTION

The following datasets were utilized for the downstream evaluation of our DIVER models, comprising a comprehensive set of benchmarks across both iEEG and EEG modalities. An overview of the dataset specifications and task definitions is provided in Table 3.

**Neuroprobe** Neuroprobe (Zahorodnii et al., 2025) is a large scale iEEG benchmarks with naturalistic labels during movie watching. 10 subjects watch 25 movies, age from 6 to 19. There are 3 types of evaluation in neuroprobe; single subject-single movie (WithinSession) (splits within the movies), single subject-different movie(CrossSession) (splits within subjects), different subject-different movie(CrossSubject). We evaluated the model in WithinSession. Additionally, Neuroprobe provides an option to subset subjects and trials. We used the LITE option (default configuration), which includes two movies per subject and a total of six subjects. [5] Detailed description of each task is provided below (adapted from (Zahorodnii et al., 2025):

1. **frame_brightness** *(visual)*: The mean brightness computed as the average HSV value over all pixels. Low (percentiles 0%-25%) vs High (75%-100%)

2. **global_flow** *(visual)*: A camera motion proxy. The maximal average dense optical flow vector magnitude. Same as above.

---

[5]PopT's performance on the same task differs between its original paper and its evaluation in the neuroprobe benchmark because neuroprobe implemented proper train/test splits across time so that no temporal leakage occurs between training and test sets, whereas the original PopT evaluation used random sampling that can lead to data contamination across temporal boundaries.

3. **local_flow** *(visual)*: A large displacement proxy. The maximal optical flow vector magnitude. Same as above.

4. **face_num** *(visual)*: The maximum number of faces per frame during the word. 0, or $\geq 1$.

5. **volume** *(auditory)*: Average root mean squared watts of the audio. Low (0%-25%) vs High (75%-100%).

6. **pitch** *(auditory)*: Average pitch of the audio. Same as above.

7. **delta_volume** *(auditory)*: The difference in average RMS of the 500 ms windows pre- and post-word onset. Same as above.

8. **speech** *(language)*: Whether any speech is present in the given time interval.

9. **onset** *(language)*: Whether a new sentence starts in the interval, or there is no speech at all.

10. **gpt2_surprisal** *(language)*: Negative-log transformed GPT-2 word probability (given preceding 20s of language context). Low (0%-25%) vs High (75%-100%).

11. **word_length** *(language)*: Word length (ms). Same as above.

12. **word_gap** *(language)*: Difference between previous word offset and current word onset (ms). Same as above.

13. **word_index** *(language)*: The word index in its context sentence. The first word in the sentence (0), or other (1).

14. **word_head_pos** *(language)*: The relative position (left/right) of the word's dependency tree head.

15. **word_part_speech** *(language)*: The word Universal Part-of-Speech (UPOS) tag. Verb (0), or other (1).

**EEG tasks** We evaluate our model on five publicly available EEG datasets spanning emotion recognition, motor imagery, mental workload tasks and mental disorder diagnosis and event type classification. We adopted the preprocessing procedure from CBraMod with minimal modifications; specifically, the resampling rate was adjusted to 500 Hz while all other steps remained consistent with the original pipeline. Detailed description of each dataset is provided below:

1. **FACED** (Chen et al., 2023): A large-scale EEG corpus for emotion recognition. It contains recordings from 123 subjects with 32-channel EEG while watching 28 emotion-eliciting video clips. Emotions are categorized into 9 discrete classes: amusement, inspiration, joy, tenderness, anger, fear, disgust, sadness, and neutral. We evaluated the model with the 9-class emotion classification task.

2. **PhysioNet-MI** (Goldberger et al., 2000; Schalk et al., 2004): An EEG dataset for motor imagery–based BCI tasks. It includes recordings from 109 subjects using a 64-channel 10–20 montage and contains four motor imagery classes: left fist, right fist, both fists, and both feet.

3. **MentalArithmetic** (Zyma et al., 2019): An EEG dataset for mental stress detection. It contains recordings from 36 subjects using 20 channels. We used 19 channels in total, excluding 1 reference channel. The dataset consists of recordings during mental arithmetic tasks under two conditions: with mental stress and without mental stress.

4. **Mumtaz2016** (Mumtaz, 2016): A clinical EEG dataset designed to distinguish major depressive disorder patients from healthy individuals. This dataset comprises 64 subjects (34 with MDD, 30 healthy controls), with signals acquired from 19 scalp locations following the standard 10-20 electrode placement system. We employed the resting-state conditions for binary MDD classification.

5. **TUEV** (Obeid & Picone, 2016): An EEG dataset for event type classification in clinical neurophysiology. This corpus provides annotated EEG segments categorized into six classes: spike and sharp wave (SPSW), generalized periodic epileptiform discharges (GPED), periodic lateralized epileptiform discharges (PLED), eye movement (EYEM), artifact (ARTF), and background (BCKG). We evaluated the model on this 6-class event classification task.

### F.3 QAQC AND PREPROCESSING

All data underwent quality assessment and control (QAQC) and preprocessing with a philosophy of minimal intervention to retain as much original signal information as possible. For QAQC, we normalized signals by dividing EEG by 100 μV and iEEG by 200 μV (the latter accounting for larger amplitudes in intracranial recordings). While Jiang et al. (2024) applied normalization without QAQC and Wang et al. (2024c) removed entire segments if even one timepoint exceeded 100 μV, we adopted a more conservative clipping approach to prevent data loss. We clipped amplitude values exceeding these normalization thresholds, only discarding electrodes when more than 3.33% of samples required clipping and removing whole segments when more than 50% of channels were compromised. This conservative strategy enabled us to preserve substantially more usable data: whereas CBRaMod's preprocessing yielded approximately 174.7k channel-hours of pretraining data on the same TUEG dataset (refer to Appendix Table 26), our QAQC pipeline retained 422k channel-hours, a 2.4× increase.

For preprocessing, we applied minimal filtering: a high-pass filter (0.5 Hz for private iEEG, 0.3 Hz for other datasets) to remove low-frequency drift, a 60 Hz notch filter for power line noise suppression, and no low-pass filtering to preserve high-frequency components. All datasets were resampled to 500 Hz and segmented into 30-second non-overlapping windows.

## G COMPARISON WITH EXISTING EEG/iEEG FOUNDATION MODELS

A direct comparison of training epochs for prior EEG/iEEG foundation models is misleading due to varying dataset sizes (Table 25). To enable a fair assessment, we introduce the "Scaled Epochs on Our Data" metric. This normalizes the total data processed during training (channel-hours × epochs) into an equivalent number of epochs on our dataset, allowing for a direct comparison of training epochs across all prior models and our own. For entries in Table 25 that use estimated values (marked with ∗), the corresponding estimation procedures are documented in Table 26.

Table 25: **Comparison of prior EEG/iEEG foundation models**

| Models | Modality | Model Size (Parameters) | Volume (Channel-hours) | Training Epochs | Scaled Epochs on Our Data[a] |
|---|---|---|---|---|---|
| BENDR (Kostas et al., 2021) | EEG | 155M[*] | N/A | 1 | N/A |
| BrainBERT (Wang et al., 2023) | SEEG | 43M[*] | 4.5k | 39[*] | 0.5 |
| Brant (Zhang et al., 2023) | SEEG | 505.68M | 281k | 32[*] | 25.6 |
| BIOT (Yang et al., 2023) | EEG | 3.3M | 312k | 100 | 88.6 |
| Neuro–GPT (Cui et al., 2024) | EEG | 90M[*] | 541k | 135 | 207.6 |
| LaBraM (Jiang et al., 2024) | EEG | 5.8M, 46M, 369M | 76.8–83.7k | 50 | 11.4 |
| EEGPT (Wang et al., 2024b) | EEG | 0.4M, 0.5M, 1.6M, 6.4M, 19M, 25M, 76M, 101M | 11.1k[*] | 200 | 6.8 |
| CBraMod (Wang et al., 2024c) | EEG | 0.1M, 0.4M, 0.8M, 1.2M, 1.5M, 2M, 3M, 4M | 175.7k[*] | 40 | 20 |
| Ours (DIVER$_I$) | ECoG+SEEG | 12.72M–1.83B | 352k | 64 | |
| Ours (DIVER$_{IE}$) | ECoG+SEEG+EEG | 13.03–812.85M | 1,662k | 1 | |

[a] This metric normalizes the total training compute across studies to represent the equivalent number of training compute on our dataset. It is calculated as: Scaled Epochs $= \dfrac{\text{Source Dataset (channel-hours)} \times \text{Source Epochs}}{\text{Our Dataset (channel-hours)}}$, where our iEEG dataset = 352,035 channel-hours.

[*] Values marked with an asterisk are our estimates, as they were not explicitly stated in the source paper; the estimation methods are summarized in Table 26.

Table 26: **Estimation of model and training specifications.** We detail the assumptions and calculations used to derive model size, training epochs, and data volume (channel-hours) for prior models. Bold values represent the final estimates derived from the reported configurations.

| Models | Parameters | Est. Value | Justification / Method |
|---|---|---|---|
| BENDR | Model Size | **155M** | **Assume** $d$=1536, $r$=3076, $L$=8. Decompose $P = P_{\text{conv}} + P_{\text{pos}} + P_{\text{in}} + L\,P_\ell$. Here $P_{\text{conv}} = (3\cdot20\cdot512{+}512) + 5(2\cdot512\cdot512{+}512) + 6(2\cdot512)$, $P_{\text{pos}} = 25\cdot(512/16)\cdot512 + 512$, $P_{\text{in}} = 512\cdot1536 + 1536$, and $P_\ell \approx 4d^2{+}2dr{+}r{+}9d$. Numerically $\approx$ **155M**. |
| BrainBERT | Model Size | **43M** | **Assume** $d$=768, $r$=4$d$=3072, $L$=6. Per layer $P_\ell \approx 4d^2{+}2dr{+}r{+}9d \approx 7.08$M. Transformer total $LP_\ell \approx 42.5$M. Add 2-layer head 768→768→40: $P_{\text{head}} \approx 0.6$M. Hence $P \approx 42.5{+}0.6 \approx$ **43M**. |
|  | Training Epochs | **39** | Total hours $H$=4,551, segment $\tau$=5s. $N_{\text{samp}} = \frac{H\cdot3600}{\tau} = \frac{4551\cdot3600}{5} = 3{,}276{,}720$. Steps/epoch $\approx N_{\text{samp}}/256 \approx 12{,}800$. With $U$=500,000 updates: epochs $\approx U/12{,}800 \approx$ **39**. |
| Brant | Training Epochs | **32** | Use epochs $= \frac{U\cdot(B\cdot A)}{N_{\text{samp}}}$. Reported $U$=750,000, $B$=16, $A$=4 $\Rightarrow$ 48M sample-passes. Dataset size $N_{\text{samp}} \approx 1.5$M $\Rightarrow$ epochs $\approx 48/1.5 \approx$ **32**. |
| Neuro–GPT | Model Size | **90M** | **Assume** GPT-2-like decoder with $d$=1024, $r$=4096, $L$=6. Per layer $P_\ell \approx 12.6$M $\Rightarrow P_{\text{GPT}} = LP_\ell \approx 75.6$M. Adding EEG encoder and a linear projector ($P_0$) yields $P = P_0 + P_{\text{GPT}} \approx$ **90M** (projector $\sim$ 1.1M; remainder in encoder). |
| EEGPT | Channel-hours | **11.1k** | Apply ch-hr $= \frac{(\#\text{trials})\cdot(\#\text{ch})\cdot(\text{s})}{3600}$ per dataset and sum: PhysioMI $\approx$ 107, HGD $\approx$ 1,991, TSU $\approx$ 597, SEED $\approx$ 116, M3CV $\approx$ 8,267 $\Rightarrow$ total $\approx$ **11,078** ch-hr. |
| CBraMod | Channel-hours | **175.7k** | Common channels $c$=19, non-overlapping segment $\tau$=30s, kept segments $N$=1,109,545: ch-hr $= \frac{c\cdot N\cdot\tau}{3600} = \frac{19\cdot1,109,545\cdot30}{3600} \approx$ **175.7k**. |

*Conventions.* Hidden size $d$, FFN expansion $r$, layers $L$, per-layer parameters $P_\ell$, total parameters $P$, segment length $\tau$ (seconds), minibatch $B$, gradient accumulation $A$, updates $U$, number of samples $N_{\text{samp}}$. Transformer block (per layer): $P_\ell = \underbrace{4d^2}_{\text{Q,K,V,Out}} + \underbrace{2dr + r +}_{\text{FFN}} \underbrace{9d}_{2\times\text{LN+biases}} \approx 4d^2 + 2dr + r + 9d$. Total params: $P = P_0 + L\,P_\ell$ (non-Transformer parts $P_0$ separated when needed). Epochs: epochs $= \frac{U\cdot(B\cdot A)}{N_{\text{samp}}}$, with $N_{\text{samp}} = \frac{(\text{hours}\cdot3600)}{\tau}$. Channel-hours: ch-hr $= \sum \frac{(\#\text{trials})\cdot(\#\text{channels})\cdot(\text{seconds})}{3600}$.

