# OpenReview forum: "DIVER-1 : Deep Integration of Vast Electrophysiological Recordings at Scale"
_ICLR.cc/2026/Conference — Submitted to ICLR 2026_

### Official Review · Reviewer_6kz9 · 2025-10-27

**Soundness:** 3
**Presentation:** 2
**Contribution:** 2
**Rating:** 4
**Confidence:** 3

**Summary:**

This paper presents DIVER-1, a large-scale family of electrophysiology foundation models (EFMs) designed for EEG and iEEG data. The authors conduct the first systematic scaling law analysis in this domain, extending scaling dimensions beyond model, data, and compute to include epoch and subject diversity. DIVER-1 models (up to 1.82B parameters) are pretrained on 59k hours of neural recordings and evaluated on the NeuroProbe (iEEG) and FACED (EEG) benchmarks. The results show data-constrained scaling behaviors distinct from NLP/Vision domains, with smaller models trained longer outperforming larger models trained briefly. The authors propose practical guidelines for compute-efficient EFM scaling.

**Strengths:**

- The paper conducts the most extensive scaling study to date for electrophysiological models, spanning model/data/compute/subject axes and revealing data-constrained scaling laws that challenge prior assumptions borrowed from NLP/Vision.
- The authors rigorously fit their empirical trends to data-constrained scaling law formulations, providing quantitative evidence that electrophysiology follows different scaling constants and exponents compared to language models.
- The paper introduces several domain-specific innovations, any-variate attention, STCPE, and multi-domain reconstructio, which together improve multimodal integration and robustness to electrode heterogeneity.

**Weaknesses:**

- Despite impressive pretraining coverage, the downstream evaluation is narrow. For EEG, only FACED (emotion recognition) is used; other essential EEG-based BCI or cognitive decoding benchmarks (e.g., motor imagery, sleep, attention) are missing, which weakens generalization claims.
- Most validation relies on binary NeuroProbe classification tasks, some of which the authors acknowledge as trivial (e.g., frame brightness, flow angle). These tasks may not sufficiently challenge the models to demonstrate meaningful representation quality, limiting the impact of reported gains.
- The model integrates multiple complex modules (any-variate attention, register tokens, multi-domain reconstruction, STCPE), yet no clear ablation quantifies each module's contribution. Likewise, the effect of loss weighting parameters $\lambda_1, \lambda_2, \lambda_3$ is unexplored.
- DIVER-1 underperforms existing models (e.g., CBraMod) on FACED, even with much larger data and model size. The explanations ("insufficient epochs", "dataset mismatch") are speculative and would benefit from diagnostic analysis.

**Questions:**

- The notation for $\mathcal{L}_{\text{total}}$ (pretraining loss) and its weighting $\lambda_i$ is given, but no ablation is provided. How sensitive is model performance to the choice of $\lambda_1, \lambda_2, \lambda_3$? Could you include an ablation or rationale for these settings (either in-text or appendix)?
- Given the relatively weak EEG results, have you analyzed potential causes beyond epoch count or data mismatc such as QAQC differences, patch length, or fine-tuning hyperparameters? It would be valuable to report comparisons across multiple EEG datasets.
- Will the pretrained weights, scaling analysis code, and data preprocessing pipeline be made public to enable replication and extension by the community?

---

> ### Author Response · Authors · 2025-11-25
>
> Dear Reviewer 6kz9, thank you for your constructive feedback. We made updates to the PDF, highlighted in blue. Below we address each of your questions below :
>
> ## W1. and W2.  Limited downstream evaluations
> > “Despite impressive pretraining coverage, the downstream evaluation is narrow”
>
> > “NeuroProbe classification tasks, … may not sufficiently challenge the models to demonstrate meaningful representation quality, ”
>
> We appreciate this feedback and have significantly strengthened our downstream evaluation, particularly for EEG tasks.
>
> * **Expanded EEG benchmark coverage**: Beyond the initial FACED emotion recognition task, we have incorporated additional established EEG benchmarks spanning diverse cognitive and motor functions. These now include PhysioNet-MI for motor imagery classification and MentalArithmetic for cognitive load assessment. **DIVER-1 achieves state-of-the-art performance across all three EEG benchmarks (line 423, Figure 2)**, demonstrating robust generalization beyond emotion recognition to motor control and cognitive decoding domains.
>
> * **Clear evidence of downstream scaling benefits in EEG**: We have also added comprehensive scaling analyses specifically for EEG downstream tasks (Figure 2 (m),(n),(o)). These experiments reveal a critical distinction: while some NeuroProbe tasks show limited scaling behavior, EEG downstream performance exhibits consistent, predictable improvements with both increased pretraining data and model capacity. This demonstrates that pretraining delivers substantial, measurable value on challenging, clinically-relevant tasks rather than only on simpler binary classification problems.
>
> We are actively incorporating additional EEG and iEEG tasks to further strengthen these generalization claims in the revised manuscript.
>
>
> ## W3. Performance contribution of model components
> > “... no clear ablation quantifies each module's contribution”
>
> We have conducted comprehensive ablation studies to quantify the contribution of DIVER's core components. Table 17 presents ablations across multiple DIVER modules for iEEG tasks, while Table 18 extends this analysis to EEG tasks. These experiments reveal that component effectiveness varies by both modality (iEEG vs. EEG) and task type. Notably, while certain modules show limited impact on iEEG performance, **the same components consistently contribute meaningful improvements in EEG tasks**. (see below) :
>
> ### FACED (9-class) Ablation
>
> | | ACC | kappa | F1 |
> |---|---|---|---|
> | w.o. RoPE | 0.408 ± 0.025 | 0.333 ± 0.027 | 0.414 ± 0.024 |
> | w.o anyV attention | 0.414 ± 0.012 | 0.339 ± 0.014 | 0.417 ± 0.012 |
> | w.o RoPE and anyV attention | 0.446 ± 0.007 | 0.376 ± 0.007 | 0.450 ± 0.005 |
> | w.o. STCPE | 0.463 ± 0.024 | 0.395 ± 0.027 | 0.471 ± 0.026 |
> | w.o Channel modality + subtype emb. | 0.474 ± 0.018 | 0.406 ± 0.021 | 0.482 ± 0.019 |
> | w.o Channel 3d position emb. | 0.481 ± 0.016 | 0.415 ± 0.018 | 0.487 ± 0.016 |
> | w.o Spectral feature emb. | 0.454 ± 0.015 | 0.386 ± 0.016 | 0.462 ± 0.013 |
> | w.o Multi-domain reconstruction (only raw) | 0.435 ± 0.008 | 0.364 ± 0.009 | 0.437 ± 0.006 |
> | **DIVER_Tiny/E/1s** | **0.491 ± 0.023** | **0.428 ± 0.025** | **0.502 ± 0.023** |
>
> ### PhysioNet-MI (4-class) Ablation
>
> | | ACC | kappa | F1 |
> |---|---|---|---|
> | w.o. RoPE | 0.614 ± 0.005 | 0.485 ± 0.006 | 0.615 ± 0.005 |
> | w.o anyV attention | 0.611 ± 0.003 | 0.481 ± 0.004 | 0.612 ± 0.004 |
> | w.o RoPE and anyV attention | 0.591 ± 0.005 | 0.454 ± 0.006 | 0.593 ± 0.005 |
> | w.o. STCPE | 0.626 ± 0.006 | 0.502 ± 0.008 | 0.627 ± 0.006 |
> | w.o Channel modality + subtype emb. | 0.629 ± 0.006 | 0.505 ± 0.007 | **0.632 ± 0.005** |
> | w.o Channel 3d position emb. | **0.629 ± 0.008** | **0.506 ± 0.010** | 0.631 ± 0.007 |
> | w.o Spectral feature emb. | 0.626 ± 0.005 | 0.501 ± 0.006 | 0.627 ± 0.004 |
> | w.o Multi-domain reconstruction (only raw) | 0.614 ± 0.005 | 0.485 ± 0.006 | 0.616 ± 0.004 |
> | **DIVER_Tiny/E/1s** | 0.628 ± 0.005 | 0.504 ± 0.007 | 0.630 ± 0.005 |
>
>
> We also **compared DIVER to prior models under a matched pretraining dataset condition**, training a DIVER-tiny variant solely on the BrainTreeBank (BTB) dataset used by PopT and BrainBERT (Appendix Table 19 (also included below for reference)). Even with identical pretraining data, DIVER-tiny outperforms both baselines, demonstrating that its architectural design—not dataset scale—drives the performance gains.

---

> ### Author Response · Authors · 2025-11-25
>
> ### Same pretraining dataset ablation
>
> | Model                              | speech              | onset              |
> |-----------------------------------|----------------------|---------------------|
> | **BrainBERT (frozen)** (Zahorodnii et al., 2025) | 0.611 ± 0.022      | 0.757 ± 0.027      |
> | **BrainBERT (ours)**              | 0.575 ± 0.018        | 0.659 ± 0.026      |
> | **PopT (ours)**                   | 0.702 ± 0.029        | 0.780 ± 0.025      |
> | **PopT** (Zahorodnii et al., 2025) | 0.677 ± 0.044        | 0.689 ± 0.050      |
> | **DIVER Tiny/I/0.1s (frozen)** | **0.770 ± 0.028** | **0.859 ± 0.018** |
>
>
> > “... effect of loss weighting parameters … unexplored”
>
> Regarding the multi-domain reconstruction loss weighting parameters (λ), we provide heuristic justification for our design choices in line 270. We have conducted ablation studies removing each component of the multi-domain reconstruction loss for EEG (Table 18, line 1978). We acknowledge that more systematic ablations of λ values would strengthen our analysis and will include this in the revised manuscript.
>
> ## W4. Underperforming existing models
> > “...underperforms existing models (e.g., CBraMod) on FACED, even with much larger data and model size.”
>
> **Achieved state-of-the-art EEG performance**: We thank the reviewer for raising this concern. After diagnosing the issue, we found that the initial underperformance on FACED did not arise from limitations in the DIVER-1 encoder nor from insufficient epochs or dataset mismatch. Instead, **the discrepancy originated from a mismatch in downstream evaluation protocol**: our initial submission used a linear classification head, whereas existing EEG foundation models (CBraMod) rely on 3 layer MLP heads.
>
> **Once we aligned our evaluation with this standard setting (i.e., adopting the same 3-layer MLP architecture used in CBraMod), DIVER-1 now achieves state-of-the-art performance across all EEG benchmarks**, including FACED (emotion recognition), PhysioNet-MI (motor imagery), and MentalArithmetic (cognitive workload). This adjustment does not introduce any special advantage; it simply ensures a fair, apples-to-apples comparison with prior work. Comprehensive results are provided in Figure 2 and Appendix D.1. (Results below)
>
> |Model| |FACED| | |PhysioNet| | |MentalArith| |
> |----|----|----|----|----|----|----|----|----|----|
> ||ACC±SD|Kappa±SD|F1±SD|ACC±SD|Kappa±SD|F1±SD|ACC±SD|auc_pr±SD|AUROC±SD|
> |LaBraM|0.5273±0.0107|0.4698±0.0188|0.5288±0.0102|0.6173±0.0122|0.4912±0.0192|0.6177±0.0141|0.6909±0.0125|0.5999±0.0155|0.7721±0.0093|
> |CBraMod|0.5509±0.0089|0.5041±0.0122|0.5618±0.0093|0.6417±0.0091|0.5222±0.0169|0.6427±0.0100|0.7256±0.0132|0.6267±0.0099|0.7905±0.0073|
> |**DIVER(ours)**|**0.6014±0.0075**|**0.5501±0.0086**|**0.6066±0.0088**|**0.6756±0.0032**|**0.5674±0.0043**|**0.6777±0.0036**|**0.7271±0.0176**|**0.6755±0.0460**|**0.8140±0.0259**|
>
> >”...The explanations ("insufficient epochs", "dataset mismatch") are speculative and would benefit from diagnostic analysis.”
>
> **Comprehensive diagnostic analysis:** To identify the source of the initial EEG underperformance and rule out speculative explanations, we conducted extensive controlled ablations examining: (1) pretraining duration effects on FACED performance (Appendix C.4), (2) the influence of classifier architecture via systematic head-depth ablations (1–5 layer MLPs; Table 22 and Appendix E.2), and (3) the impact of pretraining dataset composition (Appendix D.4).
>
> -------
>
> ## Q1. Choice of lambda values and ablation studies
> >"...How sensitive is model performance to the choice of λ? Could you include an ablation or rationale…"
>
> We have written our justification behind the lambda value choices in line 270. We are planning on adding ablation results on this soon.
>
> ## Q2. Reason for relatively weak EEG results
> >”...Given the relatively weak EEG results, have you analyzed potential causes beyond epoch count or data mismatch such as QAQC differences, patch length, or fine-tuning hyperparameters? It would be valuable to report comparisons across multiple EEG datasets.”
>
> As we detail in our response to “W4. Underperforming existing models”, we have identified the reason for the weak EEG performance and analyzed the reason deeply in Appendix E.2.
>
> We also, as suggested, did this comparison across two EEG datasets, FACED, and Physionet-MI (see Appendix, D.1, E.2).
> ## Q3. Public code release
> >”...Will the pretrained weights, scaling analysis code, and data preprocessing pipeline be made public to enable replication and extension by the community?”
>
> Yes. During rebuttal, or at least before camera ready deadline we will release fine tuning code and weights of the DIVER models. We will also release all the intermediate weights so others can build on top of our work that already used lots of compute.

---

### Official Review · Reviewer_UXXE · 2025-10-28

**Soundness:** 2
**Presentation:** 2
**Contribution:** 2
**Rating:** 4
**Confidence:** 4

**Summary:**

This paper presents DIVER-1, a family of multimodal foundational models for both EEG and iEEG, training by multitask SSL objective across both modalities. The authors demonstrate that DIVER-1 can be pretrained using a large collection of diverse, across modality data with distinct electrode configurations, yielding competitive downstream performance on new iEEG data. The authors also conduct extensive scaling law analysis on the proposed method, showing that it follows data constrained scaling law of language models, and this finding is valuable for the community. However, the evaluated iEEG task is relatively simple and the finetuning performance on EEG data is consistently worse than the baseline. This suggests that more investigation is likely needed on the proposed model. Hence I don’t recommend acceptance in my rating.

**Strengths:**

- DIVER-1 is a multimodal model that can take both EEG and iEEG data which have distinct electrode configurations. Being able to incorporate heterogeneous data is essential for a foundational model of neural data.
- Several architectural innovations are proposed in the DIVER-1 model, including spatial-temporal conditional positional embedding, spatial temporal register tokens and multitask output for SSL training.
- A scaling analysis revealing that model follows existing data constrained scaling laws.

**Weaknesses:**

- One of the major conclusions is to advocate smaller model training for more epochs, compared to larger model training for limited amount of epochs. There seems to be an assumption on a limited compute budget. But I don’t understand why that should be a pretext since the goal is to train large-scale foundational models, which is known to be compute-intensive.
- The evaluations on the new datasets are limited. There is one iEEG dataset whose tasks are all binary classification, which is acknowledged by the authors that it can be too simplistic. And another EEG dataset where baseline methods are consistently outperforming the proposed model. Hence they are not strong evidence to support the authors' claim.

**Questions:**

1. The authors mention that larger models initially underperform smaller ones (line 332 page 7). But In Figure 2 (d), except for the biggest one (XXL), increasing parameter sizes did result in better test performance at epoch 2 (e.g. XL one is the best of all). Can authors comment on the poor performance of XXL?
2. In the same plot (Figure 2 (d)), DIVER XL has promising testing results with respect to number of epochs, and it can already outperform smaller model (DIVER Tiny) at epochs 4. Isn’t that a sign that larger models can perform well at shorter training time? And why didn’t that model train for longer epochs?
3. Compared to the plot in the original data-contrained scaling law paper, the empirical isoLoss contour in Figure 2 (j) is much less smooth and irregular. Could authors provide insights on what could lead to the difference?
4. Could authors elaborate on how to understand Figure 2 (k)? For example, it seems to assume LaBram was trained from 8-10 epochs and BrainBert is less than 1 epoch, whereas it was 50 total epochs listed in LaBram paper and 500k steps in BrainBert paper. Also, it looks like DIVER-1 model (the white star) is not on the compute-optimal frontier either?
5. Could authors provide details on how STCPE operates: from Figure 1 there seems to be multiple sliding windows involved in the computation, but there is no mentioning of that in section 2.2.
6. Two patch sizes are adopted on the input time series: 1s and 100ms. Will the use of a small patch size negatively affect the capturing of lower frequency bands, which is used by existing EEG model (e.g. Brant model cited in the paper)
7. It is surprising to me that linear probing is better than full finetuning on the iEEG task. Are there any other finetuning strategies tried for that dataset, for example, freezing only a subset of all the parameters?
8. In the downstream iEEG dataset, there are multiple tasks with near chance performance and the author argues that it is because they are non-informative. Could authors provide more details on why that is the case?
9. Please provide evidence on the several factors mentioned in the EEG FACED task, which leads to poor performance of DIVER-1 on it. Have authors tried other EEG dataset for evaluation?

---

> ### Author Response · Authors · 2025-11-26
>
> ## W1. Assumption of limited compute budget
> > “..There seems to be an assumption on a limited compute budget. But I don’t understand why that should be a pretext since the goal is to train large-scale foundational models..”
>
> > “ I don’t understand why that should be a pretext since the goal is to train large-scale foundational models, which is known to be compute-intensive.”
>
> ​​
> Yes — **that is exactly the point: we assume a limited compute budget**. While training large-scale foundation models is inherently compute-intensive, even well-resourced groups **must balance how they invest compute: larger models help, but more epochs also help**, and the **two compete for the same resources**.
>
>
> To make this clearer, we **added explicit clarification in lines 374-380 and 402-412** of the revised manuscript. In simple terms: electrophysiology is a data-limited domain, so models must train over the same dataset many times. When both data and compute are limited, there is a **natural trade-off between model size and training duration**. Our analysis finds the efficient scaling frontier, where given the data size and compute, one can find the model size and training epochs that achieve the lowest loss.
>
> **Case in point**: for iEEG, the DIVER-XXL model would need more than 100 epochs to reach the compute-optimal region, as observable by looking at "data-constrained efficient scaling frontier" on Fig. 2(p) (right). **Running XXL to this (optimal) depth would require** >1e21 FLOPs (≈ 11.6 PF-days), which translates to running **128 A100s for around 8.75 days (3.06 A100-years)**(128GPU * (126min/epoch) *100epoch / (60min*24hour))—far beyond what ordinary labs can do. In contrast, smaller and mid-sized models can reach their optimal regime within tractable compute budgets, and therefore outperform XXL under realistic constraints.
>
> Our goal is **not to argue against large models, but to provide principled guidance on how to allocate limited compute** when both data and epochs are constrained in electrophysiology settings.
>
>
> ## W2. EEG not being SOTA and finetuning dataset limitations
>
> > "The evaluations on the new datasets are limited."
>
> >  "... iEEG dataset ... all binary classification, ... too simplistic"
>
> We appreciate this concern and have significantly expanded our downstream evaluation in the revision. For EEG, we now report results on **three well-established benchmarks**—FACED (emotion), PhysioNet-MI (motor imagery), and MentalArithmetic (cognitive load)—covering a broader range of cognitive and motor decoding tasks (line 421, Fig. 2). For iEEG, we extended evaluation to **long-context iEEG task, MAYO seizure detection (6 s) (line 895-900)**.
> To tackle a more difficult task, we evaluated the neuroprobe multilabel setting (which is included in the new version of the released neuroprobe code but not described in the paper, lines 892–894), and it still outperforms the linear STFT–Laplacian baseline (Figure 2r).
>
> > "... EEG ... baseline methods are consistently outperforming the proposed model.
>
> Thank you for raising this important concern. After re-examining our EEG pipeline, we discovered that the discrepancy stemmed from an **evaluation mismatch**, rather than any limitation of DIVER-1. Specifically, in our initial submission we finetuned using a **linear classification head** for EEG tasks, whereas prior EEG foundation models—most notably **CBraMod**—use **3-layer MLP heads**. This mismatch caused the initially reported DIVER-1 scores to appear lower than they should have been, and now DIVER-1 achieves state-of-the-art performance.
>
> ### **Correcting the evaluation mismatch**
>
> After aligning our evaluation protocol to match the standard used in prior work—i.e., using the **same depth MLP head** as CBraMod—we find that **DIVER-1 achieves state-of-the-art results across all three EEG benchmarks**. This correction ensures an apples-to-apples comparison and accurately reflects the strength of DIVER-1’s representations.
>
> **Updated SOTA EEG results (Figure 2, Appendix D.1)**
>
> |Model| |FACED| | |PhysioNet| | |MentalArith| |
> |----|----|----|----|----|----|----|----|----|----|
> ||ACC±SD|Kappa±SD|F1±SD|ACC±SD|Kappa±SD|F1±SD|ACC±SD|auc_pr±SD|AUROC±SD|
> |LaBraM|0.5273±0.0107|0.4698±0.0188|0.5288±0.0102|0.6173±0.0122|0.4912±0.0192|0.6177±0.0141|0.6909±0.0125|0.5999±0.0155|0.7721±0.0093|
> |CBraMod|0.5509±0.0089|0.5041±0.0122|0.5618±0.0093|0.6417±0.0091|0.5222±0.0169|0.6427±0.0100|0.7256±0.0132|0.6267±0.0099|0.7905±0.0073|
> |DIVER (ours)|0.6014±0.0075|0.5501±0.0086|0.6066±0.0088|0.6756±0.0032|0.5674±0.0043|0.6777±0.0036|0.7271±0.0176|0.6755±0.0460|0.8140±0.0259|

---

> ### Author Response · Authors · 2025-11-26
>
> ### **Ablation: confirming the importance of head depth**
>
> To ensure that the discrepancy truly arose from the evaluation head (and not architecture-specific tuning), we conducted a structured ablation over head depth (Appendix Table 22). This analysis shows that 3–4 layer MLP heads are optimal for EEG tasks, exactly matching the configuration used by CBraMod.
>
> |HeadDepth| |FACED| | |PhysioNet-MI| |
> |----|----|----|----|----|----|----|
> ||ACC±SD|Kappa±SD|F1±SD|ACC±SD|Kappa±SD|F1±SD|
> |Linear|0.523±0.018|0.461±0.020|0.521±0.018|0.653±0.013|0.538±0.017|0.654±0.013|
> |2-Layer|0.584±0.005|0.530±0.005|0.584±0.005|0.674±0.003|0.565±0.004|0.676±0.003|
> |3-Layer|0.601±0.008|0.550±0.009|0.607±0.009|0.676±0.003|0.567±0.004|0.678±0.004|
> |4-Layer|0.603±0.007|0.552±0.008|0.609±0.007|0.672±0.008|0.573±0.011|0.674±0.008|
> |5-Layer|0.594±0.021|0.543±0.023|0.601±0.019|0.660±0.007|0.547±0.009|0.662±0.006|
>
> ### **Scaling behavior reinforces these findings**
>
> Our expanded EEG scaling experiments (Fig. 2m–o) further show that EEG downstream accuracy improves consistently with:
> * increased model size,
> * increased pretraining data, and
> * increased training repetitions,
>
> unlike some NeuroProbe tasks which exhibit shallow scaling behavior. This is likely due to the EEG task being more challenging, where larger models are more beneficial.
>
> ----
> ## Q1.
>
> >“…In Figure 2 (d), except for the biggest one (XXL), increasing parameter sizes did result in better test performance at epoch 2…”
>
> Thank you for this helpful observation. We have expanded and clarified this point substantially in the revision **(see lines 366–370, 374–380, and 402–412)**. In particular, **lines 366–370**, added in direct response to your comment, now explain this behavior explicitly.
> As described there, increasing the number of epochs improves performance across all model sizes, and larger models do tend to achieve lower loss given the same epochs. However, **excessively large models** (e.g., XXL in Fig. 2(d) (iEEG) and Large in Fig. 2(h)(EEG)) underperform at very small epoch counts and **only overtake smaller models after many more epochs**—a hallmark pattern predicted by the data-constrained scaling law, where repeated passes effectively increase the usable dataset size $D'$ needed for high-capacity models to reach their regime of advantage.
>
> This is precisely why DIVER-XXL appears worse at 2–8 epochs but eventually becomes competitive; the model simply requires more repetitions to be competitive.
>
> ## Q2.
> >“...Isn’t that a sign that larger models can perform well at shorter training time?”
>
> Thank you for this insightful observation. You are correct that DIVER-XL surpasses DIVER-Tiny by epoch 4, which indeed shows that larger models can achieve stronger performance even with relatively few epochs. However, as we clarify in our response to W1 and in the revised text, the **key consideration is compute, not just epoch count**. Training DIVER-XL for 4 epochs requires substantially more FLOPs than training smaller models for many more epochs, and **under a fixed compute budget the smaller models achieve lower loss**.
>
> Concretely, in the empirical IsoLoss contour plot in Fig. 2(p) (left), together with the newly added iso-compute curves, we observe that the loss obtained by training the 813M-parameter XL model for 4 epochs can be roughly matched by training a 51M-parameter Small model for ~8 epochs—roughly **one-tenth of the compute**. Thus, while larger models can perform well at short wall-clock times, they are not compute-efficient in the data-constrained regime, and smaller models trained longer achieve better pretext performance for the same compute.
>
>
> > “ …why didn’t that model train for longer epochs?”
>
> As discussed in our response to W1, scaling compute to the point where the largest models become compute-optimal is far beyond what is feasible for typical labs. For example, pushing DIVER-XL or DIVER-XXL to their optimal region of the scaling frontier would **require running 128 A100 GPUs for ~8 days (multiple A100-years)**, which is not practical even in well-resourced academic environments. This computational constraint—not performance saturation—sets the effective limit on how long the largest models can be trained.

---

> ### Author Response · Authors · 2025-11-26
>
> ## Q3.
> >“…the empirical isoLoss contour in Figure 2 (j) is much less smooth and irregular. Could authors provide insights on what could lead to the difference…”
>
> Yes, we acknowledge that the empirical isoloss contour of our paper is much less smooth and irregular compared to the original data-constrained scaling paper. We believe such differences are due to the **smaller number of model configurations tested, which led to sparser plots**. Despite the lack of samples, our empirical data aligns well with the predicted data-constrained scaling law (Figure 2 (p), Appendix Figure 4), and in fact as shown in Appendix Table 7, the $R^2$ value of the fit is high (0.75 for 0.1s patch model, and 0.78 for 1s patch model).
>
>
> ## Q4.
> >“...Could authors elaborate on how to understand Figure 2 (k)?”
>
> Thank you for pointing this out — we realized the explanation was insufficient, and we have now added a clearer description in **line 405** (with detailed estimation steps in **Appendix G, Table 25**).
>
> In Figure 2(k) (now (p)), the epoch values for other models (LaBraM, BrainBERT, etc.) represent “scaled epochs on our data” as reported in Table 25. Because each model was trained on a different dataset and with different amounts of data per epoch, we converted their reported training durations into a **common equivalent scale** : the number of epochs they *would have trained* if each of their epochs contained the same number of unique tokens as one DIVER epoch. This normalization allows us to **place heterogeneous models within the same compute–epoch coordinate system** for comparison.
>
> We acknowledge that this conversion involves approximations due to differences in architecture and data composition, but it provides a useful and interpretable way to understand where existing models approximately fall relative to the **compute-optimal scaling curve**.
>
>
> > “ it looks like DIVER-1 model (the white star) is not on the compute-optimal frontier either?”
>
> Thank you for this important observation. As clarified in the newly added discussion in **lines 409–412**, our original presentation may have been misleading. While training **at** the compute-optimal frontier is ideal for understanding efficiency trade-offs, it is not necessarily the point that yields the **best possible performance** for a *given* model size. In practice, if the aim is to push a smaller model to its full potential, training for **additional epochs beyond the compute-optimal point will continue to improve its loss, even though it is no longer compute-optimal**. This is why the DIVER-1 checkpoint used for downstream evaluation lies slightly beyond the strict compute-optimal frontier.
>
> We have also removed the star in the updated figure, as it was not used and caused confusion.
>
> ## Q5.
> >“…Could authors provide details on how STCPE operates…”
>
> Thank you for pointing out this insufficient explanation. We have now added a detailed description of STCPE in the updated Section 2.2 lines 197 to 224.
>
> Briefly, STCPE uses sliding temporal windows (width w, stride 1) over the patch-encoded sequence. For each window centered at position t′, a MOIRAI block processes the spatiotemporal slice and produces outputs for each relative temporal offset. The final STCPE embedding at position t aggregates contributions from all overlapping windows, maintaining both temporal translation equivariance and channel permutation equivariance. Complete details are available in the revised Section 2.2.

---

> ### Author Response · Authors · 2025-11-26
>
> ## Q6.
> >“…Will the use of a small patch size negatively affect the capturing of lower frequency bands…”
> Thank you for this thoughtful question.
>
> **Regarding the ability to capture low-frequency content:**
>
> A 1-second patch at typical EEG sampling rates (200–500 Hz) contains more than enough temporal resolution to represent frequencies down to 3 Hz and above (1/1s * 3 cycles). Moreover, **because the transformer integrates sequences of patches**, the model **can reconstruct lower-frequency structure across multiple adjacent patches**. In practice, the **0.1 s patch model is trained on up to 3-second input windows**, allowing it to represent **frequencies as low as ~1 Hz** (three cycles within 3 seconds).
>
> In fact, as shown in **Appendix Table 15, 0.1 s patches perform substantially better on NeuroProbe than 1s models**, since they provide 10 temporal tokens per 1-second input—whereas 1-second patches would provide no temporal structure for the model to attend over.
>
> > “...small patch size negatively affect the capturing of lower frequency bands … (e.g. Brant model cited in the paper)”
>
> While Brant uses a long 6-second window, **most recent EEG/iEEG foundation models use substantially shorter patches**. For example, CBraMod and LaBraM (EEG) both adopt 1-second patches, and PopT and BrainBERT (iEEG) use sub-100 ms patches. This reflects the fact that many EEG downstream tasks operate on slow time scales (≈1 s), and many iEEG naturalistic tasks operate on <1 s windows. Brant’s 6-second window is primarily motivated by (i) seizure-detection applications that require long receptive fields, and (ii) substantially reducing compute by lowering the number of tokens (a 6-second patch yields roughly 60× fewer tokens than a 0.1-second patch).
>
> ## Q7.
> >“…Are there any other finetuning strategies tried for that dataset…”
>
> Thank you for raising this interesting point. For EEG, as outlined in our response to W2, we also evaluated **MLP heads** as an alternative finetuning strategy for EEG. We agree that exploring additional parameter-efficient approaches, such as **LoRA (Low-Rank Adaptation)**, is a promising direction for future work and may further enhance finetuning efficiency in electrophysiology models.
>
> ## Q8.
> > “In the downstream iEEG dataset, there are multiple tasks with near chance performance ... Could authors provide more details on why that is the case?”
>
> Thank you for seeking clarification on this point. In the updated iEEG performance radial plot (Figure 2(q)), the tasks where DIVER achieves near-chance performance—specifically FrameBrightness and FaceNum—are primarily visual tasks requiring processing of visual stimulus features. However, the neuropixels probes used in our dataset have very limited coverage of visual cortical areas, with no electrodes in lower-level visual regions. Without neural recordings from brain regions that process visual information, these tasks are fundamentally non-informative for any model.
>
> Importantly, this is not a limitation specific to DIVER. As shown in the radial plot, baseline models also exhibit chance-level performance on these visual tasks, with the best baseline reaching only 53% for FrameBrightness and 52% for FaceNum (both from Linear classifiers)—essentially random chance level. This confirms that the limitation stems from the absence of relevant neural signals rather than model-specific deficiency. In contrast, DIVER achieves above-chance performance on all tasks where the recorded brain regions are functionally relevant.
>
> ## Q9.
> >“...Please provide evidence on the several factors mentioned in the EEG FACED task, which leads to poor performance of DIVER-1 on it…”
>
> Thank you for this important question. We have conducted extensive investigations into DIVER-1's initially lower performance on the FACED task and **identified the classifier architecture as the primary issue**, and added detailed analysis in Appendix E.2 (Table 22).
> In our initial experiments, we used a linear classifier head for FACED. However, upon adopting a 3-layer MLP classifier head—consistent with the finetuning methods of CBraMod–the current state-of-the-art EEG foundation model—we observed substantial performance improvements and achieved SOTA results, as discussed in our response to W2.
>
>
> > “ Have authors tried other EEG dataset for evaluation?”
>
> We have expanded our experiments to **include PhysioNet-MI and MentalArithmetic**. After addressing the classifier architecture issue, DIVER-1 demonstrates superior performance compared to baseline models on both datasets as well. Detailed results are provided in Appendix D.1 and Table 14. We plan to further validate on additional EEG datasets during the rebuttal period to provide more comprehensive evidence of generalization capability.

---

### Official Review · Reviewer_i3iQ · 2025-11-02

**Soundness:** 2
**Presentation:** 2
**Contribution:** 2
**Rating:** 2
**Confidence:** 3

**Summary:**

This paper presents DIVER-1, multimodal EEG and iEEG foundation models trained at large scale, up to 59k hours and 1.82B parameters. The key claim is a data-constrained regime where training longer on limited data beats scaling parameters alone. The authors fit scaling laws, outline a compute-optimal frontier, and report strong iEEG results on NeuroProbe with competitive EEG results on FACED. Architecture choices include any-variate attention, sliding temporal conditional positional encoding, register tokens, and multi-domain masked reconstruction.

**Strengths:**

The study offers the first systematic scaling analysis for electrophysiology and provides actionable guidance on allocating compute between model size and epochs. The corpus scale and subject diversity are substantial, and the any-variate attention plus STCPE address heterogeneous montages. Multi-domain reconstruction is well motivated. NeuroProbe results are strong, and the compute-optimal frontier is practically useful for planning.

**Weaknesses:**

The EEG performance on FACED lags prior state of the art despite far larger pretraining, which undermines the universality of the scaling conclusions. Several comparisons are missing or limited, for example to models that use different pretext tasks or longer temporal receptive fields, which complicates claims of state of the art. Important dataset and preprocessing choices that can swing outcomes are scattered in appendices, including QAQC criteria, referencing, resampling, clipping, and filtering. The scaling-law fits are reported for pretext loss, but mapping those improvements to downstream accuracy is not always consistent, especially across modalities. Some reproducibility elements are incomplete for a study whose central result is about principled scaling.

**Questions:**

To what extent do the findings hold when the downstream tasks are regression or continuous decoding rather than binary AUROC classification, especially language embedding regression. How much of the EEG performance gap is attributable to insufficient epochs versus differences in QAQC or dataset composition quality, and do the conclusions change when EEG pretraining uses multiple epochs and clipping strategies matched to iEEG.
Add baselines with longer contexts and alternative SSL objectives. Fit downstream scaling relations and report correlations between pretext loss and task metrics.

---

> ### Author Response · Authors · 2025-11-26
>
> Dear Reviewer i3iQ,
>
> We thank the reviewer for the thoughtful comment and careful assessment. To strengthen the manuscript, we incorporated additional explanations and supporting results. These revisions are marked in blue in the updated version.
>
> ## **W1:** EEG on FACED underperformance and Limits of Scaling Universality
>
> > “The EEG performance on FACED lags prior state of the art despite far larger pretraining, which undermines the universality of the scaling conclusions.”
>
> After investigating, we found that the **discrepancy did not arise from limitations of the DIVER-1 encoder, but from a mismatch in evaluation protocol in the initial submission**. Specifically, our EEG finetuning used a *linear head*, whereas state-of-the-art baseline EEG foundation models (CBraMod) use *3 layer MLP heads*. This inconsistency artificially suppressed DIVER-1’s EEG results in the first version.
>
> After correcting this and adopting the same multi-layer MLP head used by existing baselines (i.e., we standardized the evaluation rather than modifying the model), **DIVER-1 now achieves SOTA performance across all EEG benchmarks**. Importantly, this does not “help” the model; it simply restores fairness by using the same head depth that the CBraMod baseline already rely on.
>
> As shown in Figure 2 (t) and Appendix D.1, DIVER-1 surpasses prior EEG foundation models on all three tasks:
>
> |Model| |FACED| | |PhysioNet| | |MentalArith| |
> |----|----|----|----|----|----|----|----|----|----|
> ||ACC±SD|Kappa±SD|F1±SD|ACC±SD|Kappa±SD|F1±SD|ACC±SD|auc_pr±SD|AUROC±SD|
> |LaBraM|0.5273±0.0107|0.4698±0.0188|0.5288±0.0102|0.6173±0.0122|0.4912±0.0192|0.6177±0.0141|0.6909±0.0125|0.5999±0.0155|0.7721±0.0093|
> |CBraMod|0.5509±0.0089|0.5041±0.0122|0.5618±0.0093|0.6417±0.0091|0.5222±0.0169|0.6427±0.0100|0.7256±0.0132|0.6267±0.0099|0.7905±0.0073|
> |DIVER (ours)|0.6014±0.0075|0.5501±0.0086|0.6066±0.0088|0.6756±0.0032|0.5674±0.0043|0.6777±0.0036|0.7271±0.0176|0.6755±0.0460|0.8140±0.0259|
>
> Our expanded EEG scaling experiments (Figure 2m–o) also demonstrate that EEG performance scales predictably with both model size and pretraining data size, reinforcing that large-scale pretraining benefits EEG.
>
> We further provide a systematic head-depth ablation (Table 22; Section E.2 lines 2189-2220), confirming that deeper classifier heads are important — the exact design used in CBraMod and now matched in DIVER:
>
> |HeadDepth| |FACED| | |PhysioNet-MI| |
> |----|----|----|----|----|----|----|
> ||ACC±SD|Kappa±SD|F1±SD|ACC±SD|Kappa±SD|F1±SD|
> |Linear|0.523±0.018|0.461±0.020|0.521±0.018|0.653±0.013|0.538±0.017|0.654±0.013|
> |2-Layer|0.584±0.005|0.530±0.005|0.584±0.005|0.674±0.003|0.565±0.004|0.676±0.003|
> |3-Layer|0.601±0.008|0.550±0.009|0.607±0.009|0.676±0.003|0.567±0.004|0.678±0.004|
> |4-Layer|0.603±0.007|0.552±0.008|0.609±0.007|0.672±0.008|0.573±0.011|0.674±0.008|
> |5-Layer|0.594±0.021|0.543±0.023|0.601±0.019|0.660±0.007|0.547±0.009|0.662±0.006|
>
> These results confirm that the earlier discrepancy was due solely to mismatched evaluation protocols, not a shortcoming of the DIVER-1 model itself.
>
> ## **W2:** Missing SOTA Comparisons in SSL Objectives and Temporal Context
>
> > “Several comparisons are missing or limited, for example to models that use different pretext tasks or longer temporal receptive fields, which complicates claims of state of the art.”
>
> We agree with the reviewer that the current version can be strengthened by including comparisons to EFMs that (i) use different SSL objectives and (ii) operate with longer temporal receptive fields.
>
> **Long-context iEEG models (e.g., Brant) could not be evaluated on NeuroProbe, so we extended evaluation on the long-window iEEG benchmarks.** Brant uses 6-second input patches, which cannot be evaluated on NeuroProbe’s 1-second downstream format (as also noted in the line 1096). To address this limitation, we added MAYO iEEG seizure detection benchmark (6 s windows), where Brant and DIVER-1 can be compared both. In MAYO, DIVER overperformed Brant and BrainBERT (figure 2s and Table 12)
>
> **SSL-diverse EEG models (e.g., EEGPT) will be included**. For EEG, we will add comparisons to **EEGPT**, which uses a new SSL objective. This complements our existing comparisons against CBraMod and LaBraM, and broadens the spectrum of SSL objectives evaluated on EEG.
>
> If there are additional EFMs or SSL formulations the reviewer considers essential, we are happy to include them.

---

> ### Author Response · Authors · 2025-11-26
>
> ## **W3:** Scattered Reporting of Critical QAQC and Preprocessing Details
>
> > “Important dataset and preprocessing choices that can swing outcomes are scattered in appendices, including QAQC criteria, referencing, resampling, clipping, and filtering.”
>
> We appreciate this comment and have reorganized the description of our data and preprocessing pipeline to improve readability. In the revised **Section 3.1 (lines 287-294)**, we now provide a concise, self-contained summary of the QAQC and preprocessing procedures for both EEG and iEEG, including referencing, high-pass and notch filtering, resampling to 500 Hz, segmentation into 30 s windows, conservative QAQC, and channel/segment rejection criteria.
>
> Full details remain in **Appendix F.3 lines 2432-2446**. There, we also added how our QAQC recovers substantially more high-quality TUEG data than prior work (CBraMod) (e.g., 174.7k → 422k channel-hours, a 2.4× increase).
>
>
> ## **W4:** Inconsistent Mapping from Pretext Scaling to Downstream Accuracy
>
> > “The scaling-law fits are reported for pretext loss, but mapping those improvements to downstream accuracy is not always consistent, especially across modalities.”
>
> We thank the reviewer for highlighting the nuance between pretext scaling and downstream performance.
>
> First, we note that **in the newly added EEG downstream scaling results (Fig. 2 (m)–(o)), scaling trends across dataset size, model size, and epochs are much clearer than in iEEG** (discussed in lines 393-398), demonstrating that downstream scaling can indeed emerge when task complexity and sample size are sufficient.
>
> As suggested in Question 4, we **will add fitted downstream scaling curves and report correlations between pretraining loss and downstream metrics**.
>
> At the same time, we emphasize that **classical scaling-law literature is primarily framed in terms of training/validation loss, not downstream performance**, and **downstream scaling can be influenced by several confounding factors beyond pretext loss alone**: (i) the number of labeled samples in the downstream task (larger models may overfit in low-data regimes, as seen in iEEG where linear probing outperforms full finetuning), (ii) the inherent performance ceiling or difficulty of each task, (iii) hyperparameter sensitivity, and (iv) the choice of evaluation method (linear probing vs. full finetuning).
>
> We would like to end by noting that *data scaling* behaves very predictably, and increasing data volume reliably improves performance.
>
> ## **W5:** Incomplete Reproducibility for a Principled Scaling Study
>
> > “Some reproducibility elements are incomplete for a study whose central result is about principled scaling.”
>
> We appreciate the reviewer’s emphasis on reproducibility, which is especially important for a principled scaling analysis. In the revised manuscript we have substantially expanded the implementation and documentation needed to reproduce all experiments.
>
> - **Implementation details and hyperparameters.** We now include a dedicated section (Appendix B.5, Table 4) that enumerates all architectural hyperparameters of the DIVER-1 backbone, including CNN patch encoders, STCPE settings, transformer depth/width, attention heads, feed-forward dimensions, and loss-domain configurations. We also note that the initial submission already contained Appendix B.4 (pretraining setup, optimizer, µP initialization, LR schedulers, and distributed training environment) and Appendix B.6, which documents the full hyperparameter search procedure (grid + Optuna) used to select modality-specific learning rate and weight decay. Plus, we described finetuning details in Appendix B.8, which includes hyperparameter settings, optimizer configurations, and model-specific adjustments for both iEEG and EEG tasks.
>
> - **Code release and reproducibility materials.** We have publicly released the code and the tiny model weights. The full set of weights used in the scaling-law experiments—covering all size/epoch configurations—will be released upon acceptance.
>
> If there are specific reproducibility components the reviewer feels are still missing, we would be happy to incorporate them.

---

> ### Author Response · Authors · 2025-11-26
>
> ## **Q1:**
>
> > “To what extent do the findings hold when the downstream tasks are regression or continuous decoding rather than binary AUROC classification, especially language embedding regression.”
>
> We fully agree that regression and continuous decoding tasks are important evaluation targets for electrophysiology foundation models. Methodologically, our framework is agnostic to the downstream label type: the same pretrained representations can be paired with regression heads just as in CBraMod and other EFMs. However, public iEEG datasets with regression labels are scarce.Nevertheless, in an effort to make our evaluation as comprehensive as possible, we also leverage the updated multi-label tasks implemented in the latest (unpublished) version of the NeuroProbe code and report results on these as well (Figure 2r and line 892-894). For EEG, suitable regression tasks do exist, and we are actively incorporating at least one regression-based EEG evaluation; if we are able to complete this before the end of the rebuttal period we will include it, and otherwise we will add the full regression results in the camera-ready version, where new experimental results are permitted.
>
>
> ## **Q2**
>
> > “How much of the EEG performance gap is attributable to insufficient epochs versus differences in QAQC or dataset composition quality, and do the conclusions change when EEG pretraining uses multiple epochs and clipping strategies matched to iEEG.”
>
> We thank the reviewer for summarizing the potential causes we initially speculated about. As clarified in our response to W1, our **follow-up investigation shows that the primary source of the EEG performance gap was the MLP finetuning head**, not pretraining-related factors such as epochs, QAQC, or dataset composition. **After correcting this issue, DIVER-1 reaches SOTA EEG performance, resolving the discrepancy.**
>
> As supporting analysis, we also include (i) the full epoch-scaling results for EEG in Fig. 2(o) and (ii) dataset-composition experiments in the newly added Appendix Table 21. These confirm that while such factors have minor effects, they do not account for the earlier performance gap.
>
> In short, although we examined multiple plausible explanations, the main cause was the finetuning head, and fixing it restores the expected performance without altering our scaling conclusions.
>
> ## **Q3**
>
> > “Add baselines with longer contexts and alternative SSL objectives.”
>
> As noted in our response to W2, we added long-context baselines (e.g., Brant on 6 sec MAYO downstream task, line 895-900) and we will add additional SSL-diverse models (e.g., EEGPT), complementing the existing BrainBERT, PopT, CBraMod, and LaBraM comparisons.
>
> ## **Q4**
>
> > “Fit downstream scaling relations and report correlations between pretext loss and task metrics.”
>
> As described in our response to W4, we will include downstream scaling-law fits and report correlations

---

### Official Review · Reviewer_MZwT · 2025-11-04

**Soundness:** 2
**Presentation:** 3
**Contribution:** 3
**Rating:** 6
**Confidence:** 4

**Summary:**

This paper introduces DIVER-1, a family of multimodal electrophysiology (EEG and iEEG) foundation models. The training data includes both EEG and iEEG data, with most of the data coming from EEG. The architecture design introduces several key components tailored for Ephys data. The model is pretrained using a multi-domain reconstruction objective, learning to simultaneously predict masked-out patches in the time series, spectrum (FFT), and spectrogram (STFT) domains . The paper then presents a systematic analysis of this architecture's scaling properties and its performance on downstream decoding tasks.

**Strengths:**

The paper's core strength is its investigation of scaling laws for electrophysiology.  Interesting findings include:

-- Studying cost-effective training recipes in data-constrained settings, such as not spending flops on the largest models.

-- Studying architecture designs, such as a multi-domain reconstruction objective (learning time, FFT, and STFT) and a novel positional encoding (STCPE) .

-- The collection of a large scale Ephys pre-training dataset

The effort put into this paper was impressive.

**Weaknesses:**

I have a number of concerns about this paper.  None of these concerns are deal-breakers, but they add up to a non-trivial amount of total concern.

-- The majority of the experiments focused on iEEG (e.g., most of Figure 2), where including EEG data actually hurt performance (noted in Appendix D.4).  Moreover, the limited EEG experiments show that the proposed model does not outperform existing EEG foundation models.  Since the large majority of the training set is EEG, this makes the total number quite a bit less impressive.  In fact, I find the way the abstract is written a bit misleading, since the tone the abstract takes strongly implies (to me) that the entire dataset which is mostly EEG was beneficial in the experiments.  The iEEG dataset (37 subjects for 5K+ hours) is already quite large.

-- I think some of the claims are not fully supported.  Notably, it's not clear how much of the performance gains are due to the dataset, versus the model design choices.  For instance, if the PopT-style approach (which factorizes per-channel embeddings vs aggregation) was trained on a larger dataset, would those perform better?  (Note that I am not suggest the authors run this experiment, as that would be too much for this paper.  But these are discussion points worth raising in limitations.)

-- I think the data-constrained scaling laws finding could be presented with more nuance.  For example, if I look at Figure 2d, it looks like the XXL model (brown curve) will perform the best if allowed train on 64 epochs.  This seems to be corroborated in Figure 2a, where eventually the largest models will reach the lowest loss.  Of course, if you have both a data-constraint AND a FLOPS constraint, then the story changes and is more aligned with your claim.


Minor: The legends in Figure 2 need to be larger

**Questions:**

No other questions, beyond asking the authors to comment on my listed weaknesses.

---

> ### Author Response · Authors · 2025-11-25
>
> Dear Reviewer MZwT,
>
> Thank you for the insightful and constructive feedback, which has greatly contributed to improving our work. We revised the paper to include clearer descriptions and several additional analyses. All changes made during the rebuttal are indicated in blue in the revised PDF.
>
> We address the questions and weaknesses as below.
>
> ## W1. Focusing too much on iEEG
>
> > “... does not outperform existing EEG foundation models…”
>
> **We appreciate this concern and have identified the source of the discrepancy.** In the initial submission, our EEG finetuning used a **linear head**, whereas state-of-the-art baselines such as **CBraMod** use 3 layer MLP heads. This mismatch **artificially suppressed DIVER-1’s EEG results**—not because the model underperformed, but because the evaluation setup was not aligned with standard practice.
>
> **After correcting this mismatch by adopting the same multi-layer MLP head used by existing baselines (CBraMod), DIVER-1 now achieves SOTA performance across all EEG benchmarks.** Importantly, this does *not* constitute “helping” the model with a special head: **we simply standardized the evaluation protocol to match prior work,** ensuring a fair comparison.
>
> As shown in Figure 2 and Appendix D.1, DIVER-1 now surpasses prior foundation models on three tasks, as noted here:
>
> |Model| |FACED| | |PhysioNet| | |MentalArith| |
> |----|----|----|----|----|----|----|----|----|----|
> ||ACC±SD|Kappa±SD|F1±SD|ACC±SD|Kappa±SD|F1±SD|ACC±SD|auc_pr±SD|AUROC±SD|
> |LaBraM|0.5273±0.0107|0.4698±0.0188|0.5288±0.0102|0.6173±0.0122|0.4912±0.0192|0.6177±0.0141|0.6909±0.0125|0.5999±0.0155|0.7721±0.0093|
> |CBraMod|0.5509±0.0089|0.5041±0.0122|0.5618±0.0093|0.6417±0.0091|0.5222±0.0169|0.6427±0.0100|0.7256±0.0132|0.6267±0.0099|0.7905±0.0073|
> |**DIVER(ours)**|**0.6014±0.0075**|**0.5501±0.0086**|**0.6066±0.0088**|**0.6756±0.0032**|**0.5674±0.0043**|**0.6777±0.0036**|**0.7271±0.0176**|**0.6755±0.0460**|**0.8140±0.0259**|
>
> Finally, our expanded EEG scaling experiments (Figure 2m–o) show that EEG downstream performance scales predictably with both model size and data size, reinforcing that large-scale pretraining does benefit EEG tasks.
> (A systematic ablation over MLP head depth (1–5 layers) on FACED and Physionet benchmarks are shown as follows (**Table 22, Section E.2 line 2189-2220**)) :
> |HeadDepth| |FACED| | |PhysioNet-MI| |
> |----|----|----|----|----|----|----|
> ||ACC±SD|Kappa±SD|F1±SD|ACC±SD|Kappa±SD|F1±SD|
> |Linear|0.523±0.018|0.461±0.020|0.521±0.018|0.653±0.013|0.538±0.017|0.654±0.013|
> |2-Layer|0.584±0.005|0.530±0.005|0.584±0.005|0.674±0.003|0.565±0.004|0.676±0.003|
> |3-Layer|0.601±0.008|0.550±0.009|0.607±0.009|**0.676±0.003**|**0.567±0.004**|**0.678±0.004**|
> |4-Layer|**0.603±0.007**|**0.552±0.008**|**0.609±0.007**|0.672±0.008|0.573±0.011|0.674±0.008|
> |5-Layer|0.594±0.021|0.543±0.023|0.601±0.019|0.660±0.007|0.547±0.009|0.662±0.006|
>
>
> > “...including EEG data actually hurt performance…”, “...way the abstract is written a bit misleading, since the tone the abstract takes strongly implies (to me) that the entire dataset which is mostly EEG was beneficial in the experiments…”
>
> Thank you for pointing this out. We have removed all references to multimodal EEG foundation models that could confuse the reader.

---

> ### Author Response · Authors · 2025-11-25
>
> ## W2. Disentangling dataset and architecture effects in performance.
>
> > “... not clear how much of the performance gains are due to the dataset, versus the model design choices… ”
>
> We thank the reviewer for raising this important question. We agree that it is essential to disentangle performance gains attributable to pretraining dataset scale versus model architectural choices. To address this, we added **two new analyses in the revision**:
>
> 1. **Controlling for pretraining dataset size (isolating architecture effects)**. As suggested, we trained a DIVER-tiny model **solely on the BrainTreeBank (BTB)** dataset—the same dataset used to train PopT and BrainBERT (Appendix Table 19 (also included below for reference)) . Under this controlled setting, **DIVER still outperforms PopT and BrainBERT**, indicating that our modeling choices contribute meaningfully beyond dataset scale.
>
> 2. **Extensive architectural ablations (isolating component contributions)**. We conducted **new ablation experiments across both iEEG and EEG (Appendix Tables 17–18)** (included below for reference) by removing encoder components, spectral embeddings, positional encodings, and reconstruction heads. These experiments show that:
>     * In **EEG, all major components consistently improve performance**.
>     * In iEEG, several components improve some tasks but not others, suggesting task- and modality-specific interactions rather than redundant architectural complexity.
>
> Together, these results clarify that although dataset scale is beneficial—as shown in our pretraining-size experiments—**the architecture itself is a major source of gains**, and pretraining on larger data alone does not reproduce DIVER-1’s performance.
> We agree that future work should examine how PopT-like architectures behave when pretrained at larger scales.
>
> ### Same pretraining dataset ablation
>
> | Model                              | speech              | onset              |
> |-----------------------------------|----------------------|---------------------|
> | **BrainBERT (frozen)** (Zahorodnii et al., 2025) | 0.611 ± 0.022      | 0.757 ± 0.027      |
> | **BrainBERT (ours)**              | 0.575 ± 0.018        | 0.659 ± 0.026      |
> | **PopT (ours)**                   | 0.702 ± 0.029        | 0.780 ± 0.025      |
> | **PopT** (Zahorodnii et al., 2025) | 0.677 ± 0.044        | 0.689 ± 0.050      |
> | **DIVER Tiny/I/0.1s (frozen)** | **0.770 ± 0.028** | **0.859 ± 0.018** |
>
>
> ### FACED (9-class) Ablation
>
> | | ACC | kappa | F1 |
> |---|---|---|---|
> | w.o. RoPE | 0.408 ± 0.025 | 0.333 ± 0.027 | 0.414 ± 0.024 |
> | w.o anyV attention | 0.414 ± 0.012 | 0.339 ± 0.014 | 0.417 ± 0.012 |
> | w.o RoPE and anyV attention | 0.446 ± 0.007 | 0.376 ± 0.007 | 0.450 ± 0.005 |
> | w.o. STCPE | 0.463 ± 0.024 | 0.395 ± 0.027 | 0.471 ± 0.026 |
> | w.o Channel modality + subtype emb. | 0.474 ± 0.018 | 0.406 ± 0.021 | 0.482 ± 0.019 |
> | w.o Channel 3d position emb. | 0.481 ± 0.016 | 0.415 ± 0.018 | 0.487 ± 0.016 |
> | w.o Spectral feature emb. | 0.454 ± 0.015 | 0.386 ± 0.016 | 0.462 ± 0.013 |
> | w.o Multi-domain reconstruction (only raw) | 0.435 ± 0.008 | 0.364 ± 0.009 | 0.437 ± 0.006 |
> | **DIVER_Tiny/E/1s** | **0.491 ± 0.023** | **0.428 ± 0.025** | **0.502 ± 0.023** |
>
>
>
>
> ### PhysioNet-MI (4-class) Ablation
>
> | | ACC | kappa | F1 |
> |---|---|---|---|
> | w.o. RoPE | 0.614 ± 0.005 | 0.485 ± 0.006 | 0.615 ± 0.005 |
> | w.o anyV attention | 0.611 ± 0.003 | 0.481 ± 0.004 | 0.612 ± 0.004 |
> | w.o RoPE and anyV attention | 0.591 ± 0.005 | 0.454 ± 0.006 | 0.593 ± 0.005 |
> | w.o. STCPE | 0.626 ± 0.006 | 0.502 ± 0.008 | 0.627 ± 0.006 |
> | w.o Channel modality + subtype emb. | 0.629 ± 0.006 | 0.505 ± 0.007 | **0.632 ± 0.005** |
> | w.o Channel 3d position emb. | **0.629 ± 0.008** | **0.506 ± 0.010** | 0.631 ± 0.007 |
> | w.o Spectral feature emb. | 0.626 ± 0.005 | 0.501 ± 0.006 | 0.627 ± 0.004 |
> | w.o Multi-domain reconstruction (only raw) | 0.614 ± 0.005 | 0.485 ± 0.006 | 0.616 ± 0.004 |
> | **DIVER_Tiny/E/1s** | 0.628 ± 0.005 | 0.504 ± 0.007 | 0.630 ± 0.005 |

---

> ### Author Response · Authors · 2025-11-25
>
> ## W3. Scaling Law Finding and Message
> > “...data-constrained scaling laws finding could be presented with more nuance…”
>
> > “...at Figure 2d, it looks like the XXL model (brown curve) will perform the best if allowed train on 64 epochs…”, “...eventually the largest models will reach the lowest loss…”
>
> > “...if you have both a data-constraint AND a FLOPS constraint, then the story changes and is more aligned with your claim…”
>
> Thank you for pointing out this very important point. Indeed we should have more clearly conveyed that we are considering when there is **both a data AND FLOPS constraint**. We have updated our paper throughout–abstract (lines 18-19), introduction (lines 60-67), and results and discussion (374-380, 402-412)–to better express this very important nuance.
>
> In particular, we **added a dedicated “Isoloss Analysis” Section at line 374** better discusses this.
>
> Furthermore, we added iso-compute lines to the iso-compute plots in Figure 2 (p)’s compute optimal frontier plot for better understanding.
>
>
> ## W4. Larger figure
> > “Minor: The legends in Figure 2 need to be larger”
>
> We appreciate the reviewer for the helpful feedback regarding the visualization. We have enlarged the font size of the legends in Figure 2 in the revised manuscript, improving readability.

---

### Official Review · Reviewer_5x2y · 2025-11-07

**Soundness:** 2
**Presentation:** 4
**Contribution:** 3
**Rating:** 2
**Confidence:** 4

**Summary:**

In this paper, the authors systematically study the scaling of foundation models for electrophysiology data from macro-electrodes (EFMs), particularly iEEG and EEG. They introduce a family of models, DIVER-1, which share a fixed architecture and training regime but vary in data sources and model sizes. They then characterize scaling properties along several axes: model size, data scale, compute, number of training epochs, and number of subjects in the training data. Their analysis yields a set of scaling laws that describe EFM scaling in a data-constrained regime analogous to that observed in language models [1]. Results show state-of-the-art performance on the Neuroprobe benchmark (iEEG movie-watching) and performance comparable to prior methods on the FACED EEG benchmark for emotion classification.

---

[1] Muennighoff, N., Rush, A., Barak, B., Le Scao, T., Tazi, N., Piktus, A., ... & Raffel, C. A. (2023). Scaling data-constrained language models. Advances in Neural Information Processing Systems, 36, 50358-50376.

**Strengths:**

(S1) I appreciate the attention to detail in this paper to subtle yet important factors such as data quality (e.g., the QA/QC pipeline), hyperparameter tuning (thorough tuning at smaller scales and use of $\mu$-parameterization for transfer to larger models), and choices like patch size and fine-tuning strategy that can significantly affect performance.

(S2) To my knowledge, this is the first work to systematically study scaling with population-level electrophysiology data. Although the scope of results is somewhat limited (discussed in (W2) below), the derived scaling laws offer valuable guidance for future work in this area.

(S3) The choice to evaluate on comprehensive benchmarks such as Neuroprobe and FACED demonstrates an emphasis on thorough evaluation, and it’s encouraging that DIVER-1 performs consistently across most Neuroprobe tasks.

**Weaknesses:**

(W1) A major concern is the consistency of the Neuroprobe results. In the current version of the benchmark [1], the reported numbers for the Linear (Laplacian + spectrogram) baseline on SS-SM differ from those in the manuscript. For instance, the Global Optical Flow result for the linear baseline is $<0.62$, and DIVER-1 achieves roughly $0.62$, yet the Neuroprobe paper reports $0.625$. Some task results align with the Linear (spectrogram) baseline, others with Linear (Laplacian + spectrogram). It’s possible I’ve misunderstood, but I couldn’t find a consistent basis for comparison. Since the benchmark is concurrently being developed, some variation is expected, but it should be clarified whether baseline results were re-run or taken directly from a source, and cited accordingly. As far as I know, the Linear (Laplacian + spectrogram) results are only present in the latest version of Neuroprobe, so I would expect consistency with [1]. This paper also includes 19 tasks, several of which are absent in [1]. Additionally, including tables with full results would help, as the plots alone make it difficult to gauge precise values when differences are marginal.

(W2) While robust benchmarking for iEEG data is still an open problem, the current results suggest that extensive pretraining yields only marginal gains over linear baselines. In some sense, I believe the cost of pretraining is not justified by the demonstrated results and hence the pursuit of a scaling study is not fully justified as well.

a) Note that on some Neuroprobe tasks, careful feature extraction on linear models (e.g., Laplacian + spectrogram vs. raw voltage) achieves greater relative improvement vs the effect of scaling or the performance gain achieved by DIVER-1 over the linear baselines. This raises questions about whether the observed gains would persist against stronger baselines. While I acknowledge the lack of robust benchmarking for iEEG reflects a broader issue in the field, I do think the choice to use the binary classification tasks in Neuroprobe as a primary evaluation metric undermines the credibility of the scaling claims.

b) I would suggest leaning more on EEG benchmarks, which are comparatively better developed. Notably, DIVER-1 does not achieve SOTA performance on FACED, hence I am not confident that the current evaluation setup adequately supports the premise of the scaling study.

(W3) Because DIVER-1 combines novel architectural design choices and large-scale pretraining, it’s difficult to disentangle their respective effects. The only apparent comparison with a supervised variant is in Table 6 (Appendix), which is described as “fine-tuned from scratch.” Clarification is needed on whether a supervised baseline using the same architecture was directly compared. Moreover, comparisons with SSL baselines like PopT, BrainBERT, or Brant (which wasn't compared at all) are incomplete without controlling for pretraining data and scale (e.g. by comparing supervised variants).

(W4) The ablations in Table 7 (Appendix) indicate that most DIVER-1 components have limited contribution--only the multi-domain reconstruction objective and STCPE embeddings seem clearly beneficial, while the other components seem to in fact hurt performance. While it’s possible other components help at larger scales, no evidence is provided.

(W5) Evaluation is limited to the SS-SM setting on Neuroprobe. Since linear probing is already used, comparisons against the SS-DM and DS-DM settings (as reported in Neuroprobe) would be lightweight to derive, greatly strengthen the analysis, and clarify generalization behavior which is a key concern for a scaling study.

To me, (W1) and (W2) are especially critical for considering acceptance of this paper. I will consider raising my score if the authors are able to address these concerns effectively.

---

[1] Zahorodnii, A., Wang, C., Stankovits, B., Moraitaki, C., Chau, G., Barbu, A., ... & Fiete, I. R. (2025). Neuroprobe: Evaluating Intracranial Brain Responses to Naturalistic Stimuli. arXiv preprint arXiv:2509.21671.

**Questions:**

(Q1) There are limited details on some architectural components. How deep is the CNN? It’s mentioned that the STCPE embeddings used were modified versus previous implementations, but how exactly? Do the authors plan to release code?

(Q2) Limited details on decoder. Is it MAE-style? What is used to query the output, a learnable mask token? How about spatial coordinates, timing information? Since timing information is only used in the SA layers processing unmasked patches, and the multi-output decoder head features linear projection, do the reconstruction heads not include timing information?

(Q3) Why is STFT excluded from the pretraining loss for .1s patches (which seem to perform best)? How were $\lambda_1$, $\lambda_2$, $\lambda_3$ chosen? Why is FFT emphasized less for 1s patches, but more important for .1s patches?

(Q4) During input resampling, is $N’ \leq \min(N, 30)$ fixed for the number of subsampled temporal patches, irrespective of patch size? For .1s patches would that mean it’s subsampling 10% of the patches? Also are patches resampled totally randomly? If so, doesn’t this discard substantial information? What’s the intuition for why this wouldn’t harm training?

(Q5) To confirm: the models were pretrained on 30s context windows and directly inferred/finetuned on different context windows downstream (e.g. 1s for Neuroprobe, 10s for FACED)?

(Q6) Performance appears highly dependent on patch size, and I would imagine this is especially true across different tasks. Could the authors analyze how patch size affects reconstruction and downstream results? I think this could be another important axis of consideration, even at smaller scales.

(Q7) Since linear probing on Neuroprobe often outperforms full fine-tuning, it would be informative to report scaling effects across all downstream tasks and analyze differences across task domains--for example, whether scaling trends differ between language and auditory tasks?

(Q8) For the results throughout the text where only the four tasks (speech, onset, volume, pitch) were provided, I want to confirm that these are on the Neuroprobe splits and not on previous Braintreebank splits as in from [1]?

(Q9) In pretraining with AJILE12, were blocklist periods filtered out or was the full data used? If noisy segments were included, were their potential effects considered?

(Q10) AJILE12 was already preprocessed before public release. Were the QAQC steps still applied to this dataset?

---

[1] Chau, G., Wang, C., Talukder, S., Subramaniam, V., Soedarmadji, S., Yue, Y., ... & Barbu, A. (2025). Population transformer: Learning population-level representations of neural activity. ArXiv, arXiv-2406.

---

> ### Author Response · Authors · 2025-11-25
>
> Dear Reviewer 5x2y,
> Thank you for the insightful and constructive feedback, which has greatly contributed to improving our work. We have added a number of clarifications and supplemental experiments in the revision. All new or modified text is highlighted in blue in the updated PDF. We address the questions and weaknesses as below.
>
>
> ## 1. Consistency of the Neuroprobe results
> > “... results are only present in the latest version of Neuroprobe, so I would expect consistency … “
>
>
> Thank you for pointing this out. During our development, Neuroprobe underwent several updates that we did not fully track. We have now updated all results to match the current benchmark version:
>  - Corrected preprocessing to "linear STFT with Laplacian"
> - Updated task set (15 tasks instead of 19: removed local/global flow angle, delta pitch, and speaker)
> - Updated all figures and tables accordingly
>
> > “...Additionally, including tables with full results would help…”
>
>
> Thank you for the suggestion. We now include a table with full results in Appendix Table 12. This table provides a comprehensive numerical comparison of DIVER against baseline models across all 15 Neuroprobe tasks. This would enable a precise evaluation of performance differences that may be difficult to discern from plots alone.
>
> ## 2. Insufficient downstream validation and marginal gains over baselines
> >"... pretraining yields only marginal gains over linear baselines, the cost of pretraining is not justified by the demonstrated results and hence the pursuit of a scaling study is not fully justified as well."
>
> >"...suggest leaning more on EEG benchmarks, which are comparatively better developed." "...DIVER-1 does not achieve SOTA performance on FACED"
>
>
> We appreciate the reviewer’s concern that DIVER-1 “does not outperform existing EEG foundation models.”
>
>  After revisiting our EEG pipeline, we discovered that the discrepancy came from an evaluation mismatch rather than a limitation of DIVER-1. In the initial submission, we used a linear classification head for EEG finetuning, whereas existing EEG foundation models  (CBraMod) use 3 layer MLP heads. This mismatch caused our originally reported EEG scores to appear lower than they should have been.
>
> After **aligning our evaluation with the standard protocol by using the same type of multi-layer MLP head** employed by the baselines (not a special or modified head), **DIVER-1 achieves state-of-the-art results on all three EEG benchmarks**. This adjustment ensures a fair comparison and reveals the model’s true representational strength.
>
> Updated SOTA results (Figure 2, Appendix D.1):
>
> |Model| |FACED| | |PhysioNet| | |MentalArith| |
> |----|----|----|----|----|----|----|----|----|----|
> ||ACC±SD|Kappa±SD|F1±SD|ACC±SD|Kappa±SD|F1±SD|ACC±SD|auc_pr±SD|AUROC±SD|
> |LaBraM|0.5273±0.0107|0.4698±0.0188|0.5288±0.0102|0.6173±0.0122|0.4912±0.0192|0.6177±0.0141|0.6909±0.0125|0.5999±0.0155|0.7721±0.0093|
> |CBraMod|0.5509±0.0089|0.5041±0.0122|0.5618±0.0093|0.6417±0.0091|0.5222±0.0169|0.6427±0.0100|0.7256±0.0132|0.6267±0.0099|0.7905±0.0073|
> |DIVER (ours)|0.6014±0.0075|0.5501±0.0086|0.6066±0.0088|0.6756±0.0032|0.5674±0.0043|0.6777±0.0036|0.7271±0.0176|0.6755±0.0460|0.8140±0.0259|
>
> Consistent with these findings, our expanded EEG scaling experiments (Figure 2m–o) show that EEG downstream accuracy improves reliably with both model size and pretraining data size.
>
> We also include a structured ablation over head depth (Table 22; Section E.2 lines 2189–2220), illustrating that 3–4 layer MLP heads are optimal for EEG tasks—the same design used by CBraMod and now matched in our evaluation:
>
>
> |HeadDepth| |FACED| | |PhysioNet-MI| |
> |----|----|----|----|----|----|----|
> ||ACC±SD|Kappa±SD|F1±SD|ACC±SD|Kappa±SD|F1±SD|
> |Linear|0.523±0.018|0.461±0.020|0.521±0.018|0.653±0.013|0.538±0.017|0.654±0.013|
> |2-Layer|0.584±0.005|0.530±0.005|0.584±0.005|0.674±0.003|0.565±0.004|0.676±0.003|
> |3-Layer|0.601±0.008|0.550±0.009|0.607±0.009|0.676±0.003|0.567±0.004|0.678±0.004|
> |4-Layer|0.603±0.007|0.552±0.008|0.609±0.007|0.672±0.008|0.573±0.011|0.674±0.008|
> |5-Layer|0.594±0.021|0.543±0.023|0.601±0.019|0.660±0.007|0.547±0.009|0.662±0.006|
>
> Together, these updates confirm that the initial discrepancy was due entirely to inconsistent evaluation settings, not a performance gap. With matched evaluation protocol, DIVER-1 clearly exceeds previous EEG foundation models.
>
>
> ## 3. Disentangling architectural design choices and large-scale pretraining
> >”...The only apparent comparison with a supervised variant is in Table 6 (Appendix), which is described as “fine-tuned from scratch.” Clarification is needed on whether a supervised baseline using the same architecture was directly compared”
>
>
> We thank you for this comment and agree that the clarification is necessary. We have now clearly specified this information in line 1567 and in the corresponding table (now Table 15).

---

> ### Author Response · Authors · 2025-11-25
>
> > “... Moreover, comparisons with SSL baselines like PopT, BrainBERT, or Brant (which wasn't compared at all) are incomplete without controlling for pretraining data and scale (e.g. by comparing supervised variants).”
>
> First, regarding the comparison with Brant, as written in line 1096, the neuroprobe window is 1 sec, whereas Brant requires a minimum window of 6s (due to patch size being 6s), so we were unable to perform a direct comparison on the neuroprobe dataset. We extended our analysis to another iEEG dataset, MAYO (with a 6-second window size), and compared performance with Brant. Unfortunately, the Brant model with shared weights was pre-trained on 90-second context windows, resulting in a mismatch in context size. DIVER outperformed both Brant and BrainBERT on the MAYO dataset by a large margin (figure 2s).
>
> We also agree that a comparison controlling for the pretraining data is necessary, because PopT and BrainBERT were pretrained only on the BTB dataset, which is smaller than the dataset used to train our models. We trained a DIVER-tiny-0.1s model solely on the same BTB dataset and compared its performance, and **our model achieved the best performance among the other baselines (BrainBERT and PopT) pretrained on the same dataset** (Table 19 and line 1994) (Table provided below for easy reference). We are very grateful that you pointed this out, and we have added a new paragraph at line 454 in the main text.
>
>
> | Model                              | speech              | onset              |
> |-----------------------------------|----------------------|---------------------|
> | **BrainBERT (frozen)** (Zahorodnii et al., 2025) | 0.611 ± 0.022      | 0.757 ± 0.027      |
> | **BrainBERT (ours)**              | 0.575 ± 0.018        | 0.659 ± 0.026      |
> | **PopT (ours)**                   | 0.702 ± 0.029        | 0.780 ± 0.025      |
> | **PopT** (Zahorodnii et al., 2025) | 0.677 ± 0.044        | 0.689 ± 0.050      |
> | **DIVER_{Tiny/I/0.1s} (frozen)** | **0.770 ± 0.028** | **0.859 ± 0.018** |
>
>
>
> ## 4. Limited Ablation Tests
> > “... most DIVER-1 components have limited contribution--only the multi-domain reconstruction objective and STCPE embeddings seem clearly beneficial...”
>
> Thank you for pointing this out. We did two things to address this :
> - We performed ablation on more DIVER components (RoPE, any variate attention, and both RopE and any variate attention  (Table 17))
> - We also included EEG ablations (Table18), which clearly show that while for iEEG models some of the components do not meaningfully contribute to performance, for EEG all components consistently do.
>
> Overall, we found that each component can have different effects on performance depending on the task and modality (iEEG vs. EEG). A more detailed analysis of which model components help when and for which modality (iEEG/EEG), component by component, should be explored in future works.
>
> ## 5. Evaluation is limited to the SS-SM setting on Neuroprobe.
> > “comparisons against the SS-DM and DS-DM settings (as reported in Neuroprobe) would be lightweight to derive, greatly strengthen the analysis, and clarify generalization behavior which is a key concern for a scaling study”
>
> We also agree with this point. However, because our model operates on all channel–time tokens (in contrast to PopT, which compresses along the channel dimension), it is computationally challenging to train the model directly in this setting. We therefore leave the development of architectures that can address this limitation to future work.
>
> -------
>
>
> ## Q1. limited details on some architectural components.
> >”...How deep is the CNN?”
>
>
> The CNN we used has three layers, and we have added this description at line 148. We have also included CNN and other architectural details in Appendix B.5 and Table 4 (covering general hyperparameters and related settings).
> > “...It’s mentioned that the STCPE embeddings used were modified versus previous implementations, but how exactly?”
>
>
> We have substantially expanded the STCPE embedding section in the main text (Section 2.2, Key Component Details, line 197-224), including explicit equations, a more in-depth comparison with ACPE and CPE, and a clearer explanation of the underlying motivations.
> >”...Do the authors plan to release code?”
>
>
> We provide a link to our released code and to the smallest model checkpoint. Upon acceptance, we will release all model weights (across epochs, model sizes, patch sizes, and modalities) used to generate the scaling plots, thereby enabling the community to fully replicate and extend our experiments.
>
>
> ## Q2. Limited details on decoder
> > “...What is used to query the output, a learnable mask token?”
>
>
> The encoder input contains zero-padded values at the masked positions; we do not use a learnable mask token. We have clarified this in Section 2.1, line 148.

---

> ### Author Response · Authors · 2025-11-25
>
> > “...How about spatial coordinates, timing information?
>
> Temporal information and channel (spatial) information are injected in the encoder via RoPE and any-variate attention, and these are applied in the same way to both masked and unmasked positions.
>
> To further clarify our design: we only use a MOIRAI-based Transformer encoder, and do not employ a separate, dedicated decoder stack. More precisely, for each masked position, the encoder output is passed through a shallow reconstruction head consisting of a single patch-wise linear layer to reconstruct the original patch. This patch-wise reconstruction head is what we refer to as the “decoder,” rather than a full MAE-style decoder. We must admit that we did not fully grasp the intent behind the question regarding the decoder details. If you could clarify which specific aspects of the decoder you are most interested in, we would be happy to provide a more precise and detailed explanation.
>
>
> ## Q3. Justification for the lambda values in multi-domain reconstruction
> >  “..Why is STFT excluded from the pretraining loss for .1s patches (which seem to perform best)? How were $\lambda_1$, $\lambda_2$, $\lambda_3$  chosen? Why is FFT emphasized less for 1s patches, but more important for .1s patches?”
>
>
> We have written the heuristic justification behind the lambda value choices (line 270). We add the all-or-none ablation for each component in line 1978 and table 18. We will conduct more fine-grained control of the λ values and include in the camera-ready version.
>
>
> ## Q4. Clarifications on resampling, and why for 0.1s, only subsample 10%,
> > “...For .1s patches would that mean it’s subsampling 10% of the patches? Also are patches resampled totally randomly? If so, doesn’t this discard substantial information? What’s the intuition for why this wouldn’t harm training?”
>
>
> Part of the reason was that the dataset was preprocessed and saved in 30-second chunks. We have added clarifications on this in the main text line 277. Additionally, since we use longer training epochs, we assumed the model would leverage all available information (i.e. across multiple epochs, different portions of the data are shown to the model, effectively, making the model see all the original data). We believe this is reflected in the fact that the 0.1s patch size demonstrates higher epoch efficiency (higher $R_D*$ value (for discussion, please see line 387-392)).
>
> ## Q5 .
> >“...To confirm: the models were pretrained on 30s context windows and…”
>
>
> The context window lengths are not fixed: they are up to 30 s in the 1 s patch model and up to 3 s in the 0.1 s model. We have added details on how these windows are selected in line 277.
>
> >”... directly inferred/finetuned on different context windows downstream (e.g. 1s for Neuroprobe, 10s for FACED)?...”
>
>
> Yes. In downstream tasks such as Neuroprobe (1 s) and FACED (10 s), we directly fine-tune/infer with task-specific window sizes. Because the context window during pretraining is randomly sampled, DIVER-1 is naturally exposed to a range of window lengths, which we believe helps it adapt smoothly to downstream settings with different context window sizes.
>
>
> ## Q6.
> >“...Could the authors analyze how patch size affects reconstruction and downstream results?”
>
>
> Thank you for the suggestion, and yes patch size does affect performance, and we in fact compare their results in Appendix Table 15. We believe that for neuroprobe, 0.1s patches work much better as this ensures 10 temporal patches for the 1s input, whereas using 1s patch sizes would mean the DIVER model does not see any temporal patches.
>
> ## Q7.
> >“... it would be informative to report scaling effects across all downstream tasks and analyze differences across task domains--for example, whether scaling trends differ between language and auditory tasks?”
>
>
> This seems like an interesting idea, and we are currently running this. We will update you once we have it ready.
>
> ## Q8.
> >“...I want to confirm that these are on the Neuroprobe splits and not on previous Braintreebank splits as in from [1]”
>
>
> To ensure reproducibility and proper train/test splits, we use the properly train-test split in Neuroprobe.
>
> ## Q9. AJILE blocklist periods
> > “... were blocklist periods filtered out or was the full data used?”
>
>
> We utilized the full AJILE12 dataset without filtering out the blocklist periods.
> > “...If noisy segments were included, were their potential effects considered?”
>
> The noisy segments were indeed included. Our rationale was that large-scale foundation models, such as DIVER, would actually benefit from exposure to diverse neural signal patterns, including irregularities like seizures, and develop robustness.
>
> ## Q10 AJILE preprocessing
> > “...Were the QAQC steps still applied to this dataset?”
>
> Given that  AJILE12 was already preprocessed before public release, we did not apply any additional signal-level preprocessing or QAQC steps. We just apply the segmentation of the recordings into 30 s.

---

### Author Response · Authors · 2025-12-04
**Explanation of Revisions and Response**

Our study makes several important contributions to the field of EEG
foundation models:

-   **State-of-the-art DIVER-1 model family**: We introduce a family of Ephys foundation model (EFMs) for both iEEG and EEG  that each achieve state-of-the-art performance in their respective modality's downstream tasks through systematic scaling and architectural innovations.
-   **First systematic, quantitative scaling law analysis for EFMs**: Our paper is the first study to show electrophysiology scaling law
    along model size, dataset size, and the number of pretraining epochs.
-   **Unprecedented scale demonstration**: We scale EFM across data volume, model size, and compute to unprecedented scales
- **New architectural innovations** : We introduce new architecture innovations including any-variate attention, sliding temporal conditional positional encoding (STCPE), register tokens, and multihead prediction architectures.

The reviewers were unanimous in commending the rigor of our pretraining
experiments, for example, the unprecedented scale of pretraining in the
field of multichannel EFM and the systemic hyperparameter-search on
scaling. However, they also identified a common weakness: the initial
version of the manuscript did not evaluate downstream task performance
with sufficient rigor (e.g., lack of SOTA results on the EEG benchmark
and and limited range of benchmark tasks). We have
substantially addressed these issues in the revised manuscript, and our
detailed responses to the reviewers' comments are summarized below.

# 1. Overview of Major Changes

The key changes (of the now revised manuscript)
are as follows:

-   **SOTA through Consistent EEG/iEEG evaluation protocols (see §2.1)**.
    Reviewers raised concerns that our initial submission did not
    achieve SOTA performance on EEG downstream tasks. We have since
    discovered an evaluation-head mismatch: our initial EEG results used
    a linear classifier, whereas our comparison model (CBraMod) use a
    3-layer MLP in their released code. Due to this unfair comparison
    and lack of transparent head architecture information in their
    paper, our initial results appeared weaker. We have now adopted the
    same 3-layer MLP head as CBraMod for all EEG results, whilst
    ensuring that the DIVER-1 encoder remains unchanged.(Appendix E.2)
    We want to emphasize that this is a correction to fairness of
    comparison and reporting, not a new architectural contribution. We
    also discover potential problems with reproducibility issues with
    CBraMod, which we describe in Appendix E.3 and E.4. Despite trying  various finetuning options in CBraMod codes, some downstream
    performances remained below CBraMod's reported values.

-   **Expanded EEG/iEEG benchmark suite (see §2.1).** We substantially
    expanded our benchmark evaluations to demonstrate the generality of
    our EEG foundation model. In EEG, we have incorporated
    PhysioNet-MI\[1\], MentalArithmetic\[2\], to evaluate across three
    task familes (emotion, motor, cognitive) under a unified protocol.
    In iEEG, we additionally evaluated performance on another downstream
    task, MAYO (a seizure detection task)\[3\] and Neuroprobe multilabel
    setting\[4\].

-   **Demonstration of downstream scaling  (see §2.1).** We show that simliar to iEEG, the newly added EEG downstream benchmarks do scale with data, model-size, and epochs.

-   **Better description of scaling-law interpretation under realistic
    compute (see §2.3).** We now explicitly clarify within our
    conclusion, \"smaller models trained longer outperform larger models
    trained briefly\", is **derived under joint data and FLOPs
    constraints**. Furthermore, IsoLoss+IsoCompute plots are utilized
    to illustrate that while the largest models eventually win with
    unbounded FLOPs, under realistic budgets the smaller/mid‑sized
    models actually demonstrate higher performance.(Figure 2(p),
    Appendix C.3)

-   **Scaling Law for both modalities(iEEG, EEG) in pretraining and
    finetuning (see §2.3).** We have added comprehensive scaling
    analysis for EEG, covering both pretraining (Figure 2 e-h) and
    downstream finetuning (Figure 2 m-o, Appendix C.6). Combined with
    our iEEG scaling results (Figure 2 a-d for pretraining, i-l for
    finetuning), we now provide complete scaling characterization across
    both modalities.

-   **Architectural ablations and same-data comparison (see §2.2).** We
    further summarized new analyses to clarify the impact of data scale
    from architecture(Section 4.3): (i) same-data comparison where
    DIVER-tiny is pretrained only on BrainTreeBank (matching
    PopT/BrainBERT)(Appendix D.3, Figure 12, Table 19), and (ii)
    component ablations across modalities (Table 17,18).

---

> ### Author Response · Authors · 2025-12-04
> **Explanation of Revisions and Response**
>
> -   **Clearer QA/QC and implementation details (see §2.4--2.5).** We
>     integrated QA/QC preprocessing summary(Appendix F.3) and elucidated
>     CNN depth, STCPE operation and decoder design (Appendix B.5).
>
> -   **Release of finetuning code for reproducibility.** We have made our
>     complete finetuning code for all downstream tasks and weights of
>     small models publicly available.
>
> # 2. Responses grouped under Themes
>
> ## 2.1 Achieving SOTA Through Fair Comparison and Extended Evaluation
>
> **Concern (5x2y, MZwT, i3iQ, UXXE, 6kz9):** DIVER-1 appears to
> underperform CBraMod on FACED; is the evaluation protocol fair?
>
> **Response.** We agree that, as originally reported, DIVER-1 does indeed
> appear weaker than CBraMod on FACED. Our subsequent follow-up analysis
> shows that this was due entirely to an evaluation-head mismatch: to
> clarify, our initial submission used a linear classifier while baselines
> (CBraMod) utilized 3-layer MLP. Therefore, we have since adopted the
> same 3-layer MLP as evaluation-head on all our EEG results. Under the
> synonymous protocol, we now illustrate that DIVER-1 achieves greater
> Accuracy/Kappa/F1 than LaBraM and CBraMod on FACED, PhysioNet-MI, and
> MentalArithmetic (Appendix D.1, Table 14). Our head-depth ablation (1--5
> layers) confirms 3--4-layer MLPs (as in our comparison model, CBraMod) are optimal(Appendix E.2), which
> illustrates that the prior discrepancy/ concern raised was a protocol
> issue and not a reflection of our model's inherent encoder weakness. We
> want to further emphasize that this revision made was purely protocol
> correction---not of a new architectural contribution---and was
> implemented specifically to address the reviewers' concern of our
> initial underperformance relative to CBraMod.
>
> **Concern (6kz9, 5x2y, UXXE, i3iQ):** EEG evaluation is too narrow (only
> FACED); generalization unclear, downstream scaling lacking.
>
> **Response.**
>
> We have since revised our manuscript to now evaluate DIVER-1 on three
> EEG datasets: FACED (9-class emotion, 32-ch), PhysioNet-MI (4-class
> motor imagery, 64-ch), and MentalArithmetic (2-class workload). Under
> our unified protocol (encoder + 3-layer MLP), DIVER-1 achieves **SOTA**
> across all three as previously stated. We have also incorporated
> additional scaling experiments (accuracy vs. model size/data) that
> support generalization beyond emotion decoding for clarification
> purposes (Figure 2(t), Appendix D.1, Table 14).
>
>
> **Concern (5x2y, UXXE):** Lack of comparison with longer patch size baselines (Brant) [5].
>
> **Response.**
>
> We test Brant, DIVER, and BrainBERT on MAYO iEEG seizure classification task (binary classification; lines 896–900) and show that DIVER outperformed both models (Figure 2(s), Appendix D.1, Table 12). We also clarify in our responses to the reviewers that Brant couldn't be compared to our model in Neuroprobe due their patch size being larger than the input window of the Neuroprobe, and that we therefore implement MAYO downstream task and test performance there.
>
>
> ## 2.2 Disentangling architecture and data gains
>
> **Concern (5x2y, MZwT, 6kz9):** Are improvements primarily due to more
> pretraining data, or to the DIVER-1 architecture?
>
> **Response.** We have since quantified both and is reflected on the
> revised manuscript, to specify: (i) *Same-data comparison*---DIVER-tiny
> pretrained only on BrainTreeBank (matching PopT + BrainBERT) still
> outperforms on NeuroProbe speech/onset, showing architecture alone
> yields gains (Appendix Table 19). (ii) *Component ablations*---appendix
> tables have also been updated to include RoPE, any-variate attention,
> and individual components of multi-domain reconstruction in addition to
> existing component ablations: STCPE, spectral/channel embeddings, and
> multi-domain reconstruction. Most components reflect these improvements
> for EEG in our model; for iEEG, effects are task-dependent---thereby
> suggests modality-specific contributions (Appendix Table 17, 18).

---

> ### Author Response · Authors · 2025-12-04
> **Explanation of Revisions and Response**
>
> ## 2.3 Clarifying Scaling Law Interpretations and Compute-Optimality
>
> **Concern (MZwT, UXXE, i3iQ):** The "smaller model trained longer"
> message seems at odds with larger models eventually achieving lower
> loss; IsoLoss contours noisy; DIVER-1 not on compute-optimal frontier.
>
> **Response.** We now make our assumptions explicit and elucidated in the
> revised manuscript: (i) Our conclusion was under the assumption of fixed
> data + FLOPs; we acknowledge that with unconstrained compute, XXL would
> eventually reach lowest loss. (ii) Our IsoLoss+IsoCompute curves (>30
> configurations) show that for realistic FLOPs budgets, smaller/mid-sized
> models trained longer outperform briefly-trained XXL. (iii) We have also
> acknowledged that our contours are less smooth due to sparser sampling,
> but our fitted law attains $R^2 \sim 0.75$--$0.82$ and predicted
> vs. empirical contours harmonize qualitatively. (iv) We removed the
> ambiguous "frontier star" and now clarify that our chosen checkpoint
> lies slightly beyond compute-optimal to get better performance even if
> FLOPs inefficient. (See our response to reviewer UXXE Q4)
>
>
> ## 2.4 Updating NeuroProbe Evaluation and Clarifying Near-Chance Tasks
>
> **Concern (5x2y, UXXE):** Inconsistencies with latest NeuroProbe
> baselines; only SS-SM evaluated; several tasks near chance.
>
> **Response.** During development, Neuroprobe was updated multiple times
> without our awareness. We have updated our iEEG experiments to match the
> current NeuroProbe benchmark: (i) our preprocessing now uses linear STFT
> with Laplacian referencing, (ii) we retain all 15 tasks, (iii) our
> BrainBERT/PopT scores now match the official implementation (Section
> 3.2). We will add SS-DM/DS-DM results before camera ready version.
>
> Additionally, for near-chance tasks (FrameBrightness, FaceNum), we
> clarify that the electrode coverage excludes lower-level visual cortex.
> All models---including the strongest linear baselines---operate at
> chance, so we view these as non-informative under this recording
> configuration, rather than DIVER-1-specific weaknesses.
>
> ## 2.5 Clarifying Implementation Details and Ensuring Reproducibility
>
> **Concern (i3iQ, 6kz9, 5x2y, UXXE):** QA/QC scattered; STCPE/CNN/decoder
> details unclear; code release?
>
> **Response.** We have consolidated these details and are currently
> reflected in our revised manuscript:
>
> *QA/QC and preprocessing.* Our main text now summarizes (with full
> details in appendix): signal normalization (EEG/iEEG scales),
> conservative clipping/channel rejection, referencing, minimal filtering
> (high-pass+notch), 500 Hz resampling, and 30 s windows. We additionally
> highlight that our pipeline recovers substantially more usable TUEG data
> than prior work.
>
> *Architectural details.* We have also since clarified that: (i) our
> patchwise CNN has 3 layers (dims/kernels/strides illustrated in appendix
> (Appendix B.5)), (ii) our STCPE uses sliding windows through MOIRAI;
> overlapping outputs aggregate to input-dependent positional encodings
> (temporally translation-equivariant and
> channel-permutation-equivariant), (iii) we do not use an MAE-style
> decoder; masked positions are zeroed and a single linear head
> reconstructs time/FFT/STFT, (iv) for 0.1s patches, we exclude STFT;
> $\lambda$ weights are chosen heuristically (we leave full grid search to
> future work), (v) our pretraining utilizes variable contexts (up to 30s
> for 1s patches, 3s for 0.1s); downstream tasks uses its native windows.
>
> *Reproducibility and code release.* We have released: (i) complete
> finetuning code with preprocessing scripts and configurations, and (ii)
> pretrained weights for smaller models. Larger model weights and
> intermediate checkpoints used in scaling-law analyses will be made
> available progressively. We are committed to open science and will
> ensure all resources are publicly accessible.
>
> Hence, we believe that our current up-to-date revised manuscript
> clarifies (where needed) and directly addresses the core concerns raised
> by reviewers in regards to: DIVER-1 model's overall EEG performance,
> fairness of comparison, scaling-law interpretation, and robustness in
> our empirical analysis.
>
>
> ------
>
> We believe this revised manuscript comprehensively addresses the reviewers' concerns and demonstrates that DIVER-1 represents a significant advancement in electrophysiology foundation models. Through fair evaluation protocols, expanded benchmarking, and rigorous scaling analysis, we have achieved state-of-the-art performance while providing valuable insights into neural foundation model scaling.
> We are grateful to the reviewers for their constructive feedback, which has substantially strengthened our work. We have also released our finetuning code and model weights to support reproducibility and open science.
> We would like to sincerely thank the Area Chair for their careful evaluation of our submission and responses, and for the considerable effort invested in ensuring a rigorous and fair review process.

---

> ### Author Response · Authors · 2025-12-04
>
> \[1\] ry L Goldberger, Luis AN Amaral, Leon Glass, Jef-221 frey M
> Hausdorff, Plamen Ch Ivanov, Roger G Mark,222 Joseph E Mietus, George B
> Moody, Chung-Kang Peng,223 and H Eugene Stanley. Physiobank,
> physiotoolkit,224 and physionet: components of a new research
> resource225 for complex physiologic signals. circulation, 101(23):226
> e215--e220, 2000.
>
> \[2\] gor Zyma, Sergii Tukaev, Ivan Seleznov, Ken Kiyono,228 Anton
> Popov, Mariia Chernykh, and Oleksii Shpenkov.229 Electroencephalograms
> during mental arithmetic task230 performance. Data, 4(1):14, 2019.
>
> \[3\] S Bbrinkm and W Cukierski. Upenn and mayo clinic's219 seizure
> detection challenge, 2014
>
> \[4\] Zahorodnii, A., Wang, C., Stankovits, B., Moraitaki, C., Chau, G.,
> Barbu, A., \... and Fiete, I. R. (2025). Neuroprobe: Evaluating
> Intracranial Brain Responses to Naturalistic Stimuli. arXiv preprint
> arXiv:2509.21671.
>
> \[5\] Zhang, D., Yuan, Z., Yang, Y., Chen, J., Wang, J., & Li, Y. (2023). Brant: Foundation model for intracranial neural signal. Advances in Neural Information Processing Systems, 36, 26304-26321.

---

### Meta-Review · Area_Chair_cLMc · 2025-12-18

**Summary:**

# **Summary of Reviewers’ Concerns**

Based on the feedback from multiple reviewers, the evaluation primarily focused on issues related to the interpretation of the scaling conclusions, the strength of experimental evidence, and the validation of key methodological assumptions. The main concerns include the following:

- **Limited connection between scaling conclusions and downstream task performance**
The reported scaling laws are mainly derived from self-supervised reconstruction loss during pretraining, while performance gains on downstream tasks are limited, leaving the practical significance of these scaling results unclear.

- **Consistency and comparability issues in Neuroprobe benchmark results**
Reviewers raised concerns about the sources of baseline results, version consistency, and result presentation in the Neuroprobe tasks, noting that the current description does not sufficiently support fair and transparent comparisons.

- **Evaluation tasks are relatively simple and insufficient to demonstrate scaling advantages**
The evaluation relies primarily on binary classification tasks with limited discriminative power. In some cases, improvements from linear feature engineering exceed those from model scaling, weakening the credibility of the scaling claims. In addition, large-scale EEG downstream tasks are lacking.

- **Difficulty disentangling architectural innovations from scaling effects**
  The paper introduces both new architectural designs and large-scale pretraining, but lacks systematic controls under identical architectures, making it unclear whether performance gains stem from architectural choices or scale.

- **Ablation studies indicate limited contribution from several components**
  Ablation results suggest that only a small subset of key modules consistently contributes to performance improvements, while other components show limited or unclear benefits at the evaluated scales.

- **Insufficient analysis of generalization across settings**
  Reviewers noted the absence of results across different subjects or alternative data-splitting strategies, which limits the assessment of the model’s generalization ability.

**Reviewer Concerns:**

# Overall Assessment of Remaining Issues

The paper is well motivated and addresses an important question, namely understanding the scaling laws of electrophysiology foundation models (EFMs) to guide future model design. However, after considering the reviewers’ feedback, the authors’ responses, and further analysis, several key issues remain insufficiently resolved.

## **1. Limited depth of the scaling conclusions**
The proposed scaling laws are primarily derived from self-supervised reconstruction loss during pretraining, with the core conclusion being that training smaller models for more epochs outperforms training larger models for fewer epochs. While this observation is formally articulated for the first time, it is arguably unsurprising to researchers in the EFM community, given the inherent data limitations and low signal quality of EEG signals (e.g., low SNR and distribution shifts). The paper does not substantially advance this observation by, for example, analyzing how scaling laws affect different downstream tasks or how architectural choices interact with scaling behavior across tasks. As a result, the conclusions remain largely empirical and do not fully address deeper questions of interest to the field.

## **2. Limitations of the evaluation tasks**
Although the paper uses large-scale EEG data for pretraining and evaluates on three EEG downstream tasks, validation on iEEG data is mostly limited to binary classification tasks. In contrast, prior EFM studies typically evaluate across a much broader range of EEG tasks (e.g., emotion recognition, motor imagery, epilepsy detection, sleep staging, attention, workload, and speech-related tasks), often spanning six or more task types and around ten datasets. Given the scale of EEG data used for pretraining, a more comprehensive set of downstream EEG evaluations would be necessary to convincingly validate the proposed scaling laws.

## **3. Difficulty disentangling architectural innovation from scaling effects**
The paper introduces both novel architectural designs and large-scale pretraining strategies, but lacks systematic controls under identical architectures. Because the proposed model differs substantially from the baselines in both architecture and training procedures, it is difficult to attribute the observed performance trends solely to data scale, model size, or training duration. Moreover, the paper does not examine whether the identified scaling laws generalize to other existing EFM architectures, raising concerns about the broader applicability of the conclusions.

## **4. Insufficient validation of joint EEG and iEEG modeling**
The paper treats EEG and iEEG as samples from a unified electrophysiological distribution and jointly uses them for pretraining. However, this assumption is not directly validated. There are no controlled ablation studies comparing EEG-only, iEEG-only, and joint pretraining. From a neuroscience perspective, EEG and iEEG differ substantially in signal characteristics, spatial resolution, electrode configurations, and information content, and are often considered distinct modalities. The paper does not provide sufficient evidence to justify the necessity or benefit of mixing these modalities during pretraining.

---

Overall, while the paper makes a meaningful attempt to study scaling behavior in EFMs, the above issues limit the depth, generality, and interpretability of its conclusions.

**Reviewer Scores:**

# **Reviewer Scores (AC Assessment)**

**Reviewer 5x2y**
There is a low probability that this reviewer would increase their score. Although the authors provided additional EEG downstream results, the core concerns regarding the limitations of the evaluation tasks and the limited depth of the scaling conclusions were not fundamentally addressed. As a result, an increase to a score of 4 appears unlikely.

**Reviewer MZwT**
The reviewer’s concern about the limitations of the evaluation tasks remains insufficiently resolved. Consequently, there is no realistic possibility of the score increasing to a high rating (e.g., 8).

**Reviewer i3iQ**
This reviewer primarily focused on the difficulty of disentangling architectural innovation from scaling effects. Since this issue was not substantively addressed in the rebuttal or supplementary experiments, a score increase is unlikely.

**Reviewer UXXE**
Although the authors provided additional experiments, the issue of limited evaluation tasks remains unresolved. Therefore, the probability of this reviewer increasing their score to 6 is low.

**Reviewer 6kz9**
This reviewer is likely concerned with both the limitations of the evaluation tasks and the difficulty of disentangling architectural innovation from scaling effects. Given that neither issue was fully addressed, a meaningful increase in score appears unlikely.

---

### Decision · Program_Chairs · 2026-01-26

Reject